

# Investigating extreme marine summers in the Mediterranean Sea

Dimitra Denaxa[1,2], Gerasimos Korres[1], Emmanouil Flaounas[1], and Maria Hatzaki[2]

[1] Hellenic Centre for Marine Research (HCMR), Greece
[2] National and Kapodistrian University of Athens, Department of Geology and Geoenvironment, Greece

*Correspondence to*: Dimitra Denaxa (ddenaxa@hcmr.gr)

**Abstract.** The Mediterranean Sea (MS) has been experiencing significant surface warming, particularly pronounced during summers and associated with devastating impacts. This study proposes the concept of Extreme Marine Summers (EMSs) and investigates their characteristics in the MS, using Sea Surface Temperature (SST) reanalysis data spanning 1950-2020. A marine summer may evolve as extreme, in terms of mean summer SST, under different SST substructures. Results suggest

that EMSs identified in most of the basin, are formed mainly due to the warmest part of the ranked daily SST distribution being warmer than normal. Areas where the warmest (coldest) part of the ranked daily SST distribution is more variable, experience EMSs primarily due to the contribution of the warmest (coldest) part of the distribution. Marine heatwaves (MHWs) within EMSs are more intense, longer lasting, and more frequent than usual, mainly in northern MS regions. However, the relative contribution of MHWs in EMSs is more pronounced in the central and eastern basin. Furthermore, a metric is proposed to

quantify the driving role of air-sea heat fluxes in forming EMSs. Results suggest that surface fluxes primarily drive EMSs in the northern half of the MS, while oceanic processes play a major role in southern regions. Upper ocean preconditioning is also found to contribute to the EMS formation. Finally, a detrended dataset was produced to examine how the SST multi-decadal variability affects the studied EMS features. Despite leading to warmer EMSs basin-wide, the multi-decadal signal does not significantly affect the dominant SST substructures during EMSs. Results also highlight the fundamental role of latent

heat flux in modulating the surface heat budget during EMSs, regardless of long-term trends.

## 1 Introduction

Global ocean has been experiencing intensive warming over the past decades. Environmental and societal implications –current and projected in the future– underline the need for continuous ocean monitoring and deeper understanding of the ocean climate, in terms of natural variability and anthropogenic climate change. Being the interface between ocean and atmosphere, Sea

Surface Temperature (SST) has key role in connecting ocean and global climate. A rapidly growing literature has been investigating links between SST and the intensification of atmospheric/oceanic events and processes, e.g., heavy precipitation events (Pastor et al., 2015), surface air temperature variations (Xu et al., 2019), marine heatwaves under global warming (Frölicher et al., 2018), increase in global wave power (Kaur et al., 2021). Importantly, SST is the oceanic parameter that regulates the air-sea energy exchanges, thus, reflecting the role of the ocean's thermal inertia (Deser at al., 2010). In this

context, SST has become a fundamental climatic variable and global climate change indicator.



The Mediterranean Sea (MS) is considered one of the most responsive and vulnerable areas to global warming (Giorgi et al., 2006; Lionello et al., 2006; Ali et al., 2022). In fact, the Mediterranean SST warming trends over the past decades largely exceed the observed global sea surface warming. Based on satellite SST recordings over the period 1982-2018, Pisano et al.
(2020) report a trend of $0.041 \pm 0.006°$ C/year for the Mediterranean SST. Over shorter time periods, Shaltout and Omested (2014) and Mohamed et al. (2019) based on satellite SST observations compute a mean warming trend for the MS of $0.035 \pm 0.007°$ C/year (1982–2012) and $0.036 \pm 0.003°$ C/year (1993–2017), respectively. Consistently, the Ocean Monitoring Indicator (OMI) produced in the framework of the Copernicus Marine Environment Monitoring Service (CMEMS) provides a rate of Mediterranean SST change over the period 1993-2021 of $0.035 \pm 0.002°$ C/year (EU Copernicus Marine Service
Product, 2022a). Over the same period, the corresponding CMEMS OMI product for the global average SST provides a linear trend of $0.015 \pm 0.001°$ C/year (EU Copernicus Marine Service Product, 2022b), i.e., 2.3 times lower than the Mediterranean SST trend. Similarly, Bulgin et al. (2020) find a mean warming trend of $0.09°$ C/decade for the global ocean based on satellite SST recordings for 1981-2018, again highlighting the MS as one of the areas exhibiting the highest warming trends.

On top of the documented SST trends, future climate projections suggest additional warming in the basin until the end of the 21[st] century (Adloff et al., 2015; Alexander et al., 2018; Soto-Navarro et al., 2020). CMIP6 multi-model future projections under high-emission scenarios (SSP5-8.5) suggest an SST increase relative to the period 1995-2014 ranging from $0.8°$ C up to $3.5°$ C in near-term (2021-2040) and long-term (2081-2100) future periods, respectively (Iturbide et al., 2021).

The well documented warming trends over the instrumental period spanning 1980-present are also part of the significant multi-decadal variability of the Mediterranean SST. Marullo et al. (2011) first evidenced an approximately 70-year period SST oscillation in the basin in coherence with the North Atlantic Oscillation (NAO) and Atlantic Multi-decadal Oscillation (AMO). Several studies followed, suggesting different mechanisms regulating the observed multi-decadal SST fluctuations in the basin. Mariotti and Dell'Aquila (2012) attributed the transmission of AMO variability to the MS to atmospheric processes, while
Skliris et al. (2012) suggested an oceanic origin of the AMO signal transmission in the basin. The source of the Mediterranean SST multi-decadal variability is still an open question. Pisano et al. (2020) showed that the linear increase of the Mediterranean SST observed over the satellite era closely follows AMO only until 2007. By that time, AMO has entered a declining phase, in agreement with the observed upper ocean cooling in the Atlantic that reversed the previous warming trends (Robson et al., 2016). Regarding the origin of the SST multi-decadal variability in the MS, Yan et al. (2021) showed a better consistency
SST anomalies with a time-integrated NAO index, suggesting the accumulative effect of NAO atmospheric forcing on ocean circulation in the basin. Based on a stable-lag (of approximately 8 years) among the two quantities and having regressed out the signal related to anthropogenic forcing, they conclude that the cumulative NAO index may be used to predict the Mediterranean SST multi-decadal variability.



Along with the surface warming, warm extreme oceanic events such as Marine Heatwaves (MHW), have attracted great research interest. Increased MHW intensity and frequency have been documented based on observational and modelled SST datasets for the past decades (Oliver et al., 2018; Holbrook et al., 2019; Darmaraki et al., 2019a; Juza et al., 2022; Pastor and Khodayar, 2023; Dayan et al., 2023), while further enhancement is expected at global and Mediterranean scale (Oliver et al., 2019; Darmaraki et al., 2019b; Plecha and Soares, 2019; Hayashida et al., 2020). Future projections particularly suggest that

anthropogenic forcing will drive an increase in MHW frequency by 2100 (pronounced under high-emission scenarios), compared to the naturally expected MHW occurrence, i.e., in a climate with natural forcing only (Oliver et al., 2019).

The growing research interest in ocean warming trends and warm extremes is highly motivated by their detrimental impacts on marine life. Gradual warming or acute thermal stress (e.g., during a MHW) can greatly threat marine ecosystems through

direct (e.g., mortality events, decreased abundance of marine species) or indirect (e.g., nutrient availability, biodiversity) effects (Smale (2020) and references therein). Impact assessment studies focusing on past MHWs have documented massive geographical shifts, coral bleaching, diminished fertilization success of certain species, impacts on primary productivity, habitat loss (Garrabou et al., 2009; Pearce and Feng, 2013; Wernberg et al., 2016; Frölicher and Laufkotter, 2018; Smale et al., 2019; Leach et al., 2021; Garrabou et al., 2022; Smith et al., 2023). Climate-related local extinctions are already a fact and

are expected to further increase over the next decades under increasing global warming (Wiens, 2016). Such findings point out the need to better understand the stressful conditions experienced by marine ecosystems due to abnormally high SSTs, caused either by short-lasting heat anomalies or elevated temperatures persisting for longer periods.

Summer periods are of particular interest as they are associated with greater surface warming both in present and future climate

studies. Actually, the Mediterranean warming rates calculated for the past decades are greater during summer months (Giorgi and Lionello, 2008; Gualdi et al., 2013; Pastor et al., 2020; Pisano et al., 2020). Additionally, model projections suggest that maximum SST increase is expected in summers (López García and Belmonte, 2011; Shaltout and Omested, 2014; Giorgi and Lionello, 2018). Moreover, summer MHWs present the highest SST anomalies (Gupta et al., 2020) and are associated with a stronger ecological footprint (Oliver et al., 2019). For instance, mass mortalities in the MS have been attributed to thermal

stress during summers (Coma et al., 2009; Rivetti et al., 2014). Indeed, "warm range edge" species are the most vulnerable to high SST anomalies (Smale et al., 2019). In addition, climate projections highlight the summer season as the one expected to present significant increase (decrease) of extremely high (low) SSTs (Alexander et al., 2018).

Considering the above, we propose the concept of extreme marine summer (EMS), from ocean perspective, and we put our

effort to explore Mediterranean EMSs in a climatological framework. We define EMSs as the warmest summers, in terms of mean summer SST, within a study period at a specific location (see Methods). EMSs are expected to be related to warm events, as an EMS may emerge from a MHW occurrence, yet fundamental differences exist. MHWs typically last for days and may reach several weeks according to their causing factors and the concurrent atmospheric and oceanic conditions regulating their



duration. On the other hand, an extreme marine season is, by definition, of fixed duration. A marine summer may evolve as extreme due to uniformly increased SST values throughout its duration or due to changes in substructures of SST distributions, e.g., due to warmer SSTs of extreme events alone. This relates to the fact that marine species have different ways to adapt to thermal stress depending on their sensitivity (thermal tolerance) and the duration and intensity of temperature anomalies. For instance, animals able to migrate to avoid anomalously warm water conditions lasting for several days may not be able to cope with longer lasting heat stress (Alexander et al., 2018). Therefore, even in absence of extreme SST values or extreme warm

events, a summer season may present extreme mean conditions thereby strongly affecting marine life.

An EMS may arise due to elevated SST anomalies taking place over the summer duration as well as due to favoring initial thermal conditions. The former may result from the interplay of atmospheric and oceanic factors, being the air-sea heat exchanges (turbulent and radiative fluxes), horizontal (Ekman and geostrophic currents) and vertical (entrainment, Ekman

pumping) advection (Deser et al., 2010 and standard oceanography references therein). Several studies focusing on the Mediterranean SST variability have shown the crucial role of air-sea heat fluxes in SST variability and observed warming trends in particular (e.g., Skliris et al., 2012; Shaltout and Omstedt, 2014). In this context, an extra focus is put within this study on understanding what is the driving role of air-sea heat fluxes in the formation of EMSs. The role of initial conditions potentially favoring the occurrence of an EMS is additionally examined by means of a proposed preconditioning index.


Aim of this study is to explore Mediterranean EMSs, focusing on four objectives. The first is to understand how daily SST values are commonly structured within EMSs in the basin. Our second objective is to investigate the role of MHWs during EMSs. The third is to investigate physical mechanisms related to the EMS formation focusing on the driving role of air-sea heat fluxes, while briefly examining additional factors, such as wind speed, mixed layer depth and preconditioning. Finally,

the fourth objective is to understand the role of SST multi-decadal variability in the observed EMS characteristics.

The paper's content is organized as follows: Section 2 presents the datasets and methods used in the study. Section 3 discusses results on the SST substructures in EMSs (Sect. 3.1), the role of MHWs in EMSs (Sect. 3.2) and potential EMS drivers focusing on air-sea heat flux (Sect. 3.3). The role of multi-decadal variability is discussed throughout the different sub-sections. Finally,

Sect. 4 summarizes key findings and conclusions.

## 2. Data and methods

### 2.1 Datasets

In this study, we use SST fields from the ERA5 Reanalysis of the European Centre for Medium-Range Weather Forecasts (ECMWF) for the period 1950-2020 (Bell et al., 2020; Hersbach et al., 2023). This dataset corresponds to foundation SST

(SST free of diurnal variations) and derives from a combination of HadISST2 and OSTIA datasets. Atmospheric variables



from the ERA5 product are also used to investigate their role in the formation of EMSs: 10-meter wind speed, net short- and long-wave radiation at the sea surface, latent and sensible surface heat fluxes, total cloud cover, specific humidity. All fields have a grid spacing of 0.25°x0.25° in longitude and latitude and are provided in hourly time intervals. In addition to ERA5 (reference SST dataset), we use the CMEMS L4 satellite SST product (EU Copernicus Marine Service Product, 2022c) for the period 1982-2019, at 0.05°x0.05° grid spacing, to cross check the quality of the reference dataset against a high-resolution observational SST dataset. Finally, mean monthly values for the mixed layer thickness and the ocean temperature are extracted from the CMEMS Physics Reanalysis product (EU Copernicus Marine Service Product, 2022d) for the period 1987-2019 at 0.042°x0.042° spatial resolution, in order to compute the ocean heat content (OHC). CMEMS mixed layer depth (MLD) is additionally used to examine stratification conditions during EMSs.

## 2.2 Methods

### 2.2.1 Extreme marine summer definition and associated SST substructures

We consider July-August-September (JAS) as the marine summer season, i.e., the warmest part of the SST annual cycle in the MS (Pastor et al., 2020). We then define EMSs, separately at each grid point, as the four summers with the highest average JAS SST, i.e., exceeding the 95[th] percentile of the 71 available summer periods from 1950 to 2020.

After identifying EMSs, we apply the methods of Röthlisberger et al. (2020) to assess the associated SST substructures at each grid point. For this reason, we first rank all daily SST values within each summer (lowest daily SST, second-lowest daily SST and so on) as shown in the illustrative example of Fig.1. Then, we compute the anomaly of each ranked daily SST (named as Rank Day Anomaly - RDA from now on) with respect to the ranked daily mean value (Fig. 1c). To gain insight into the mean EMS state at each grid point, we compute the mean EMS RDA value over the four locally identified EMSs. By splitting the summer rank days into different parts of the SST distribution (e.g., the coldest and the warmest half), we quantify the relative contribution of each part to the observed EMS RDA. As a result, we diagnose if an EMS is formed e.g., due to the highest (lowest) rank days, i.e., warmest (coldest) days, being anomalously warm, or due to a uniform shift of the SST seasonal distribution. The example of Fig. 1a shows for a certain grid point that the summer of 2003 has been entirely warmer than usual. RDAs in this example (Fig. 1c) are much higher for the higher rank days, suggesting that this summer is identified as extreme primarily due to the warmer summer days being warmer than usual.





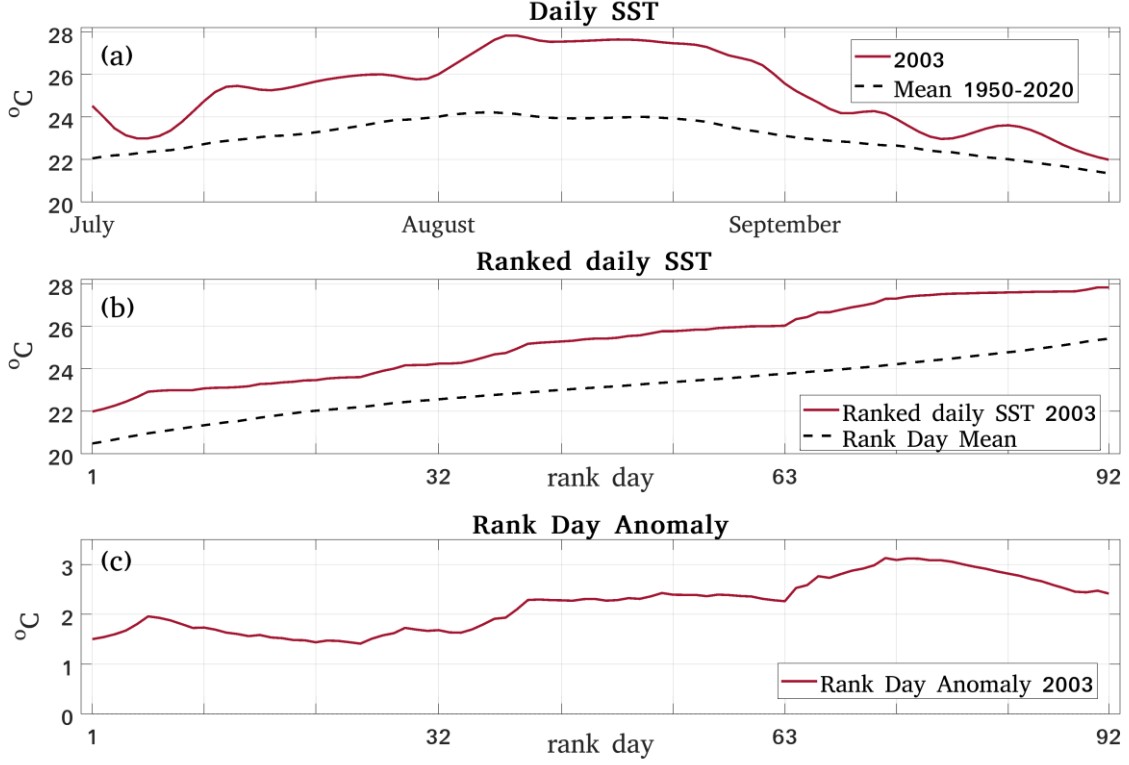

**Figure 1: a) Daily SST time series at a grid point in the Ligurian Sea (42.5° N - 8° E) for the extreme marine summer 2003 and for the mean 1950-2020 state (solid red and dashed line, respectively); b) Ranked daily SST values of panel (a) for 2003 and rank day means for 1950-2020 (solid red and dashed line, respectively); c) Anomaly of each ranked daily SST with respect to the corresponding rank day mean value for 1950-2020. There is no temporal order in the horizontal axes of panels (b) and (c).**

**2.2.2 Detrending the time series**

Figure 2a presents the summer ERA5 SST time series from 1950 to 2020 in the MS. The observed surface warming trend across the basin during the last decades directly impacts the detection of EMSs, according to the EMS definition introduced in Sect. 2.2.1. Therefore, the identified EMSs at each grid point of the domain tend to fall within the last few decades, especially in the post 2000 period. To examine characteristics and drivers of EMSs independently of long-term changes, we detrend the

SST dataset by removing the multi-decadal trend. For this reason, we apply separately at every grid point the Empirical Mode Decomposition method (EMD; Wu and Hu, 2006). In this way, we compute at each grid point, the multi-decadal trend of the mean summer SST values of the period 1950-2020. The produced trend is then subtracted from the mean summer SST time series creating a 71-year length SST anomaly time series for each grid point, as in the example presented in Fig. 2b for the basin-averaged summer SST time series. These are the time series used for the EMS detection. The trend value of each summer

season is also removed from all days belonging to this summer to obtain a detrended summer dataset of daily SST values as





well. Using the same methodology, we additionally create detrended datasets of mean summer values for the atmospheric variables used in this study (10-meter wind speed, net short- and long-wave radiation at the sea surface, latent and sensible surface heat fluxes).

Working with the detrended SST time series, the EMS identification becomes independent of the Mediterranean SST multi-decadal oscillation, the warming trend observed over the recent decades (i.e., climate change signal), and any long-term variability of the Mediterranean SST. In this study, we use both the original and the detrended data in order to i) gain insight into the actually warmest EMSs and ii) understand the role of the subtracted signal, respectively. Differences are discussed throughout the following sections.


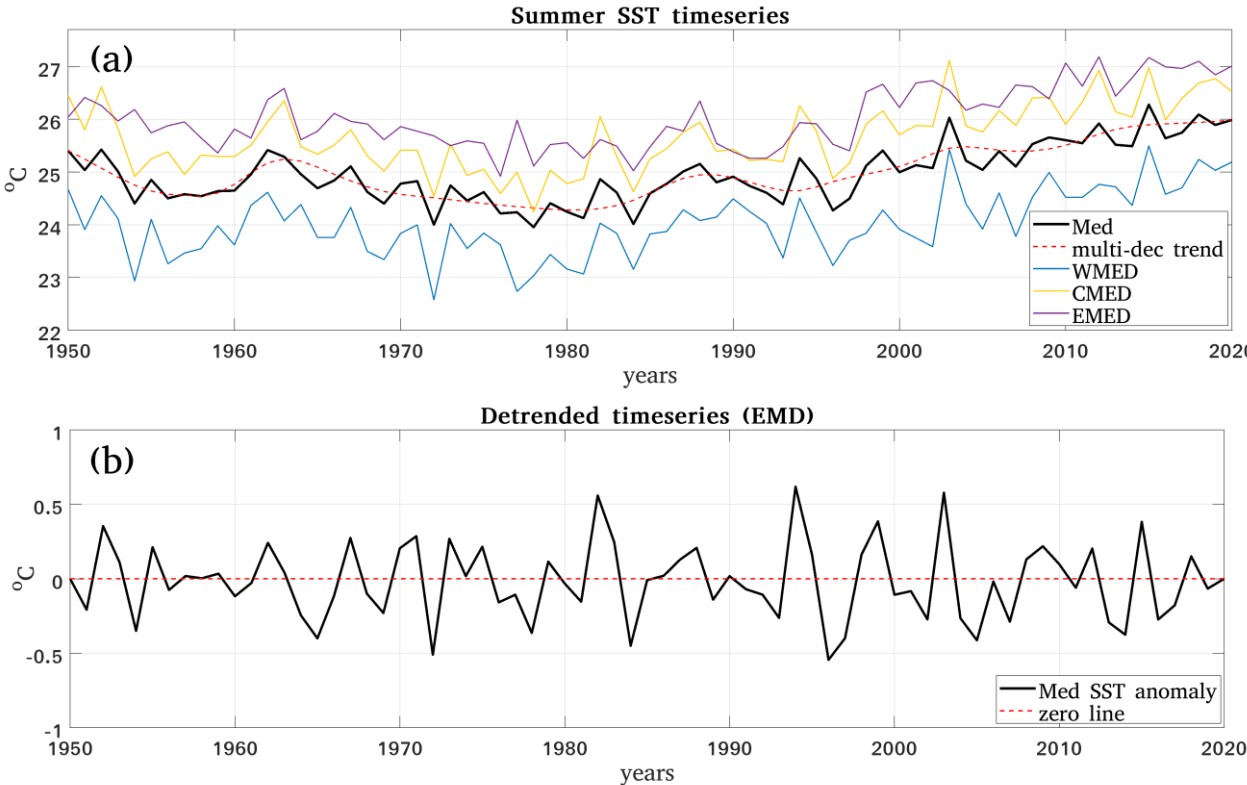

**Figure 2: Example of applying the Empirical Mode Decomposition (Wu and Hu, 2006) to remove the multi-decadal trend from summer ERA5 SST data: a) Domain-averaged mean summer SST values and multi-decadal trend (black line and red dashed line, respectively). Mean summer SST time series averaged for the western (extending from Gibraltar Strait to the Strait of Sicily), central (Strait of Sicily up to 22° E) and eastern (22° E eastwards) sub-basins are also depicted in blue, yellow and purple lines, respectively; b) Detrended summer SST time series for the MS derived from subtracting the multi-decadal trend from the domain-averaged time series of panel (a).**



### 2.2.3 Marine heatwave identification

MHWs are discrete events lasting at least five consecutive days, with temperatures exceeding a percentile-based (90%) threshold (Hobday et al., 2016). MHW detection was performed on the daily ERA5 SST dataset for the period 1950-2020,

based on the MHW definition and detection methodology of Hobday et al. (2016) and using the matlab toolbox provided by Zhao and Marin (2019). The selected reference period used to create the daily climatology is 1983-2012. The choice of the climatological period follows the general recommendation of a minimum of 30-years period for the computation of the climatology and thresholds for MHW detection (Hobday et al., 2016). Sensitivity tests using different starting/ending points show some differences in the resulting MHW properties (not shown). However, the MHW analysis in this study aims to

investigate potential changes in main MHW characteristics during extreme seasons, rather than provide a thorough discussion of MHW properties in the MS. Therefore, such discrepancies resulting from different climatological periods are not considered significant in the scope of the present study. In addition, the availability of satellite SST data during the selected climatological period allowed for detecting MHWs also using observational data. In this respect, MHW detection results based on modelled (ERA5) and observational (CMEMS) SST were found very similar (not shown). For consistency reasons, the same

climatological period was used when applying the same methodology to detrended SST data. Finally, to focus on summer MHWs, we isolated events with their onset and end day falling within the JAS summer period.

### 2.2.4 Surface heat fluxes in extreme marine summers

This Section presents the methodology we followed to study the role of air-sea heat fluxes as potential EMS drivers. Heat exchange at the air-sea interface may be quantified through the net surface heat budget equation, as follows:

$$Q_{net} = LH + SH + SWR_{net} + LWR_{net} \quad (1)$$

where $LH$, $SH$, $SWR_{net}$ and $LWR_{net}$ are the latent and sensible heat fluxes, and the net short- and long-wave radiation, respectively.

To investigate the role of surface heat fluxes during EMSs, we first compute the mean anomaly of EMS $Q_{net}$ and its

components with respect to their mean summer value over the study period. Notably, mean summer $Q_{net}$ anomaly values represent the cumulative effect of surface fluxes on the sea surface within a summer season (i.e., a heat surplus or deficit relative to climatology). Specifically, sequential warming and cooling effects of $Q_{net}$ on the sea surface are expected to occur during a season and, in turn, be enhanced or counterbalanced by atmospheric and/or oceanic processes. Therefore, a positive (negative) seasonal $Q_{net}$ anomaly of an EMS does not necessarily reflect a driving (opposing) role of $Q_{net}$ in forming this

EMS. To answer the question "*what is the driving role of air-sea heat fluxes in the formation of EMSs?*" we need to focus on the role of $Q_{net}$ in forming the SST anomalies that are responsible for a summer to evolve as extreme. On these grounds, in addition to examining seasonal heat flux anomalies, we propose a metric to quantify the contribution of $Q_{net}$ during selected summer sub-periods. During the selected sub-periods, SST evolves towards greater SST anomalies, either through warming



occurring at a greater rate than usual or through cooling occurring at a lower rate than usual. More detailed description on the
rationale and methodological steps for constructing the metric is included in Appendix A.

### 2.2.5 Extreme marine summers preconditioning

Apart from the contribution of positive SST anomalies during the season, warmer than usual initial conditions may also favor
the formation of an EMS. To explore the potential role of upper ocean preconditioning in the development of an EMS, we
examine the oceanic thermal conditions of the mixed layer prior to the JAS period. We choose June as the reference period to
calculate the ocean heat content (OHC), considering it as indicative of the initial conditions of the surface ocean "thermal
state" for each summer. We compute the OHC of the mean mixed layer depth (MLD) for every June of the 71-years period
and the corresponding EMS anomalies (June before an EMS – mean June for all years) at each grid point.

OHC in this task is computed based on the following equation:

$$OHC = \iiint \rho C_p T dx dy dz \qquad (2)$$

where $\rho$ is the water density, $C_p$ is the water heat capacity and $T(x, y, z)$ is the 3D monthly mean (June) ocean temperature
taken from the CMEMS Physics Reanalysis dataset. As the integration depth (MLD of each June) at each location varies
interannually, vertical integration in Eq. 2 makes use of all $T$ values that are available above the MLD of the specific location
and month. To overcome any unfair OHC inter-comparisons due to different integration depths (thus, different volumes of
water) from summer to summer, we normalize all produced OHC values by dividing them with the mean MLD of the respective
location and month. The produced anomalies are considered to be a (qualitative) preconditioning index for the EMS formation.

## 3. Results

### 3.1 Extreme marine summer substructures

#### 3.1.1 Decomposing the extreme marine summers

The average Rank Day Anomaly (RDA) of the four EMSs are shown in Fig. 3a. The maximum EMS RDAs reach up to 2.5°
C in the western basin, within the Gulf of Lions and the Ligurian Sea and extending to the south, surrounding Sardinia. The
Tyrrhenian and the central Adriatic Seas follow with maximum anomalies reaching up to 1.8° C. In the rest of the basin, RDAs
are of the order of less than 1.4° C, with its minimum values found in the southernmost areas of the Ionian Sea.

To gain insights into the SST substructures of different EMSs, in Fig. 4 we provide examples of different patterns for three
grid points of the domain (locations marked on maps of Fig. 3). In particular, Fig. 4a shows the SST substructures for a grid
point located in the Gulf of Lions (41.5° N-5° E). Mean EMS RDA at this location reaches up to 1.86° C. Regardless the
variability of the SST substructures, the largest contribution to the mean EMS RDA at this location comes from the warmest
part of the SST distribution (higher rank days), with the exception of EMS 1999 where the middle part contributes the most.





The EMS 2003 stands out with the larger mean RDA among all EMSs, primarily formed by the 60 warmest rank days. In the

same location, the summer of 2020 presents a similar SST substructure but with anomaly values of smaller amplitude. The

EMSs 1999 and 2018 present the smallest and the second-largest mean RDA, respectively. Both years present a varying

contribution to the corresponding mean RDA in the different parts of the SST distribution at this location.

Same as the location in the western basin, the grid point in the central Levantine (35° N-28° E) presents higher RDAs in the

warmest rank days (Fig. 4b). However, RDAs for all four EMSs at this location present a common substructure with a mean

contribution of 1.18° C, where slightly higher (lower) contribution comes from the warmest (colder) ranked days. The four

EMS years here fall into a more recent period of our climatology compared to the ones identified in the other example locations.

This is due to the non-uniform warming trend in the MS. In particular, the eastern part of the basin (EMED) presents larger

positive SST trends (Shaltout and Omstedt, 2014; Pastor et al., 2020; Pisano et al., 2020). Consequently, when using the

original (non-detrended) dataset we expect the EMSs in the EMED to fall in more recent years than the ones identified in the

WMED.

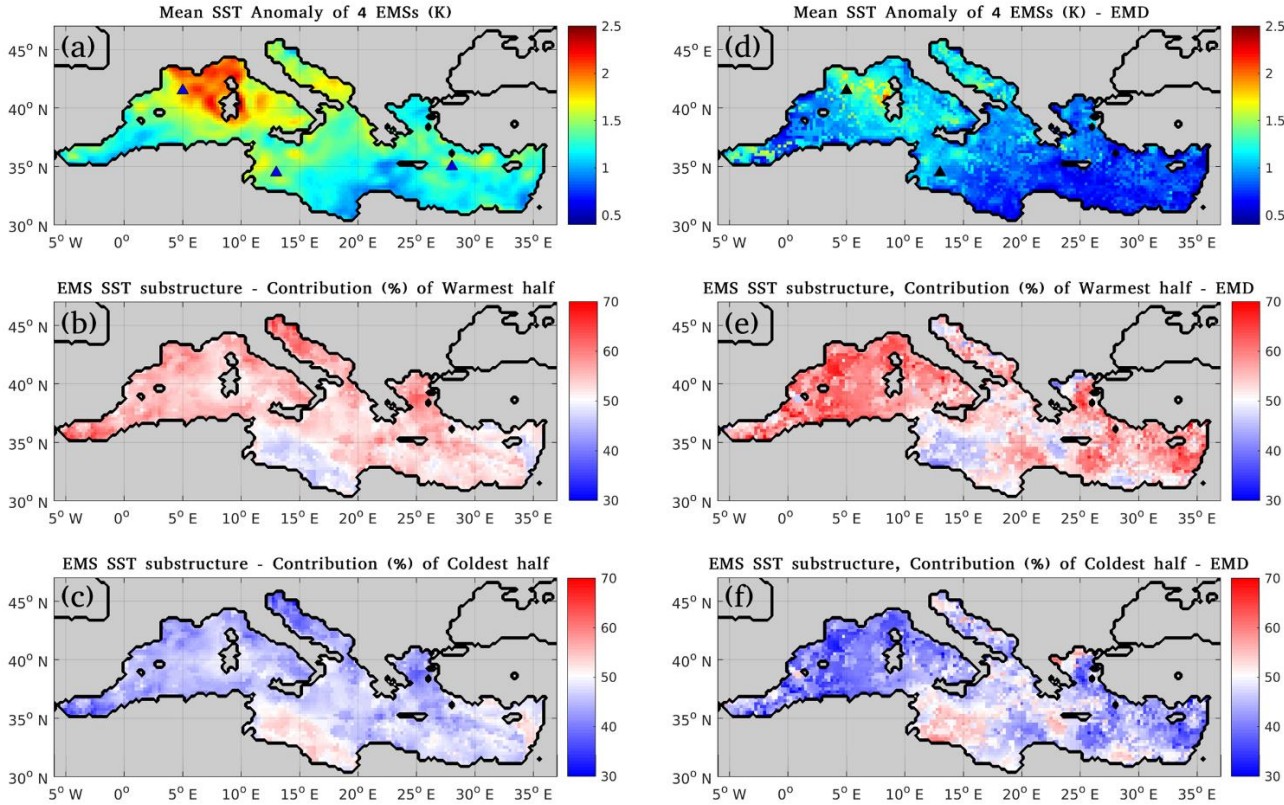

**Figure 3: Left: a) Mean Extreme marine Summer (EMS) Rank Day Anomaly (RDA) field based on ERA5 SST 1950-2020 b) Percentage of contribution to the EMS RDA of the warmest half of the EMS RDA distribution c) Percentage of contribution to the EMS RDA of the coldest half of the EMS RDA distribution Right: Same as a, b, c but using the detrended dataset. Blue triangles represent the locations of the selected grid points discussed in this section.**



Figure 4c presents the EMS SST substructures at a grid point located North of the Tunisia coasts (34.5° N-13° E). In contrast to the other two example locations, the largest contribution to the mean EMS RDA (1.4° C) here is due to the coolest 30 days

of these summers being anomalously warm. This is observed in three (out of the four) EMSs, while the EMS 2015 presents an almost uniform positive SST shift during the season. The year 2003 stands out again with the largest mean RDA, with its maximum value occurring in the lowest rank of this summer (2.6° C).

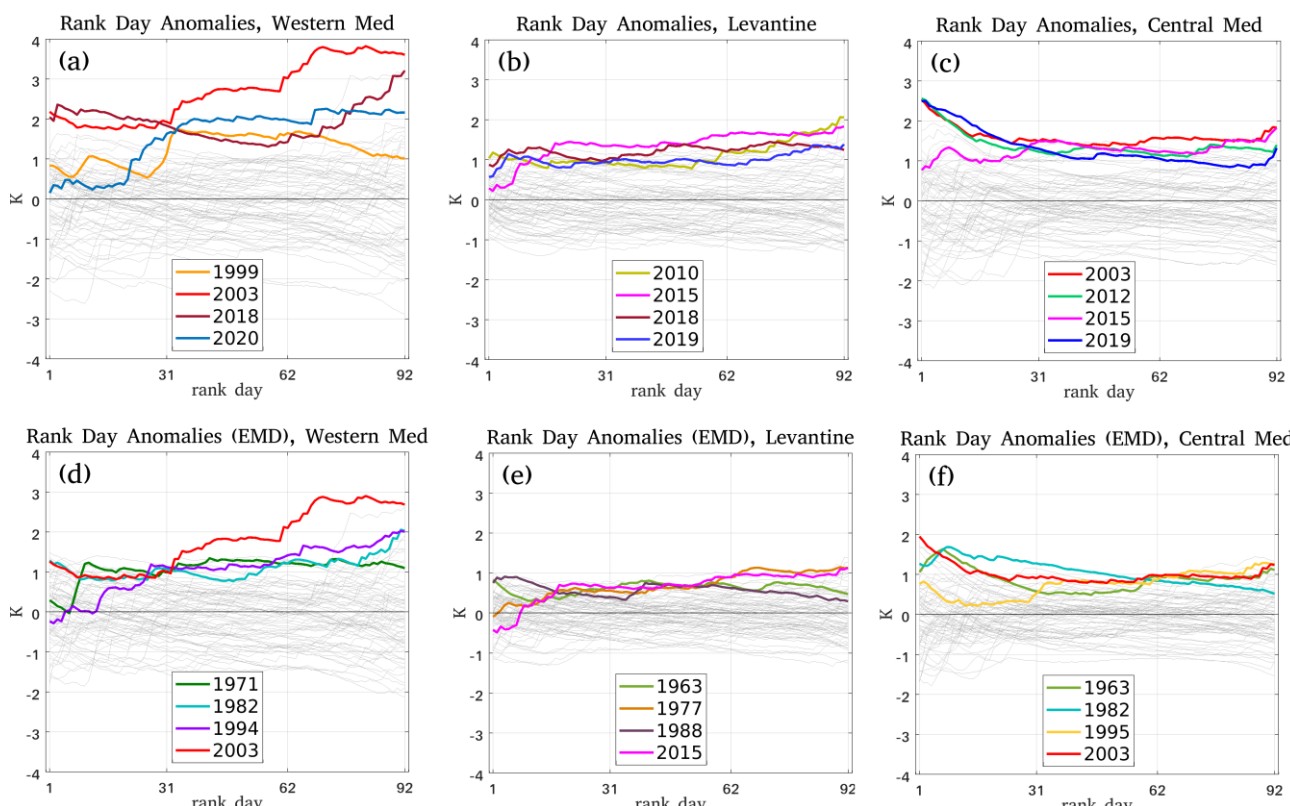

**Figure 4: SST substructures at three example locations: a) southern of the Gulf of Lions (41.5° N-5° E), b) in the central Levantine Sea (35° N-28° E) and c) eastern of the Gulf of Gabes (34.5° N-13° E) using the original (top row) and the detrended (bottom row) summer SST dataset. Colored lines stand for EMS, grey lines for non-EMS. Horizontal axis shows the 92 rank days of the JAS summer season (there is no temporal order).**


To gain insight into the SST substructures over the basin, Figs 3b and 3c show the fractional contributions of the warmest and the coldest half of the RDA to the formation of EMSs. The warmest half contributes the most to almost the entire basin. This suggests that EMSs in the MS are commonly formed due to the warmer summer days being warmer than normal. The maximum values of the contribution percentages of the warmest part reach up to 70% (North Adriatic, Alboran Sea).





On the other hand, the coldest half contributes the most, with an order of 55%, only within an area of the southern-central basin, close to the African coasts (Figs 3b,c). These results distinguish the EMSs in this area being primarily formed due to the suppression of the colder summer days, i.e., due to the colder days being warmer than normal (further discussed in Sect. 3.3).

### 3.1.2 Role of rank day SST variability in SST substructures

To examine the role of the local rank day SST variability in forming the observed EMS substructures, we first compute the variance of the RDA of all summer seasons within the study period (Fig. 5a). The spatial distribution of the variance of RDA is remarkably similar to the mean EMS RDA field in Fig. 3a. In particular, large seasonal SST anomalies are found, as generally expected, in locations of large rank day SST variability. Results come in agreement with Shaltout and Omstedt (2014) who examined the SST variability in the basin showing that the maximum and minimum seasonal stability are found close to the southern Levantine sub-basin and the Gulf of Lions, respectively.

We then calculate the fractional contribution of the coldest and the warmest half of the ranked summer days to the RDA variance for all 71 summers (Figs 5b,c). The contribution of the warmest (coldest) half of all daily summer SST values to the RDA variance is very similar to the contribution of the warmest (coldest) half of the EMS RDAs to the total EMS RDA (Figs 5b,c vs Figs 3b,c). In other words, the locations where the warmest (coldest) part of the rank day distribution has higher spread, experience EMSs primarily due to the contribution of the warmest (coldest) part of the EMS rank day distribution.

This may be observed in the example cases presented in Figs 4a-c, considering also the grey lines that correspond to the rest of the years (non-EMS years). At the location at 41.5° N-5° E (Fig. 4a), SST substructures appear particularly different among the EMSs. However, the warmest part of the SST distribution contributes to the EMS RDA more than the coldest part (as clearly appears in Fig. 3b). This is the part presenting the largest RDA spread in the rest of the years as well (6.7° C and 5.4° C for the warmest and the coldest part, respectively) (Fig. 4a). Similarly, at the location at 35° N-28° E (Fig. 4b), the highest rank days which contribute the most to the mean EMS RDA are also the most varying from summer to summer within the study period. The RDA range of the warmest part here is 3.3° C (2.7° C for the coldest part). In contrast, at the location 34.5° N-13° E (Fig. 4c), which exhibits greater RDA variability in the first rank days of the RDA distribution, RDA values of the coldest half display a range of 4.6° C (3.5° C for the warmest part) and contribute the most to the mean EMS RDA. In all




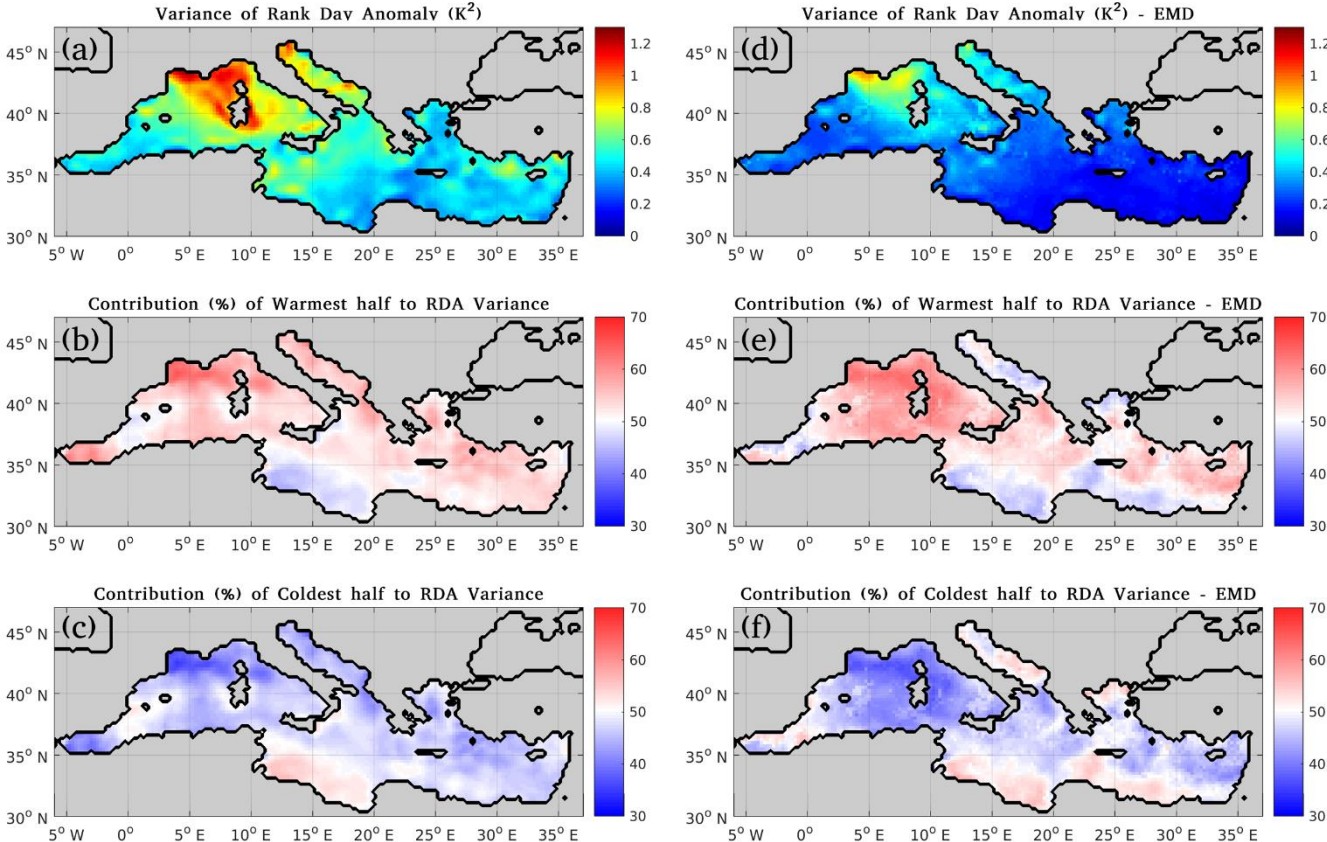

**Figure 5: Left: a) Variance of RDA based on all summers within 1950-2020 b) Percentage of contribution to the RDA Variance of the warmest half of all summer rank days within 1950-2020 and c) Percentage of contribution to the RDA Variance of the coldest half of all summer rank days within 1950-2020. Right: Same as a, b, c but using the detrended dataset.**

cases, the part of the RDA distribution that contributes the most to the EMS RDA is the one presenting the largest spread climatologically.

### 3.1.3 SST substructures using detrended data

When the multi-decadal trend is removed, smaller SST anomalies in EMSs are found in the entire MS. (Figs 3a,d). The original and detrended RDA fields differ by approximately 0.5° C. This difference is expected, as the detrended dataset is free of the climate change effect warming the MS. The spatial distribution of the RDAs and the corresponding fractional contributions of the coldest or warmest parts of the SST distribution are similar to the ones obtained when using the original dataset (left vs right column in Fig. 3). Despite the difference in the RDA magnitude, both fields in Fig. 3a and Fig. 3d, depict a west-east RDA gradient with the largest RDA values in the WMED Sea. In both fields, RDA peaks are found in the Gulf of Lions and





the Ligurian Sea, followed by the Adriatic Sea. Similarly, the contribution of the coldest and warmest part to the EMS RDA present very small differences (Figs 3b,c compared to Figs 3e,f).

This RDA spatial pattern (west-east gradient) is also present in the RDA variance field of the detrended dataset (Fig. 5a vs Fig. 5d). This suggests that a similar rank day variability pattern in the two datasets modulates the seasonal SST anomalies observed in EMSs. Additionally, the contribution fields of the two parts of the RDA distribution of all summer seasons to the RDA variance present only slight differences in certain areas compared to the original dataset (Figs 5b,c vs Figs 5e,f). In particular, in the detrended dataset, locations where the coldest rank days contribute to the RDA variability more than 50% are not

restricted in the southern-central area north of the African coasts as in the original dataset. It is also the central Adriatic, part of the southern Levantine and the north Aegean Seas that exhibit a slightly enhanced contribution of the cool summer days to the RDA variability (Figs 5e,f).

It is noteworthy that the observed common spatial pattern of SST substructures in the original and the detrended dataset (Fig.

3) results from different combinations of EMSs. To better illustrate this, the SST substructures in the three example cases discussed above are very similar for the original and the detrended dataset, despite the actual EMSs being warmer (Fig. 4a-c and Fig. 4d-f, respectively). Three out of the four identified EMS years in all example locations differ between the original and the detrended dataset. Despite this, at each location, the part of the SST distribution contributing the most to the RDA is the same in the two datasets, and it is the most variable climatologically.


In addition, SST substructures in EMSs seem to be independent of the selected study period. Sensitivity tests performed for different sub-periods show a consistent statistical behavior of the detrended SST dataset compared to the original one (not shown).

This analysis suggests that EMSs are formed based on a "background" SST substructure field, largely depending on the climatological ranked daily SST variability in the MS. On top of this field, the multi-decadal signal adds extra warming in the basin, resulting in warmer EMSs.

**3.2 The role of marine heatwaves in extreme marine summers**

Analyzing the EMS SST substructures in the previous section revealed that EMSs in the basin are commonly formed due to

the warmer summer days being warmer than normal. To complement these findings, this section investigates the role of MHW events during EMSs. We first present basic MHW properties during summers in the MS, as obtained from the MHW detection applied on the SST reference dataset (Sect. 3.2.1). Then, we examine the relative role of MHWs in EMSs by means of changes in MHW properties during EMSs (Sect. 3.2.2) with respect to mean MHW conditions. Finally, we examine the role of the SST multi-decadal trend on these findings (Sect. 3.2.3).



### 3.2.1 Detection of summer marine heatwaves in ERA5


In this section, we present the main properties (intensity, duration, frequency of occurrence) of the summer MHW events detected in the period 1950-2020 in the MS. MHWs are more intense in the northern parts of each sub-basin (Fig. 6a). Mean summer MHW intensity (mean SST anomaly during a MHW with respect to local climatology) reaches its maximum values (approximately $2.6°$ C) in the northwestern part of the basin. The Adriatic Sea displays the second highest MHW intensity

(approximately $2°$ C). Values of about $1.5-2°$ C mostly appear within the latitudinal zone between $35°$ N and $42°$ N in the Tyrrhenian, the Ionian and the Aegean Seas. The lowest mean intensity appears off the African coasts, extending from $15°$ E eastwards. Striking similarities between the MHW mean intensity (Fig. 6a) and the SST RDA variance field (Fig. 5a) reflect the expected dependency of warm extremes on the local SST variability. This is observed particularly in the north-western regions, indicating that the more variable the daily SST values over the studied summers are, the more intense both EMSs (as

shown in Sect. 3.1.2) and MHW events are.

Average MHW duration in the largest part of the basin ranges between 10 and 15 days (Fig. 6b). Longer lasting events (up to 20 days) are found in discrete spots in the Tyrrhenian, the Ionian and to a lesser extent in the southern Aegean Sea. It is noteworthy that areas such as the Gulf of Lions, the Adriatic and the North Aegean Sea, where MHW intensity is high, tend

to present shorter mean duration values, varying between 8-12 days. Concerning summer MHW frequency, in most areas a MHW occurs approximately every 2 summers (Fig. 6b), while the rest of the basin experiences MHWs slightly more or less frequently (every 1-3 years).



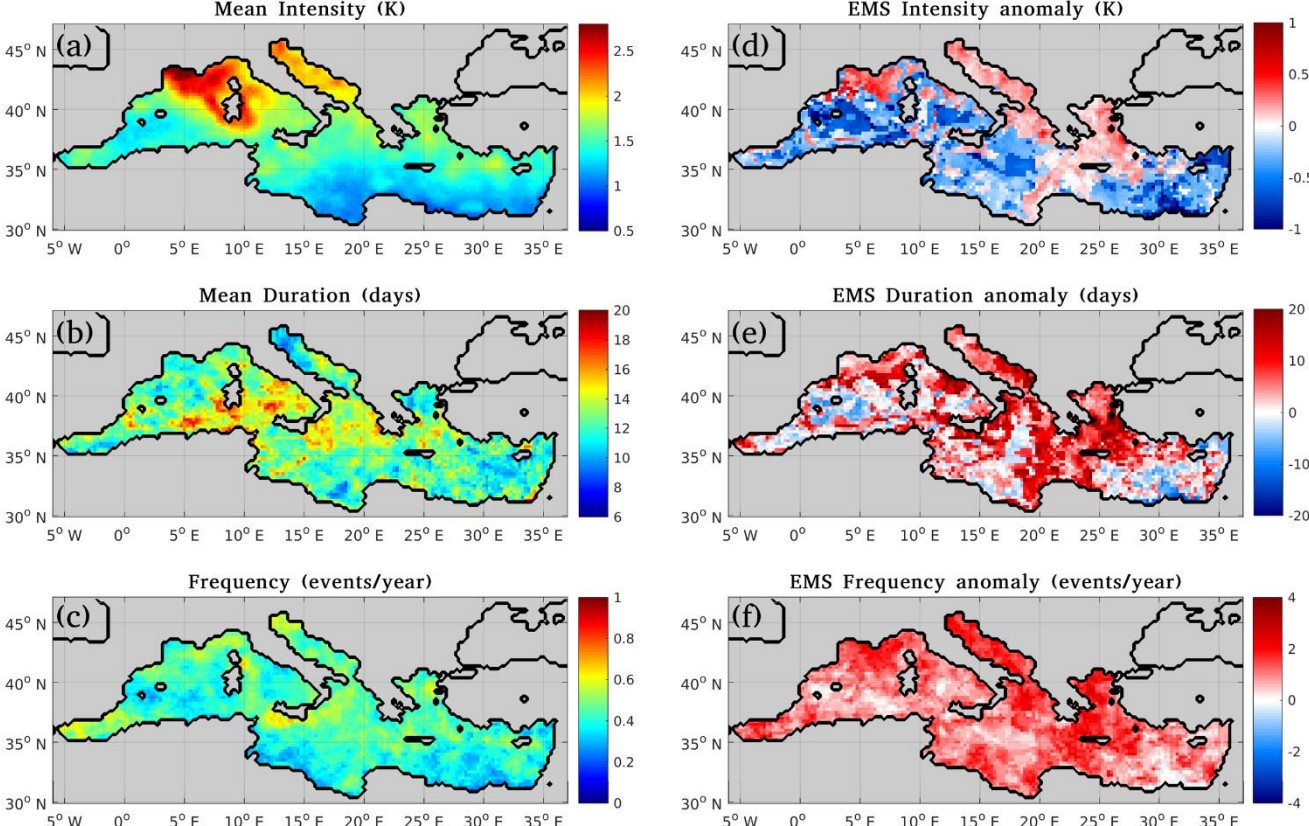

**Figure 6: Summer Marine Heatwave (MHW) properties in the Mediterranean Sea in the period 1950-2020 and their anomalies in extreme marine summers (EMS). Left: Mean Intensity (a), mean duration (b) and mean frequency of occurrence (c); Right: EMS anomalies of mean Intensity (d), mean duration (e) and mean frequency of occurrence (f) with respect to the mean summer values displayed in the left column.**

Our results on MHW analysis are in agreement with prior studies. Indeed, Darmaraki et al. (2019a) showed that mean MHW
intensity computed from observations and model data for the period 1982-2017 present the same spatial distribution in the basin, highlighting the northern regions of each sub-basin and particularly the north-western MS as the areas of the highest MHW intensity. The significantly smaller mean intensity values in the entire basin in Darmaraki et al. (2019a) result from their definition for MHW intensity as the mean SST anomaly over the event duration with respect to the selected temperature threshold, instead of the local climatology used in this study following Hobday et al. (2016).


MHW analyses based on satellite SST observations by Ibrahim et al. (2021) and Juza et al. (2022) for the period 1982-2020 as well as Dayan et al. (2023) for the period 1987-2019 are also consistent with the current findings for summer MHWs, despite their shorter reference periods and the season-independent analysis. However, all aforementioned studies report larger MHW durations in the south-eastern MS compared to our results. This difference is mostly related to our choice to focus on



summer MHWs that begin and decay within the JAS summer period. Taking into account all events throughout the year, similar MHW duration values are reproduced (not shown).

In the interest of clarity, means of MHW properties presented in Sect.3.2 (Fig. 6 and Table 1) are produced by averaging over all summer events within the 71 years of the study period and separately for the EMS-years. Consequently, they represent the
mean seasonal conditions of i) all summer MHWs in the study period and ii) the MHWs occurring in EMS, respectively, both not expected to be indicative of the most extreme MHW conditions. MHWs may promote the formation of an EMS either by being more frequent, of greater duration or of greater intensity than usual, thus, contributing in different ways –not necessarily bearing extreme characteristics– with positive SST anomalies to the seasonal SST being the focus of this study.

### 3.2.2 Marine heatwave properties during extreme marine summers

To understand the role of MHWs in EMSs, this section presents the mean EMS anomalies of MHW properties with respect to their mean summer values. Mean MHW conditions in EMSs present positive intensity anomalies mostly in the northern parts of each sub-basin, where the mean MHW intensity is generally larger (Fig. 6a vs Fig. 6d). Intensity anomaly values in EMSs in the north-western Mediterranean, the Adriatic, the Aegean and (part of) the Ionian Seas, reach up to $0.5°$ C. Surprisingly, in the rest of the basin, EMSs exhibit lower intensity relative to the mean summer conditions (negative anomalies in Fig. 6d). In
terms of mean basin values, mean summer MHW intensity equals $1.56°$ C while when isolating events during EMSs, it drops to $1.45°$ C (Table 1). This suggests either the suppression of MHWs in EMSs or the existence of alternative mechanisms potentially enhancing the MHW conditions in EMSs.

Supporting the latter scenario, MHWs in EMSs appear longer lasting as well as more frequent (Figs 6e and f, respectively).
The duration of MHWs increases in EMSs by 4 days in average (Table 1) while in certain locations this duration might exceed 15 days. Such large durations are mostly located in the Aegean, the Ionian and the Adriatic Seas and to a lesser extent in specific spots in the WMED Sea. Few small areas in the western basin and the south-eastern Levantine Sea present slightly smaller mean event duration in EMSs (Fig. 6e). Increased cumulative intensity values (intensity integral over the event duration) in EMSs (Table 1) reflect the impact of the large positive duration anomalies in EMSs, overcoming the impact of the
negative intensity anomalies in basin scale. In addition, frequency anomalies are positive almost in the entire basin, with most areas experiencing at least one additional MHW during EMSs (Fig. 6f). The largest frequency anomalies appear in the Aegean (approximately 3 extra events/year), followed by the Adriatic and the north-western MS.

As discussed in Sect. 3.1, EMSs in the basin are commonly formed due to the warmer summer days being warmer than usual
(Fig. 3b). MHW analysis of this part further suggests that enhanced MHW conditions commonly contribute to the observed SST anomalies in EMSs (Fig. 6d-f). This is particularly evident in the northern regions of each sub-basin where i) the highest contribution to the EMS RDA is primarily due to the higher rank days (Fig. 3b) and ii) all examined MHW parameters appear





increased (more intense, longer lasting and more frequent events in EMSs; Figs 6d-f). Even in the south-central basin where
the lower rank days contribute the most to the observed EMS RDA (Fig. 3c), MHWs appear more frequent during EMSs,
though of smaller intensity and of comparable duration to the climatological mean (Fig. 6d-f). On the other hand, in a small
area in the south-eastern Levantine (~32° N, 32° E), MHW intensity and duration in EMSs are smaller and frequency is of
comparable magnitude to the climatological value. This suggests that in this location, the higher rank days which are the most
responsible for the EMS formation (Fig. 3b), face more difficulty in producing distinct MHWs.

Results of this section show that MHWs in EMSs are more intense, more frequent and longer lasting in northern MS regions
and particularly in the Aegean and the Adriatic Seas (Fig. 6d-f). Notably, while the north-western part of the basin experiences
the most intense EMSs (Fig. 3a) and the most intense summer MHWs (Fig. 6a), the increase in MHW frequency and duration
during EMSs is smaller in this area compared to the eastern part of the basin, while the intensity increase shows comparable
(absolute) values. Given, however, the smaller variability of daily SST values during summers in the EMED, we conclude that
the relative role of MHW events in the formation of EMSs is more pronounced in the EMED.

**Table 1: Domain averaged summer Marine Heatwave (MHW) properties in 1950-2020 and in Extreme Marine Summers (EMS)
identified in 1950-2020: Mean intensity (Imean), Maximum intensity (Imax), Duration (D), Frequency (F), Cumulative Intensity
(Icum). Subscript EMS denotes the mean EMS value. Middle row (Orig) and bottom row (EMD) quantities are computed based on**
**the original and the detrended ERA5 daily summer SST data, respectively. MHW detection methodology and definition of metrics
follow Hobday et al. (2016).**

| SST dataset | Imean (°C) | Imean$_{EMS}$ (°C) | Imax (°C) | Imax$_{EMS}$ (°C) | D (days) | D$_{EMS}$ (days) | F (events/y) | F$_{EMS}$ (events/y) | Icum (°C Days) | Icum$_{EMS}$ (°C Days) |
|---|---|---|---|---|---|---|---|---|---|---|
| Orig | 1.56 | 1.35 | 1.86 | 1.73 | 12.69 | 16.65 | 0.43 | 1.50 | 19.74 | 22.46 |
| EMD | 1.41 | 1.15 | 1.66 | 1.40 | 11.88 | 14.60 | 0.46 | 1.34 | 16.69 | 16.77 |

### 3.2.3 Marine heatwaves using detrended data

When the multi-decadal signal is removed, MHWs are basin-wide slightly less intense (Table 1). The spatial distribution of
MHW intensity is similar to the original dataset, with the north-western basin (and particularly the Gulf of Lions) followed by
the Adriatic Sea experiencing the most intense events (not shown). This suggests the presence of similar event-driving
mechanisms and reveals the positive contribution of the multi-decadal signal to the observed sea surface warming, this time
specifically via the SST anomalies caused by MHWs. The detrended dataset also shows slightly shorter event durations,
resulting in a smaller cumulative intensity compared to the original dataset, while the event frequency remains relatively
unchanged (Table 1).

During EMSs, MHW intensification is greater when using the original dataset. This is more pronounced in the eastern basin
and particularly in the Aegean Sea and is related to a certain extent to the increasing SST trends during the past decades (where





summers are more likely to be identified as extreme) being larger in the eastern part of the MS (e.g., Pastor et al., 2020; Pisano

et al., 2020). This is also in line with the documented decreased frequency and intensity of the northerly summer prevailing

winds blowing over the Aegean Sea (Etesians) during the past few decades (Poupkou et al., 2011; Anagnostopoulou et al.,

2014; Tyrlis and Lelieveld 2013; Dafka et al., 2018).

Moreover, we observe a similar behavior of MHW properties in the detrended compared to the original dataset: longer lasting

events in EMSs (compared to the mean state) counterbalance the lower event intensity values in EMS (Table 1). Nevertheless,

some differences exist. The northernmost MS regions in the detrended dataset do not stand out in the EMSs as in the original

dataset (not shown). In particular, in absence of long term trends, it is the Aegean, the Tyrrhenian and certain areas in the

Ionian Sea that exhibit an enhanced role of MHWs in the EMS formation, as indicated by all three examined MHW quantities

(intensity, duration, frequency). Among these regions, the Aegean Sea stands out as the one presenting a pronounced MHW

contribution to EMSs both in the original and the detrended dataset, though to a lesser extent for the latter. Given that, the

pronounced role of MHWs in EMSs in the eastern basin can be only partly attributed to the sea surface warming trend in the

MS during the past few decades.

### 3.3 Investigation of potential extreme marine summer drivers

This section focuses on the role of air-sea heat exchanges during EMSs while additionally discusses other physical mechanisms

related to the EMS formation. Its content is structured as follows: We first present and discuss the anomalies of $Q_{net}$ and its

components in EMSs relative to their mean summer value over the study period (Section 3.3.1). This task is complemented by

examining wind and MLD in EMSs as well as the preconditioning factor introduced in Methods. Then, we examine the

meaning of using mean EMS values, considering the expected differences among the locally detected EMSs (Sect. 3.3.2). In

Sect. 3.3.3, we quantify the driving role of air-sea heat fluxes in the formation of EMSs. To achieve this, we use the proposed

metric described in Methods and Appendix A, to compute the contribution of $Q_{net}$ to the SST changes that are responsible for

making a summer extreme. An illustrative example on quantifying the role of $Q_{net}$ during a specific EMS based on this metric

is also provided (Sect. 3.3.4). Finally, the role of the multi-decadal variability in our findings for surface heat fluxes during

EMSs is shortly discussed (Sect. 3.3.5).

### 3.3.1 Air-sea heat flux anomalies in extreme marine summers

$Q_{net}$ anomalies in EMSs display a non-uniform distribution over the basin (Fig. 7a). Interestingly, positive anomaly values (i.e.,

heat gain or reduced loss from the sea surface) appear only in certain areas, most of which are in the northern flanks of the

Mediterranean basin. The eastern Aegean and a part of the central Levantine, the Adriatic Sea and the western part of the Gulf

of Lions present the greater positive anomalies reaching up to 15 W/m$^2$. Consistent with this finding, these are the areas

presenting a significant negative correlation between SST and the net heat loss from the sea surface, as shown by Shaltout and

Omstedt (2014) for the period 1982–2012. Moreover, positive $Q_{net}$ anomalies appear mainly in areas where enhanced MHW





conditions in EMSs were detected (northern Mediterranean regions and particularly the Aegean and the Adriatic Seas; Fig. 7a vs Figs 4d-f). This similarity is at least partly attributable to the widely explored driving role of air-sea heat fluxes in the formation of MHWs (e.g., Holbrook et al., 2019; Gupta et al., 2020; Schlegel et al., 2021, Vogt et al., 2022) and is further analyzed in Sect. 3.3.3.


The way each of the four $Q_{net}$ components contributes to the net heat balance in EMSs is highly variable throughout the basin (Fig. 7b-e). However, we distinct two primary mechanisms leading to positive $Q_{net}$ anomalies: the reduction of LH loss and the increase in net SWR. The former is mostly met in the Aegean and Levantine Seas and in the western part of the Gulf of Lions (Figs 7a,b) and the latter in the Adriatic Sea (Figs 7a,c). In the following, we discuss how the different $Q_{net}$ components, wind and mixed layer thickness behave in EMSs in the abovementioned areas.


The positive $Q_{net}$ anomalies observed in the eastern Aegean and in the central Levantine Seas are mainly formed by reduced LH losses, as shown by the positive LH anomalies in the area (Fig. 7a,b). Although wind in EMSs appears reduced almost in the entire basin, it is in the EMED and particularly in the Aegean Sea where the largest negative anomalies appear (Fig. 7f). In fact, positive LH flux anomalies appear in the EMED where the northerly Etesian winds persistently blow during the summer period (from the north-east (north-west) in the northern (southern) Aegean with maximum values in the central Aegean; Nittis et al., 2002). This suggests that the reduced strength (or eventual ceasing) of the Etesian winds plays a crucial role in the EMS formation in the area through the suppression of LH loss from the sea surface and the consequent rise in SST. In the eastern Aegean, the reduced northerly winds in EMSs are expected to promote the SST increase also through the suppression of the local upwelling processes. Partly supporting this hypothesis, negative MLD anomalies appear in the Aegean Sea –although not particularly pronounced along the Turkish coasts– suggesting that decreased entrainment of cold water at the base of the mixed layer possibly takes part in setting up the observed surface warming in EMSs (Fig. 7g). SH flux (Fig. 7d) and net SWR anomalies (Fig. 7c) in the eastern Aegean and the central Levantine Seas are negative but of a much smaller magnitude. Finally, the EMED and particularly the Aegean Sea present the highest positive LWR anomalies (Fig. 7e) in the basin. The net SWR deficit in the same area suggests that increased cloudiness over the Aegean and the Levantine Seas in EMSs has probably formed the observed net LWR surplus (further discussed below).




Similar to the Aegean Sea, in the western part of the Gulf of Lions positive $Q_{net}$ anomalies are largely determined by reduced LH losses (Fig. 7a,b). Here, winds during EMSs (north-westerly Mistral and Tramontane) present negative anomalies (Fig. 7f) consistently with the observed suppression of LH fluxes. Negative MLD anomalies in the western part of the Gulf of Lions also imply wind-induced mixing reduction in the vertical (Fig. 7f,g). Positive net SWR anomalies additionally contribute to the overall heat gain (Fig. 7c). This area belongs to a much greater part of the MS that presents positive net SWR anomalies in EMSs and covers more than half of the basin, extending from the north-western to the central Mediterranean and the Adriatic




Sea (Fig. 7c). The contribution of SH flux and net LWR in the overall heat gain detected in the area during EMSs appears non-

significant.



**Figure 7: Extreme marine summer (EMS) anomalies of a) $Q_{net}$, b) latent heat flux, c) net shortwave radiation, d) sensible heat flux, e) net longwave radiation, f) wind speed at 10m and g) ocean mixed layer depth (MLD); EMS anomalies are computed with respect to their mean summer state based on 1950-2020 for all parameters except for MLD where the 1987-2019 baseline period was used. Downwards fluxes have been considered as positive, so positive heat flux EMS anomaly values correspond to either heat gain or reduced loss from the sea surface.**

In contrast to the two areas previously discussed, the positive $Q_{net}$ anomalies appearing in the largest part of the Adriatic Sea are almost fully determined by the SWR (Figs 7a,c). Apart from a very small area close to the Strait of Otranto, the Adriatic Sea presents a negative contribution of the LH flux component to the observed net heat gain (Fig. 7a,b). The increased LH loss





in EMSs here is exceeded by the gain in net SWR, leading to the observed net heat gain. The positive anomalies of the net LWR and especially of the SH flux are of a much smaller magnitude in this area compared to the other components (Figs 7b-e).

Particularly during summers, winds blowing over the sea surface are expected to have a great impact on the MLD and SST
evolution, as the upper ocean is highly stratified and thus more sensitive to atmospheric forcing variability (D'Ortenzio and Prieur, 2012). Indeed, wind during EMSs appears weakened in the largest part of the basin (Fig. 7f). It is found to exceed (marginally) the climatological mean value only in small areas in the northern parts of the Adriatic, east of Sicily, off the Libyan coasts and in the northern Aegean. In these areas, consistently with the enhanced wind speed, positive MLD anomalies imply stronger vertical mixing near the surface (Fig. 7g). MLD and wind speed anomalies tend to present the same sign,
especially where large anomalies appear. For instance, west of Sardinia, off the eastern coasts of Tunisia and in the Aegean Sea where large negative wind anomalies (i.e., reduced winds) are found, also MLD presents the largest negative anomalies (i.e., reduced MLD), though in more contained areas. Wind anomalies, being negative in most areas, display a less variable spatial distribution compared to the MLD anomalies (Figs 7f,g). As thoroughly discussed in D'Ortenzio and Prieur (2012), wind affects the mixed layer of the MS at several spatio-temporal scales thus making wind-driven processes hard to describe
in a climatological context. Hence, we do not expect to reveal an indubitable cause-effect relation between wind and the mixed layer evolution by using mean seasonal wind speed and MLD anomalies. Despite these caveats, results show that wind and MLD anomalies tend to evolve consistently in most areas, suggesting that this approach is able to describe to a certain extent the wind effect in the stratification state at seasonal scale.

Understanding the role of net LWR anomalies in EMSs also bears some complexity. The skin SST-driven upwards LWR as part of the net LWR, makes it difficult to reveal the responding/driving role of the net LWR component. To overcome this, we could separately examine the downwards LWR. However, this inevitably includes the downward radiation that specifically results from the lower atmosphere response to the sea surface warming (e.g., after a MHW occurrence) through increased humidity, as discussed in Zeppetello et al., (2019). This positive feedback mechanism renders the driving role of the downward
longwave component, less clear. Nevertheless, important insight into the role of both LWR and SWR radiation fluxes components in EMSs can be obtained by examining Fig. 7c vs Fig. 7e. These fields suggest that LWR and SWR (standing for net values from now on) in EMSs work complementarily in the basin.

The Aegean and Levantine Seas present negative SWR and positive LWR anomalies, suggesting increased cloudiness in the
eastern basin during EMSs. Indeed, positive EMS anomalies of total cloud cover derived from ERA5 are found in this area (Fig. B3a) leading to increased downwards LWR and the observed reduced net LWR loss from the sea surfaec (Fig.7e). In contrast, the largest part of the central and western basin present positive SWR anomalies in EMSs. Particularly in parts of the Adriatic and the northern Ionian Seas, the SWR surplus is responsible for the net heat gain observed in EMSs (Fig. 7a vs Figs





7b-e). Clear-sky conditions seem to favor EMSs in these areas, as implied by the reduced total cloud cover found in EMSs
(Fig. B3a).

Results suggest that $Q_{net}$ anomalies (either positive or negative) in EMSs are primarily formed by LH fluxes in most areas, followed by SWR in the western and central basin and particularly in the Adriatic Sea. Importantly, negative $Q_{net}$ anomalies in EMSs are observed in large part of the basin. On interpreting this, it should be noted that seasonal $Q_{net}$ anomalies reflect the
cumulative role of surface fluxes in EMSs, i.e., that the heat flux gained by the sea surface is smaller or larger than the upward heat flux (relevant discussion in Methods, Sect. 2.2.4). Such information does not provide a safe conclusion on whether (and to what extent) $Q_{net}$ drives EMSs. To fill this gap, in Section 3.3.3 we apply the proposed methodology (see Methods) for the quantification of the $Q_{net}$ contribution in developing EMSs.

### 3.3.2 Differences among local extreme marine summers

Differences in the behavior of surface heat fluxes are expected to exist among the locally detected EMSs and the mean EMS state presented in the previous Section. To shed light on this, Fig. 8 illustrates how common is a positive anomaly for each of the examined heat flux components during EMSs. This is expressed as a fraction out of the 4 locally identified EMSs, ranging from 0 (no heat gain at all) up to 4/4 (heat gain in all EMSs). The same approach is used to examine the occurrence frequency of weakened winds (Fig. 8b), reduced MLD (Fig. 8c) and pre-conditioning (Fig. 8d) in EMSs.


The areas where positive $Q_{net}$ anomalies in EMSs are more frequently met are the same as where the largest positive mean EMS $Q_{net}$ anomalies were found (Fig. 7a vs Fig. 8a). In particular, positive $Q_{net}$ anomalies appear in at least 3 out of the 4 EMSs in the western part of the Gulf of Lions, the Adriatic and the Aegean Seas. Additionally, $Q_{net}$ anomalies appear most commonly dependent on LH flux in parts of the Aegean and Levantine Seas as well as in the western part of the gulf of Lions,
(Fig. 8a,f) and on SWR in the central and western basin (Fig. 8a,g), again in consistency with the mean EMS anomaly fields.

Wind appears weaker in the majority of the EMSs (3/4 or 4/4) with the exception of few areas (Fig. 8b). In the Aegean Sea, low wind conditions contribute to surface warming through the suppression of both LH fluxes (Fig. 8f) and vertical mixing (Fig. 8c). The former mechanism appears more commonly in the Aegean, in a few spots in the Levantine Sea and in the western
part of the Gulf of Lions than in any other location (Fig. 8f). In a large part of the central and western basin, LH fluxes rarely present positive anomalies, despite the reduced winds. In the same locations, SWR radiation appears increased in most EMSs, reinforcing the mean EMS findings.

Negative $Q_{net}$ anomalies appear in every EMS in the Ligurian, the south-central as well as in a few specific spots in the WMED
Sea, despite the systematically increased SWR. In these areas wind appears always reduced in EMSs (although not followed by decreased LH loss), while suppression of vertical mixing is observed only in half (or less) of the EMSs (Fig. 8c). On the



other hand, the preconditioning index is found to play an important role in these areas, being positive in almost all EMSs (Fig. 8d).

Figure 8: **Percentage of the locally identified extreme marine summers (EMS) presenting a) positive net heat flux anomalies b) reduced wind speed, c) reduced mixed layer depth, d) positive preconditioning index, e) positive sensible heat flux anomalies, f) positive latent heat flux anomalies, g) positive net shortwave radiation anomalies and h) positive net longwave radiation anomalies. Zero value (red) corresponds to a non-favoring role in any of the considered EMS; 4/4 (purple) corresponds to a favoring role in all EMSs.**

Among the areas of negative $Q_{net}$ anomalies, the south-central basin off the African coasts is where the coldest part of the SST distribution was found to contribute the most to the EMS SST anomalies (Fig. 3c). Relying on the increased mixed layer heat content before the beginning of almost every EMS in this area (Fig. 8d), we expect that the responsible cool summer days (being warmer than usual) are mostly the early summer ones. This is confirmed by looking into the area-averaged SST seasonal



cycle (not shown). In all EMSs in this area, the early summer (July) SST anomalies are the largest within the season. The
        increased mixed layer heat content in June is reflected in the positive SST anomalies of June while, also in May, SST in this
        area was found marginally larger than climatologically.

        The temporal coverage of the modelled temperature and MLD used in the OHC calculation inevitably limits this task in using
the original dataset, since the EMSs identified based on the detrended SST time series often fall out of their temporal coverage.
        For this reason, a complementary task included the use of the detrended dataset only for locations that experience EMSs within
        the available period, in order to examine if the observed positive contribution of preconditioning results from the surface
        warming trend in the basin (not shown). In the region discussed above (south-central basin), increased upper ocean heat content
        and spring SST values are observed before each EMS also in the detrended dataset, i.e., independently of the warming trend.


        Upper ocean heat accumulation is generally expected to precondition anomalously warm surface conditions (Marin et al.,
        2022). Indeed, in the greatest part of the basin, the preconditioning factor presents a positive contribution in more than half of
        the EMSs (Fig. 8d). Usefulness of this index is highlighted when examining single summers (e.g., the exemplary case of EMS
        2015 in Sect. 3.3.4). Even if it is positive in most cases, its contribution is often enhanced when and where there is no causal
link between the surface net heat balance and the observed surface warming thus revealing it constitutes an actually
        contributing EMS formation factor. Notably, occurrence frequency of preconditioning seems to differentiate in the Alboran
        Sea and the area surrounding the Strait of Sicily (Fig. 9d). Although horizontal advection is not investigated in this study, this
        finding potentially suggests that EMS preconditioning hardly takes place in areas of intensified circulation.

### 3.3.3 Quantifying the driving role of air-sea heat fluxes in the formation of extreme marine summers

In this section, we quantify the driving role of surface heat fluxes in the formation of EMSs. To this aim, we use the proposed
        metric ($P_{EMS}$) described in Sect. 2.2.4 and Appendix A, to compute the contribution percentage of $Q_{net}$ to the SST changes that
        are particularly responsible for making a summer extreme. $P_{EMS}$ metric values in the MS basin are presented in Fig. 9a. Results
        reveal a crucial role of $Q_{net}$ in driving EMSs in the northern half of the basin where the highest contribution percentages are
        encountered. A latitudinal gradient of $P_{EMS}$ is generally observed in the basin: negative contribution percentages are found
along the African coasts while positive percentages of increasing value are found while moving towards northern MS areas.
        In positive $P_{EMS}$ areas with $0 < P_{EMS} < 100\%$, other mechanisms are expected to work complementarily (i.e., towards greater
        SST anomalies). Wind-induced mixed layer shoaling likely contribute in such cases, as suggested by negative wind speed and
        MLD anomalies (e.g., in Balearic and Aegean Seas; Figs 7f,g). $P_{EMS}$ commonly exceeds 100% mainly in northern
        Mediterranean regions (e.g., in Ligurian, Adriatic, North Aegean Seas), meaning that additional processes (e.g., currents,
vertical mixing) are expected to counteract the extra heating caused by surface fluxes in these cases. Negative $P_{EMS}$ values in
        the southern regions reveal that air-sea heat exchanges work against the SST anomalies that are mostly responsible for the
        EMS formation; hence oceanic processes definitely drive EMSs in these areas.



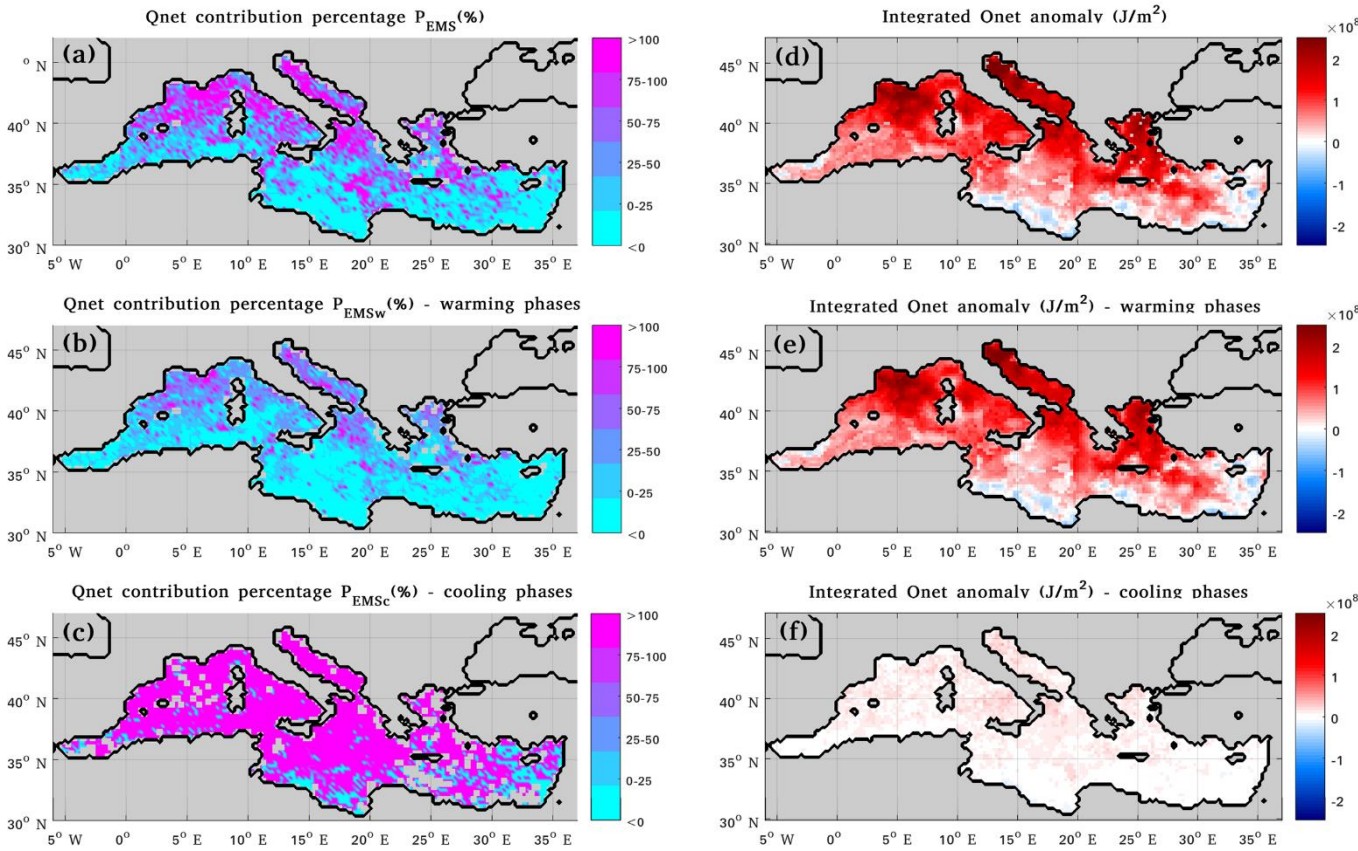

**Figure 9: Left: a) $P_{EMS}$ metric values (%) for the contribution of $Q_{net}$ to the formation of EMSs (based on periods of positive cumulative SST anomalies and positive change in SST anomaly), b) same as (a) but further focusing on warming phases ($P_{EMS_w}$), c) same as (a) but further focusing on cooling phases ($P_{EMS_c}$); Right: d) Integrated $Q_{net}$ anomalies during the periods selected for the $P_{EMS}$ computation, e) same as (d) but further focusing on warming phases and f) same as (d) but further focusing cooling phases. Note that $P_{EMS_w}$ and $P_{EMS_c}$ percentages in (b) and (c) are relative to different reference periods (the selected warming and cooling phases, respectively; thus, their sum is, by construction, not expected to be equal to $P_{EMS}$ in (a). Downwards fluxes have been considered as positive, so positive anomaly values in d-f correspond to either heat gain or reduced loss from the sea surface during the selected phases. Sum of (e) and (f) equals the sum of $Q_{net}$ anomalies in all selected phases (d).**

As described in Methods (Sect. 2.2.4), this metric quantifies the role of surface fluxes only during specific EMS sub-periods.
These are the periods when SST is kept above climatology via either a) faster warming or b) slower cooling compared to the corresponding climatological period. Examining $Q_{net}$ anomalies during these selected phases provides useful insight. Cumulative $Q_{net}$ anomalies during warming phases are almost equal to the $Q_{net}$ anomalies of all (warming and cooling) examined phases (Figs 9d-f). Moreover, SST anomalies during warming phases cover more than 90% of the SST anomalies during all examined phases (not shown). The above suggest that warming phases happening at a higher rate than usual are





primarily responsible for the EMSs. As expected, $P_{EMS_w}$ (see Appendix A) displays a very similar spatial distribution and slightly lower values compared to $P_{EMS}$ (Figs 9a,b). Interestingly, metric values for the cooling phases ($P_{EMS_c}$) show that cooling at a lower rate than usual is a process totally driven by surface fluxes ($P_{EMS_c}$ >100%) almost in the entire basin (Fig. 9c). However, given that these phases correspond to a very small percentage of the observed SST anomalies, this heat flux-driven cooling mechanism is not as important for the formation of an EMS as warming towards higher SST anomalies.


Negative $P_{EMS}$ values specifically observed in the southern Mediterranean regions along the African coasts (Fig. 9a) are of particular interest. These values indicate that oceanic processes are primarily responsible for the observed EMS SSTs in these areas. Even during the selected warming phases (Fig. 9e), negative $Q_{net}$ anomalies are observed, especially off the Libyan coasts and in the south-eastern Levantine Sea. Hence high SST anomalies leading to EMSs in these areas are formed despite

the thermal energy deficit at the sea surface during the same periods. As noted in Sect. 3.3.2, increased LH losses in all EMSs mainly form the negative $Q_{net}$ balance in these areas (Figs 8a,f). The present section complements this finding by revealing that a thermal energy deficit at the sea surface is consistently observed in these areas even when focusing solely on highly effective (for the EMS formation) warming phases. This suggests that this negative $Q_{net}$ balance actually occurs during the EMS development rather than being an effect of averaging over the season. Considering that wind speed appears generally

reduced during EMSs, to understand what systematically increases LH loss in the southern part of the basin, we additionally examine the ERA5 specific humidity anomalies in EMSs. Results indicate the presence of drier air masses over these areas during EMSs, compared to the northern MS regions (Fig. B3b).

In MS areas characterized by negative values of $P_{EMS}$, weakened winds are most commonly encountered, occasionally

accompanied by reduced mixed layer depth (Fig. 9a, Figs 8b,c). In such cases, surface warming may be partly attributed to enhanced stratification under low wind conditions. Otherwise, e.g., in locations where despite the weaker winds, MLD appears increased, other oceanic mechanisms as the horizontal advection (not examined in this work) are apparently responsible for the high EMS SSTs. In fact, the spatial distribution of negative $P_{EMS}$ values (Fig. 9a) suggests a potential association with the eastward circulation encountered along the African coasts, especially to the east of the Sicily Strait. This hypothesis suggests

that warmer surface currents moving eastwards may act in favor of the development of EMSs in the southern Mediterranean regions. Moreover, results suggest that in the negative $P_{EMS}$ areas initial conditions commonly play an actual role in the local EMS formation (Fig. 8d). In fact, in the southern-central basin the early summer days in EMSs were the ones found to be warmer than climatologically.

Results also reveal a strong link between MHW properties and surface heat fluxes (Fig. 9a vs Fig. 6d). MHWs in EMSs have been found to present greater intensity, duration and frequency anomalies relative to mean MHW conditions in northern MS regions. In contrast, in southern MS regions they contribute to EMSs by occurring more frequently and lasting longer, while




their intensity is lower than usual (Fig 6d-e). The spatial similarity of MHW intensity and $P_{EMS}$ over the basin suggests that the crucial role of surface heat fluxes found for the northern MS is associated with their ability to drive high SST anomalies (thus intense MHWs). Consistently, heat fluxes were found to work against the EMS formation in the southern MS where MHWs during EMSs are less intense. This further supports that $Q_{net}$ modulates particularly the intensity of MHWs (Fig 6d-e).

### 3.3.4 Illustrative example of extreme marine summer 2015

To further illustrate the usefulness of metric $P_{EMS}$, we analyze an example related to the EMS 2015. Summer 2015 has been the most widely experienced EMS in the basin (non-grey locations in Fig. 8). This is also the summer when one of the most extreme MHWs in the MS took place, covering 89% of the basin and lasting 63 days within July-September 2015 (Darmaraki et al., 2019).

$Q_{net}$ appears to explain the EMS occurrence in most of the central and eastern basin (named as positive part from now on) (Fig. 10a). The Adriatic, Ionian, southern Aegean and central Levantine Seas present the highest $P_{EMS}$ values, often exceeding 100%. Few smaller areas, mostly scattered in the southern positive part, present negative $P_{EMS}$ values. In such locations, other local processes are expected to surpass the "cooling" impact of the surface net heat balance and further drive the EMS SST anomalies.

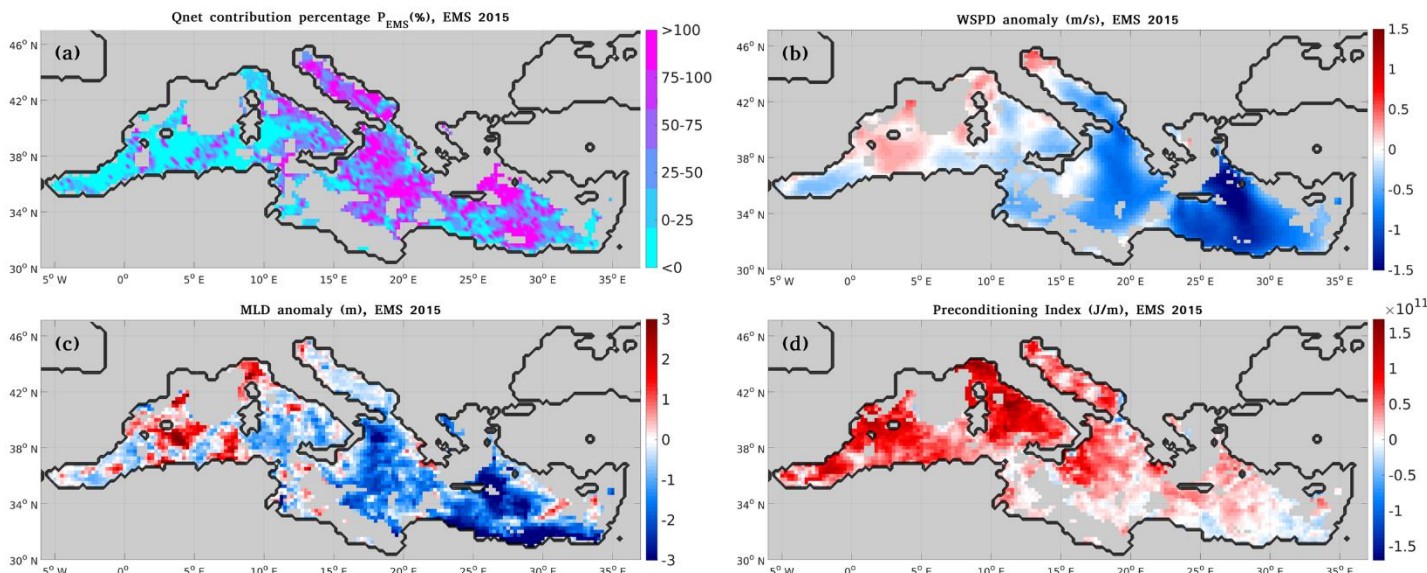

**Figure 10: Left: a) $P_{EMS}$ metric values (%) for the contribution of $Q_{net}$ to the formation of the Extreme Marine Summer (EMS) 2015 (based on periods of positive cumulative SST anomalies and positive change in SST anomaly), b-d) EMS anomalies of b) wind speed at 10m (m/s), c) mixed layer depth (m) and d) preconditioning index (J/m) for June 2015. EMS anomalies are computed with respect to their mean summer state based on 1950-2020 for wind speed and 1987-2019 for the mixed layer depth and the preconditioning Index. Non-coloured sea grid points in the Mediterranean Sea stand for locations that did not experience summer 2015 as extreme.**





Wind speed during this summer appears lower than usual in the entire positive part and particularly lower where $P_{EMS}$ appears
larger (Fig. 10a,b). Wind-forced LH flux anomalies are responsible here for the strong $Q_{net}$ contribution, similarly to the mean
EMS state in most Mediterranean areas. Consistently with the reduced winds, a decrease in MLD appears to further contribute
to the EMS SST anomalies (Fig. 10b,c). In areas where $Q_{net}$ corresponds to greater warming than the observed ($P_{EMS} > 100\%$),
such as in the southeastern Aegean and the central Ionian (where reduced vertical mixing additionally favors high SSTs),
surface currents are highly expected to dump the residual warming.


In contrast, $Q_{net}$ does not contribute to the EMS formation in several WMED areas (negative part) (Fig. 10a). This is related to
the winds during this summer being stronger than usual in most of this area (Fig. 10b). Stronger winds in the negative part lead
to i) increased LH losses controlling the net heat flux signal (as typically occurs in EMSs) and ii) increased vertical mixing as
implied by the MLD anomalies being mostly positive in the negative part (Fig. 10b,c). On the other hand, the preconditioning
index is clearly more pronounced here than in the positive part (Fig. 10d). Indeed, early summer days (July 2015) present the
largest SST anomalies within the season in this part of the basin. Further looking into the SST evolution throughout the year
reveals positive SST anomalies (relevant to climatology) also during the precedent spring months (not shown). Therefore, the
initial thermal state of the upper ocean in this part of the basin has worked in favor of the formation of this EMS.

Following 2015, other widely experienced EMSs such as 2003, 2012, 2018 (in descending order by means of spatial extent)
are examined (not shown). Results show that in case surface fluxes and stratification conditions do not contribute to surface
warming, the preconditioning index presents an enhanced positive signal. Although not quantitative, the approach of using this
index serves to diagnose if there is a contributing role of this ocean memory-related factor in the formation of a summer as
extreme. At the same time, it highlights the importance of considering in such studies the longer, compared to the atmosphere,
ocean time scales.

### 3.3.5 Air-sea heat fluxes using detrended data

To shed light on how the multi-decadal variability affects the role of surface heat fluxes in EMSs, we inter-compare the EMS
$Q_{net}$ anomalies derived from the original and from the detrended dataset. When the multi-decadal trend is removed, the EMS
$Q_{net}$ anomalies appear to be modulated by LH fluxes in the entire basin (Fig. B1). This constitutes the major difference
compared to the original dataset where in certain locations in the western and central basin, and particularly in the Adriatic
Sea, clear sky conditions were found to play a major role during EMSs (Sect. 3.3.1).

In particular, positive SWR anomalies in EMSs in the western and central basin are much smaller in the detrended dataset and
appear only in certain areas (Fig. B1). This is related to the increasing trend of the summer net SWR since the mid-1970s being
larger over the western and central basin (Fig. B2). As a consequence, SWR anomalies in these areas are larger in the original



dataset where EMSs are identified mostly within the latest years. Results suggest that the key role of SWR in the EMSs actually experienced in western and central Mediterranean areas, originates from the SWR long-term trend.

Wind speed EMS anomalies in the detrended dataset present the same sign and spatial distribution as in the original dataset (not shown), further supporting the promoting role of low wind conditions for an EMS to occur, regardless of long-term variability. Nevertheless, wind during EMSs is less weakened in the detrended dataset. This magnitude difference is more pronounced over the Aegean and central Levantine Seas, where also LH fluxes in EMSs are less suppressed in the detrended dataset (not shown). The above is consistent with the documented weakening of the Etesians (Tyrlis and Lelieveld, 2013; Poupkou et al., 2011; Dafka et al., 2018). In agreement with the literature, significant decreasing trend of wind speed in the

EMED was also found based on the ERA5 JAS winds over the recent decades (not shown).

Summarizing, results suggest that in absence of multi-decadal variability, $Q_{net}$ in EMSs is basin-wide dependent on the LH component. On top of this dependency, the actually observed $Q_{net}$ during EMSs is further determined by the multi-decadal variability of i) the SWR in the western and central basin and ii) the wind-induced LH fluxes in the eastern basin.

**4. Summary and conclusions**

The Mediterranean Sea surface has undergone significant warming over the past decades, surpassing the warming observed in the global ocean and exhibiting even higher temperature trends during the summer season. The present study proposes the concept of Extreme Marine Summers (EMSs) and investigates their characteristics in the MS in a climatological framework, using daily SST data from the ERA5 Reanalysis for the period 1950-2020. It explores: i) the SST substructures during EMSs,

ii) the role of MHWs during EMSs iii) physical mechanisms related to the EMS formation focusing on the role of air-sea heat fluxes and iv) the impact of the SST multi-decadal variability in the studied EMS features.

EMSs identified in the MS over the study period display the largest mean seasonal SST anomalies in the western part of the basin, reaching up to 2.5° C with respect to the local climatological mean summer values. In most of the basin, they are

commonly formed due to the warmer summer days being anomalously warm. SST values in EMSs are organized closely following the climatological ranked daily SST variability. In particular, locations where the warmest (coldest) part of the ranked daily SST distribution is more variable climatologically, experience EMSs primarily due to the contribution of the warmest (coldest) part of the distribution.

Summer MHW events detected within 1950-2020 are more intense in northern MS regions, with the north-western basin exhibiting the highest MHW intensity, i.e., where the daily summer SSTs are more variable. MHWs in EMSs present greater intensity, duration and frequency anomalies relative to mean MHW conditions in the northern parts of each sub-basin.





Although the north-western part of the basin experiences the most intense EMSs and summer MHWs, the relative role of MHWs in the formation of EMSs is more pronounced in the central and eastern basin and particularly in the Aegean and Adriatic Seas. In the rest of the basin (mainly in southern Mediterranean regions), MHWs contribute to EMSs by occurring more frequently and lasting longer, despite their intensity being lower than usual.

Wind in EMSs appears weakened in the largest part of the basin and particularly in the Aegean and Levantine Seas. These areas present the highest decreasing trends in wind magnitude over the past decades, in line with the observed suppression of LH losses. Reduced MLD in a great part of the basin during EMSs also suggests that highly stratified surface waters associated with concurrent low wind conditions are favoring conditions for the development of EMSs.

Surface heat flux anomalies in EMSs present strong spatial variability over the basin. They are primarily driven by LH fluxes in most areas, followed by SWR in the western and central basin and particularly in the Adriatic Sea. Positive $Q_{net}$ anomalies are mainly formed by decreased LH loss from the sea surface and increased net SWR. The former is mostly met in the Aegean and Levantine Seas as well as in the western part of the Gulf of Lions and the latter in the Adriatic Sea and is associated with reduced cloudiness during EMSs. Importantly, negative $Q_{net}$ anomalies in EMSs are observed in large part of the basin. This indicates that the elevated SST anomalies during EMSs in these areas are formed despite the non-favoring air-sea heat exchange, thus suggesting a key role of oceanic processes.

To quantify the driving role of $Q_{net}$ in the EMS formation, a metric has been proposed based on the surface heat budget equation. This metric –expressed as a contribution percentage– focuses on selected SST changes, considered to be the most responsible for making a summer extreme. During these summer sub-periods, SST is kept above climatology through either a) faster warming or b) slower cooling compared to the corresponding climatological period.

Results show that EMSs are driven to a great extent by air-sea heat fluxes in the northern half of the MS. A latitudinal gradient is generally observed, with negative $Q_{net}$ contribution percentages in the southern MS along the African coasts that progressively become positive while moving towards the North. Mixed layer shoaling due to decreased winds is a commonly observed complementary mechanism in areas where the examined SST anomalies are not entirely explained by surface heat fluxes. $Q_{net}$ contribution percentages exceeding 100% are also commonly met in the northern half of the basin (e.g., in Ligurian, Adriatic, north Aegean Seas), meaning that additional processes (e.g., surface currents, vertical mixing) are expected to cancel the extra heating caused by surface fluxes.

Splitting the examined sub-periods in warming/cooling phases revealed that warming happening at a higher rate than usual is the main mechanism driving EMSs. Cooling at a lower rate than usual was found to be a process entirely explained by surface heat fluxes almost in the entire basin. However, such sub-periods correspond to a very small percentage (lower than 10%) of





the observed SST anomalies in EMSs. Therefore, this flux-driven cooling mechanism is not as important for the formation of an EMS as warming towards higher SST anomalies.

In the southernmost basin where air-sea heat exchanges were found to oppose the development of EMSs, oceanic processes, such as horizontal advection, are expected to be responsible for the high EMS SSTs. The thermal deficit observed at the sea surface during EMSs in these areas is driven by enhanced LH loss (despite the diminished winds), associated with drier air masses in EMSs over the southern –compared to the northern– MS regions.

The role of initial thermal conditions –potentially favoring the development of an EMS– is additionally examined by means of a proposed preconditioning index. EMS preconditioning was diagnosed as a commonly contributing EMS formation factor. Decreased occurrence frequency of EMS preconditioning in the Alboran Sea and the Sicily Strait suggests that it hardly develops in areas of intensified circulation. However, further investigation following these qualitative conclusions and assumptions is necessary towards a complete assessment of physical mechanisms related to EMSs in these areas.


Moreover, results suggest a link between MHW properties and surface heat fluxes in EMSs. The crucial role of $Q_{net}$ found for the northernmost basin during EMSs is associated with its ability to drive MHWs of higher intensity. Consistently, heat fluxes were found to oppose the EMS formation in the southern MS where MHWs in EMSs are less intense, further supporting that $Q_{net}$ modulates particularly the intensity of MHWs.


Finally, to gain insight into the impact of the SST multi-decadal variability in the studied EMS characteristics, a summer SST dataset free of multi-decadal trend was additionally produced. Removing the multi-decadal signal allowed us to explore EMS features beyond the long-term Mediterranean SST internal oscillation and climate change effect, in addition to the investigation of the summers actually experienced as extreme. To this end, the actual sea surface state –here represented by the ERA5

Reanalysis– was inter-compared with the detrended one. Results suggest the presence of a background SST anomaly field closely following the ranked daily SST variability pattern, on top of which the multi-decadal signal poses additional warming in the basin during EMSs. The dominant substructure revealed in the largest part of the basin (i.e., the warmer part of the SST distribution being responsible for the greatest part of the observed SST anomalies in EMSs) appears relatively independent of the multi-decadal trend. Nevertheless, the multi-decadal signal contributes to the observed surface warming through more

intense and slightly longer lasting MHWs, without affecting the MHW intensity spatial distribution, thus most probably also the event-driving mechanisms. Importantly, in absence of multi-decadal variability, the contribution of the net surface heat budget to the EMS formation is found to be modulated by the LH component in the entire basin. Results suggest that, in addition to the fundamental role of LH flux, the observed net surface heat budget during EMSs is further determined by the long-term variability of i) the net SWR in the western and central basin and ii) the wind-induced LH fluxes in the eastern basin.





Building upon this study, investigating SST substructures in the MS using model ensemble data for present and future climate would strengthen the statistical confidence on the current results and point out differences among the observed and projected EMS conditions, respectively. Additionally, a broader assessment of physical mechanisms potentially contributing to the formation of local EMSs (e.g., horizontal advection, vertical mixing, water/heat transport through the Gibraltar and Sicily

Straits) would complement the current findings by providing insight into the relative role of oceanic factors. Finally, in the framework of building prediction tools, the potential use of an ocean heat content index, as an indicator for future anomalously warm seasons, could be a promising direction to explore.

**Appendix A: Description of proposed metric**

The proposed metric presented here is used to quantify the driving role of air-sea heat fluxes in the formation of EMSs. The

metric focuses on summer sub-periods considered to be the most responsible for making a summer extreme. These are the periods where SST is kept above climatology through either a) faster warming or b) slower cooling compared to the corresponding climatological period. The metric is constructed according to the following steps:

Step 1: For each grid point, we split the 92 JAS summer days of each locally detected EMS in sequential warming and cooling

phases.

Step 2: We isolate phases of positive SST anomaly:

$$SSTA_{cum} > 0, \qquad SSTA_{cum} = \sum_{t_0}^{t_1}(SST_t - SST_{tclim})$$

where $SST_t$ and $SST_{t_{clim}}$ the daily SST of day $t$ and the corresponding climatological day $t_{clim}$, respectively. Indices $t_0$ and $t_1$ stand for the start and end day of each phase, respectively. This step isolates periods that contribute the most to the mean

EMS SST.

Step 3: Among these phases, we further detect the ones where SST evolves towards greater SST anomalies:

$$\Delta SST'_{obs} > 0,$$
$$\Delta SST'_{obs} = SST'_{t_1} - SST'_{t_0} = (SST_{t_1} - SST_{t_1 clim}) - (SST_{t_0} - SST_{t_0 clim})$$

where $SST'_{t_0}$ and $SST'_{t_1}$ are the daily SST anomalies at the start and the end day of each phase, respectively. During phases of

$\Delta SST'_{obs} > 0$, SST either increases at a greater rate than usual (i.e., than climatologically) or decreases at a slower rate than usual.

To explain this criterion, let us suppose that a cooling phase evolves faster than climatologically ($\Delta SST'_{obs} < 0$). The criterion of step 2 ($SSTA_{cum} > 0$) is potentially satisfied (depending on the initial conditions) but this phase will not be included in the

metric since SST changes towards lower anomalies. Even if $Q_{net}$ entirely explains the observed cooling, such a positive



contribution to the observed SST change does not constitute a positive contribution to making this summer extreme (thus considered irrelevant to our purpose). In contrast, we are interested in examining the role of surface fluxes during phases when cooling occurs at a lower rate than usual, thus contributing to the development of an EMS through maintaining SST values above climatology. Similarly, we take into account phases during which warming occurs at a greater rate than usual, while a decelerated warming (with respect to climatology) falls out of our interest.

Step 4: For the selected phases (i.e., phases of both $SSTA_{cum} > 0$ and $\Delta SST'_{obs} > 0$), we apply the surface heat budget equation using SST and $Q_{net}$ anomalies:

$$SST'_{t_1} - SST'_{t_0} = \int_{t_0}^{t_1} \frac{Q'_{net}}{\rho_o c_p h} dt + R \qquad (A1)$$

The left-hand side of Eq. A1 represents the change in SST anomaly relative to climatology, between days $t_0$ and $t_1$ (i.e., the term $\Delta SST'_{obs}$ computed at step 3). In the right-hand side of Eq. A1, $Q'_{net}$ is the daily anomaly of $Q_{net}$ relative to the mean climatological $Q_{net}$ for this phase, i.e., $Q'_{net} = Q_t - \overline{Q}_{tclim}$, where $\overline{Q}_{tclim}$ is the mean $Q_{net}$ value within the climatological days $t_{0clim}$ and $t_{1clim}$ as implemented in Fewings and Brown (2019). The time integral of $Q'_{net}$ divided by the product of the constant values $\rho$ (seawater density), $c_p$ (specific heat capacity) and $h$ (mixed layer thickness) represents the part of $\Delta SST'_{obs}$ that is attributed to $Q'_{net}$ during this phase (denoted by $\Delta SST_{Qnet}$ from now on).

Accordingly, the second term of the right-hand side in Eq. A1 stands for a change in SST anomaly due to non-heat flux (i.e., oceanic) factors affecting SST. These include: vertical mixing processes between the surface mixed layer and the rest of the water column, horizontal advection of SST, horizontal eddy heat fluxes, radiative heat loss due to SWR penetration below the mixed layer depth.

Step 5: We compute the percentage of $Q_{net}$ contribution for each phase N:

$$P(N) = \frac{\Delta SST_{Qnet}(N)}{\Delta SST'_{obs}(N)} \cdot 100 \%.$$

Step 6: To obtain a mean percentage for the entire summer, we weight the contribution percentage of each phase according to the phase's $SSTA_{cum}$. For the (NS) selected phases, the weighting coefficient of each (N) phase may be written as:

$$W(N) = \frac{SSTA_{cum}(N)}{SSTA_{cum_{ALL}}} = \frac{\sum_{t_0}^{t_1}(SST_t(N) - SST_{tclim})}{\sum_{N=1}^{NS} \sum_{t_0}^{t_1}(SST_t(N) - SST_{tclim})}, \sum_{N=1}^{NS} W(N) = 1$$

The metric value ($P_i$) for the $Q_{net}$ contribution to each EMS (i) is then:

$$P_i = \sum_{N=1}^{NS} P(N) \cdot W(N)$$

While the final metric for the mean EMS state, being the mean (among the 4 EMSs) of the above percentage, is:





$$P_{EMS} = \frac{1}{4}\sum_{i=1}^{4} P_i$$

This weighting approach promotes phases of higher $SSTA_{cum}$ considering their higher impact on the mean EMS SST. Nevertheless, sensitivity tests using different scaling (e.g., based on phase duration) present very similar results, suggesting a negligible sensitivity of the metric to these choices for averaging.

Step 7: As an additional task, we repeat steps 3 to 5 separately for the warming and cooling phases (subscripts w and c, respectively). Then, analogously to step 6, the weighting coefficient of each (N) phase for the selected warming ($NS_W$) and cooling ($NS_C$) phases, is:

$$W_{w|c}(N) = \frac{SSTA_{cum,w|c}(N)}{SSTA_{cum,w|c}_{ALL}} = \frac{\sum_{t_0}^{t_1}(SST_t(N)-SST_{tclim})}{\sum_{N=1}^{NS_{W|C}}\sum_{t_0}^{t_1}(SST_t(N)-SST_{tclim})}, \quad \sum_{N=1}^{NS_{W|C}} W_{w|c}(N) = 1$$

The metric value for the $Q_{net}$ contribution in the selected warming/cooling phases, for each EMS (i), is:

$$P_{i_{w|c}} = \sum_{N=1}^{NS_{W|C}} P(N) \cdot W_{w|c}(N)$$

While the final metric, being the mean (among the 4 EMSs) of the above percentage for the selected warming/cooling phases, is:

$$P_{EMS_{w|c}} = \frac{1}{4}\sum_{i=1}^{4} P_{i_{w|c}}$$



## Appendix B: Supplementary figures



**Figure B1: Left: Extreme marine summer (EMS) anomalies of surface fluxes and wind speed at 10m with respect to the mean summer state (1950-2020) a) Q$_{net}$, b) sensible heat flux, c) latent heat flux, d) net longwave radiation, e) net shortwave radiation and f) wind speed; Right: Same as left but using detrended data. Downwards fluxes have been considered as positive, so positive EMS anomaly values correspond to either heat gain or reduced loss from the sea surface.**



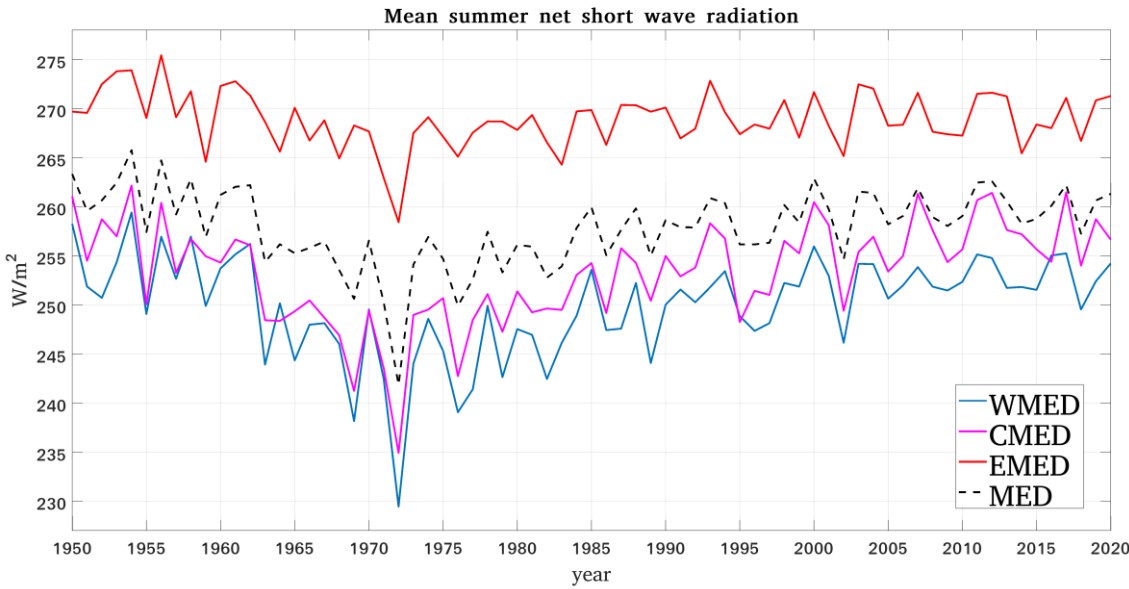


**Figure B2: ERA5 mean Summer net short wave radiation (W/m² ) in the Mediterranean Sea (1950-2020) for the entire basin (dashed black line) and for the western (extending from Gibraltar Strait to the Strait of Sicily; blue line), central (Strait of Sicily up to 22° E; purple line) and eastern (22° E eastwards; red line) sub-basins.**

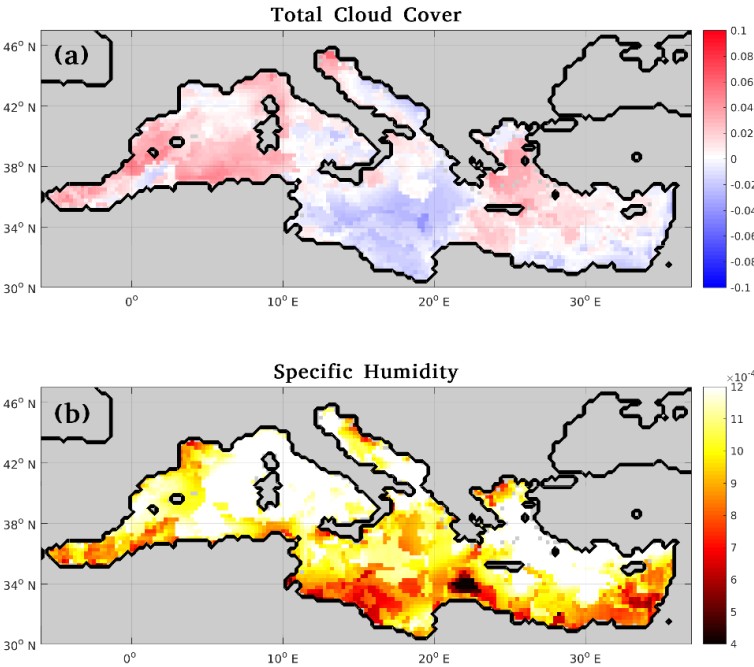

**Figure B3: Extreme marine summer anomalies (relative to the mean summer state based on the period 1950-2020) for the a) ERA5 Total Cloud Cover and b) ERA5 specific humidity (kg/kg).**



**Data availability**

All data used in this study is open access. The ERA5 reanalysis data can be obtained from the Copernicus Climate Data Store
web page. The period 1950-1978 is included in the preliminary back extension product "ERA5 hourly data on single levels
from 1950 to 1978 (preliminary version)" available here: https://cds.climate.copernicus.eu/cdsapp#!/dataset/reanalysis-era5-
single-levels-preliminary-back-extension?tab=overview (Bell et al., 2020), and the period 1979-2020 is available in the
updated      product      "ERA5      hourly      data      on      single      levels      from      1940      to      present",      here:
https://cds.climate.copernicus.eu/cdsapp#!/dataset/reanalysis-era5-single-levels?tab=overview (Hersbach et al., 2023). The
High Resolution L4 Sea Surface Temperature Reprocessed product (EU Copernicus Marine Service Product, 2022c) and the
Mediterranean Sea Physics Reanalysis product (EU Copernicus Marine Service Product, 2022d) are available through the
Copernicus Marine Service portal.

**Author contribution**

DD defined the research problem. All authors contributed to the methodology. DD conducted the analysis and wrote the
manuscript, with contributions from GK, EF, and MH. All authors contributed to the interpretation of results.

**Competing interests**

The contact author has declared that none of the authors has any competing interests.

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
