# Peer review of "Investigating extreme marine summers in the Mediterranean Sea"

_EGUsphere, 2023_

## Referee Comment (RC1)

**Review of egusphere-2023-1709: "Investigating extreme marine summers in the Mediterranean Sea".**

General review

In this manuscript, the authors introduce the new (to my knowledge) and interesting concept of *extreme marine summers*. The manuscript presents a huge amount of work from the analysis of EMS to the different drivers/mechanisms involved in such EMS with interesting results.

But, in my opinion, there is a fundamental problem regarding the definition of EMS. When reading the manuscript, I could not get a clear idea of how EMS are defined. First, the authors define EMS as *"define EMSs, separately at each grid point, as the four summers with the highest average JAS SST, i.e., exceeding the 95th percentile of the 71 available summer periods from 1950 to 2020"*. I would expect a basin definition for EMS, maybe coming after the grid point analysis. As we are talking about a season, I would expect that the categorization as extreme would affect the whole region (not with the same intensity in all grid points but with a big enough extent). Later in the manuscript the authors analyse different variables for the whole basin, calculate mean summer values,… maybe suggesting a basin-wide concept. But later there is a section named *"Differences among local extreme marine summers"*.

Another question that is not clarified is the choice of the 95% percentile. Why not 90% as for extreme air and sea temperature to define extreme events? Why 4 summers? Are there only 4 summers exceeding the percentile? Is this needed for any single grid-point or as a basin mean? Are the 4 summers the same for the whole basin?

Figure 4 shows results for three grid points where EMS occur in different years. I don't feel comfortable with a year being extreme in WMED but not in CMED. Or having a pixel suffering an EMS but not the closest one (figure 10 grey areas).

There is, of course, the possibility I did not properly understand the proposed definition. In any case, better explanation of the actual EMS definition is needed.

I also have some major concerns about the analysis of MHWs. Please, see comments below.

The authors have done extensive, very solid and interesting work on the analysis of EMS drivers. A review of the above will add value to all this work and may lead to a really interesting publication. So, I encourage the authors to send a revised version of the manuscript and please understand these criticisms as a way to improve an already good work.

From the above comments, my recommendation is that the manuscript needs major revision before it is accepted for publication.

**Main comments**

Consider introducing the Martínez et al (2023) paper regarding the impact of detrending SST on MHWs in the Mediterranean: *Martínez, J., Elisa Leonelli, F., García-Ladona, E., Garrabou, J., Kersting, D. K., Bensoussan, N., & Pisano, A. (2023). Evolution of marine heatwaves in warming seas: the Mediterranean Sea case study. Frontiers in Marine Science. https://doi.org/10.3389/fmars.2023.1193164*

2.2 Methods

2.2.1 Extreme marine summer definition and associated SST substructures

In lines 142-144 you *define "as the four summers with the highest average JAS SST, i.e., exceeding the 95th percentile of the 71 available summer periods from 1950 to 2020"*. Why four? Are they the only ones exceeding the 95th percentile? Why not another percentile? Have you checked if this percentile excludes some major MHW events? Please, explain clearly the process to select/exclude EMS/non EMS years. Do you choose the percentile first and then there are only four? Do you choose 4 years and look for a common

percentile? I think I understand the methodology, and mostly agree, but please explain better to avoid confusion.

EMS definition needs to be more robustly justified. What will happen when 2022 and 2023 are included in the study? 95th percentile will rise and maybe (probably) some of the 4 years will not meet this condition. Which impact can this have in the definition or the subsequent analysis. Please, if the data for 2022 are available, check this question.

**2.2.2 Detrending time series**

In this section Figure 2a is mentioned. In this figure time series for sub-basins are shown and sub-basins are defined in the figure caption. It would be better to explain the sub-basins, if they are used throughout the paper for analysis, in the methodology section and it would be good to add (up to you) a figure showing them or some indication in one of the actual figures. You should also explain how you decided the limits of the basins (22º, Sicily strait,…) Is the Thyrrenian sea included in the WMED or CMED? Please explain.

**2.2.3 Marine heatwave identification**

For the detection of MHW events you use the fixed climatology 1983-2012, which selection is adequately explained. In a recent work Rosselló et al suggest the use of *"a moving baseline covering the 20 years prior to the year under study"*. As you try to remove the effect of multidecadal trend, the use of a moving climatology it could also be useful. This is just a suggestion you can try or at least keep in mind for future work.

Rosselló, P., Pascual, A., & Combes, V. (2023). Assessing marine heat waves in the Mediterranean Sea: a comparison of fixed and moving baseline methods. Frontiers in Marine Science, 10. https://doi.org/10.3389/fmars.2023.1168368

Lines 195-196: *"Finally, to focus on summer MHWs, we isolated events with their onset and end day falling within the JAS summer period."* Have you checked how many events are removed by using this condition? How many MHW days are removed? In the case an event started on June 28 and lasted for 30 days, would it be considered?

Do you apply any spatial extent filter to detect MHWs? Just to avoid single pixel heat spikes.

**3.1.1 Decomposing the extreme marine summers**

Lines 242-243 "*the largest contribution to the mean EMS RDA at this location comes from the warmest part of the SST distribution (higher rank days), with the exception of EMS 1999 where the middle part contributes the most.*" Has this been calculated? Is there a numerical value? I can understand from looking at figure 4a but, what exactly is "the middle part"? Do you divide the rank daily anomaly in 3 periods of 30 days? Do you divide the RDA by some anomaly threshold?
Line 244: Why *"the 60 warmest rank days"*?

I suppose you are following the methods of Roethlisberger (2020) but a clearer explanation would be good.

In this section you comment figure 3 with contributions of the warmest/coldest half to the EMS but in the analysis of the three locations you refer to "the middle part". It would be better to use the same process for EMS analysis/characterization. Or explain what you mean as "the middle part"? Is it a percentile interval of the rank days? Is there an anomaly threshold to define the RDA parts? You also mention "the 60 warmest rank days" for 2003. Please clearly explain the methodology/process to analyse rank days series.

**3.1.3 SST substructures 300 using detrended data**

If I understood well, there are no relevant differences in the analysis from detrended SST data.

3.2.2 Marine heatwave properties during extreme marine summers

Line 384 *"EMSs exhibit lower intensity relative to the mean summer conditions"*. Do you mean "mean summer MHW conditions"?

Lines 386-387 *"This suggests either the suppression of MHWs in EMSs or the existence of alternative mechanisms potentially enhancing the MHW conditions in EMSs"*. Please, rephrase in a clearer way (can't understand).

Line 389: You say MHWs in EMS are more frequent in a just 4 summers series. I'm not sure this can be statistically sound. Please, consider the count of summer days meeting MHW conditions as an alternative variable to analyse.

Lines 395-397: Please, reconsider the frequency analysis.

Line 400: Have you checked that MHWs occur in the "warmer summer days" during the 4 EMS? You can indeed have MHW events in the "colder summer days". How do you attribute MHWs to the different parts of the RDA?

Lines 414-415 *"Given, however, the smaller variability of daily SST values during summers in the EMED, we conclude that the relative role of MHW events in the formation of EMSs is more pronounced in the EMED"*. I'm not sure this is strongly supported from the previous lines. It seems logical but it needs a more robust support. Please, extend this paragraph to support this affirmation.

Figure 6 (d,e,f). These figures show some MHW anomalies in the EMS. I see all grid points in the Mediterranean present MHW anomalies. Do you mean that any place in the Mediterranean experienced an MHW in EMS, at least one time? I'm not sure if these maps make sense, unless you can state that there have been MHWs in all grid points during EMS.

Table 1: In caption you say *"EMS identified in 1950-2020"*. Please, state that this are only 4 years.

This whole section needs to be reconsidered. I don't think that the way the four EMS are compared to the mean 1950-2020 series is appropriate (see previous comments and figure 6) comments. Consider that the four EMS experienced strong and/or long events. You should also analyse which Med areas were affected to properly decide if basin averages are adequate. It is possible that MHW events did not affect the majority of the basin so spatial MHW averages could not be appropriate, please check and justify your assumptions.

3.3.4 Illustrative example of extreme marine summer 2015

Lines 658-659: What does "*Summer 2015 has been the most widely experienced EMS in the basin"* mean? Is it possible that part of the basin experiences EMS but not the whole basin? Does EMS refere to extreme summer for the whole basin or not?

Line 659: *"non-grey locations in figure 8"*. Does this refer to figure 10?

Figure 10: What are the grey areas? Non-EMS points?

Figure 10: "*Non-coloured sea grid points in the Mediterranean Sea stand for locations that did not experience summer 2015 as extreme"*. Does your definition apply only to grid points? Is there any EMS for the whole basin? Is there an spatial extent threshold to declare a summer as an EMS for the basin?

3.3.5 Air-sea heat fluxes using detrended data

Do you detrend all datasets? Wind, SWR,… ?

***Minor comments***

Lines 45-49 do not need to be a separate paragraph; they could join the previous one.

Figure 9: Numbers in legend for figures a, b and c lead to confusion, please place them in the middle of the colour interval they stand for. The caption is too long.

Figure 10: Use different colours for land/Atlantic/Black sea and Med sea areas. Same comment for legends than for figure 9.

---

## Referee Comment (RC3)

Review: **Investigating extreme marine summers in the Mediterranean Sea**

**Major Comments:**

**Abstract:**
In general the abstract quotes the main results of the paper. However, the way it is currently written does not form a clear and concise story, rather it presents a series of results concerning the EMSs, without showing how are these connected between them. The entire abstract needs to be re-written in a more cohesive manner, forming an "argument", which shows what is known for the SST/MHW/EMS of the Med. Sea, what is the current problem faced and what is the current study's aim/methodology/added value on the subject tackled.

**Introduction:**
The introduction is very long and currently structured as:
- Importance of SST (Lines 22-30)
- Past Med. Sea SST trends (Lines 31-43). Such a detailed information on the SST variability is not necessary in the Introduction of the paper. Rather on the discussion section where the writers can compare their results with previous literature.
- Future Med. Sea SST trends (Lines 45-48)
- Decadal variability of Med. Sea SST (Lines 50-64)
- MHW trends (Lines 65 - 71)
- MHW impacts on marine life (Lines 73 - 82)
- Motivation for focusing on summer Med.SST (Lines 84 - 92)
- Explanation of the EMS concept (this belongs to Methods section) (Lines 94 -105). This paragraph describes the concept of EMS very vaguely and the reader can get confused. Is the EMS based on the MHW drivers or something else? Better to describe this in Methods and not here.
- Drivers of EMS. This again belongs to Methods (Lines 107 – 114) and it needs to be clarified if it is a paragraph on the drivers of EMS or on motivation for the investigation of the air-sea heat flux effect on EMS. See comments below
- Paper objectives (Lines 116-120). This is the only clear paragraph of the Introduction so far. However, the language needs to be improved. In addition the 4 objectives of the paper can be re-arranged to:
- Structure of daily SST in EMTs
- Role of SST multi-decadal variability of in the EMS characteristics
- Role of MHWs in EMS
- Drivers of EMS

Once the objectives of the paper are clear then the Introduction structure should describe brief background knowledge on these aspects. Also the writers should define better and briefly the EMS in the Introduction, making clear that EMS is something different from the MHWs, because currently it is not clear. The reader is confused between MHWs and EMS, and whether qualitatively and quantitatively they are the same thing or not or if one is driving the other. In general, it is not clear what the relationship between the two is and what is the added value of the EMS (or what is an EMS in general) compared to MHWs. The writers need to remove some sections from the Introduction that belong mostly to methods and should devote a brief paragraph on explaining the idea behind the EMS and its substructures here.
Additionally, in many paragraphs it is not always clear whether the writers refer to MHW/SST characteristics in the global ocean or in the Med.Sea (for example in lines 84-92), whereas the Introduction section should be more devoted to conditions relevant for

the basin, as it progresses. The relevant information from the global ocean (SST) characteristics currently are given here and there but should be used less as a motivation for a study on the basin. Also the period examined should be briefly mentioned in the objectives paragraph.

Given the entire paper needs restructuring (see general comments to the editor ), the writers should more clearly define the objective of their paper and then write an introduction providing information relevant for the paper and not just general characteristics of Mediterranean SST.
Alternative structure of the Introduction could be:
1. Brief description of Mediterranean SST, trends, decadal variability, future trends & its importance
2.Briew review of MHWs in the Med.Sea
3. Brief introduction of EMS idea and how this differs from MHWs
4. Motivation of the study/objectives.

**Methods:**
The definition of the EMS has fundamental problems:
1)  "*We then define EMSs, separately at each grid point, as the four summers with the highest average JAS SST, i.e., exceeding the 95th percentile of the 71 available summer periods from 1950 to 2020*". If the four summer periods (JAS) where the SST exceeds the 95$^{th}$ percentile are selected then it is likely these 4 summers will be towards the end of the 1950-2020(This is the first impression the readers get when they start reading the manuscript since the de-trending of the SST comes as an explanation much later in the Methods, which is confusing). In addition why the four warmest summers and not 5 or less summers? Or why not all the summers which are above the 95$^{th}$ percentile in the 1950-2020 period?Is there a specific reason for this choice? Have any sensitivity tests been performed for this choice? The choice of the 4 summers has to be well justified. Is it based on any physical quantity or previous study?

2) "*By splitting the summer rank days into different parts of the SST distribution (e.g., the coldest and the warmest half), we quantify the relative contribution of each part to the observed EMS RDA*." I do not see that kind of processing anywhere in Figure 1 so I am not really sure I understand what do the writers mean here. What do you mean by "*contribution of the observed EMS RDA*"? I only see Rank Day anomalies in Fig.1c.

**Lines 170-173:** "*The trend value of each summer season is also removed from all days belonging to this summer to obtain a detrended summer dataset of daily SST values as well*". On top of the de-trending of the multi-decadal SST you also remove the summer trend: Only from the already detrended dataset or from the non-detrended dataset as well? Why is this? What does that summer detrending offer in the study? Also, which months represent the summer detrending here? Jun-August or the Jul-Sep months used to define the EMS? How do you calculate the summer trend? Is it based on the daily summer SST of each year or the trend of the yearly-averaged SST?It is not clear. And what are the days that "*belonging to this (which) summer*" the de-trending is happening on? The choice of the summer detrending has to be justified and also the method by which the summer detrending happened has to be mentioned somewhere. At the moment this step of the method is not clearly explained.

- The term "SST substructure" could be better replaced by "SST distribution" since the ranking of daily SST resembles an SST distribution.

Alternative structure of Methods section is

**2.1** Datasets

**2.2** Definition and Detection of Extreme Marine Summers (Currently named subsections 2.2.1 and 2.2.2 belong to the same section). This is also because the detrending of the SST timeseries should be explained along with the definition of the EMSs. Otherwise the readers get the impression that the EMS are the four warm summers towards the end of the period. These two subsections need to be re-written and merged in the right order so that the reader undestands the SST warming trend does not affect the EMS definition. Currently the definition of the EMS is scattered a bit here and there in these paragraphs. Also,  justification of the choice of definition/examples on the SST substructures & SST de-trending

**2.3** If the idea of the EMS idea is retained then it has to be clearly differentiated from the MHW definition and then it would merit giving a brief descriptions of the MHWs.

**2.4** Description of EMS drivers a) Air-sea heat fluxes, b) Upper ocean preconditioning (Sections currretly named as 2.2.4 and 2.2.5 could be merged)
 - Description of the $P_{EMS}$ metric.

**Results:**

The Results section should be differentiated from the Discussion section (that needs to be created) by providing solely indicative numbers on the variables described. There has to be a quantification of the results and not a qualititative description based on visuals.
Re-structuring of Results section could be:
1. Characteristics of EMSs (in original and detrended dataset). Not sure what is the essential difference between Sections 3.1.1 and 3.1.2 but they could be merged and shortened into one clear message.
2. Role of MHWs in EMS (?) ( if EMS are clearly differentiated from MHWs)
- MHW detection
- MHW properties (original & detrended datasets)
3. Drivers of EMS. Merging of Sections 3.3.1, 3.3.2,3.3.3 where one quantifiable method of air-sea heat fluxes contribution should be chosen and described with indicative numbers.
 - original dataset
 - detrended dataset
 - Case study the MHW/EMS 2015.

**Section 3.1.1:**
**-** I understand the rational behind the focus on the contribution of the warmest and coolest half summer days in Figures 3, however, what about the days that are ranked in the middle of the summer? What about their contribution? At the moment the method sort of examines the SST distribution by only looking at its extreme statistical moments (the warmest and coolest quantiles of the distribution). But how about the mean of the SST distribution? Does this change too apart from the warmest days getting warmer? It is well known that there is a mean SST shift which drivers warm days to become warmer in the basin.
- More importantly, how is the contribution calculated? The writers simply refer to a contribution to an EMS. How is the EMS represented here as a quantity whose contribution is split into warmest and coldest days? Not clear.

**Section 3.2.2**
**Line 389**:"...*Supporting the latter scenario, MHWs in EMSs appear longer lasting as well as more frequent..*".Although increased frequency and duration of MHWs is a very

commonly highlighted result in most studies,i am first curious about a) the increased frequency. In the selected examples of Fig.4 the EMS years were only 4 out of the entire period. How can MHWs be more frequent when the EMS years are only 4 out of 70 years (1950-2020)?Or how can you compare the MHWs during EMS years which (drawn from the example of Fig.4) are less in number compared to the entire period of 70 years? And how can this comparison yield an increased frequency? How can there be an increased frequency of events that are happening in a limited number of years? Unless the writers mean something else here. The only way for a particular point to present more MHWs during EMS period compared to the normal summer periods is if these events are very short and somehow they happen very often, much more often than the normal summer. This however, contradicts the increased duration found for most points in the Med.Sea. In general this comparison of MHW characteristics during EMS and non-EMS years does not seem to have a solid foundation in a sense that the sample of EMS years is very small compared to the average summer period examined. To this end, I do not see the point of comparing the frequencies of MHWs. I am not sure what kind of quantities are compared here.

**Section 3.3.1**
 - Any kind of attempt to correlate the spatial distributions (maps) between the different air-sea heat fluxes should be accompanied by at least a pattern correlation coefficient. At the moment the comparison are described solely as visuals, without any indicative numbers or coefficients denoting correlation between the different variables. Otherwise the claims made by the writers are not substantiated. I suggest the writers re-write the entire section, at least performing pattern correlations coefficient between the maps of this section. I feel this section belongs better to a Discussion section where the writers need to be careful not to repeat the results from section 3.3.3, where the contributions od the different air-sea heat fluxes are actually quantified. Section 3.3.3 belongs better to the Results section.

**Lines 495-496:** "*Negative MLD anomalies in the western part of the Gulf of Lions also imply wind-induced mixing reduction in the vertical (Fig. 7f,g)*". First of all, I am not sure whether the MLD anomalies shown in Fig(7g) can be directly compared and linked with the anomalies of the rest of the Fig.7 figures. According to the caption of Fig.7 all the anomalies were calculated relative to the 1950-2020 period, whereas the MLD ones were based on the 1987-2019 period. Does that mean that the MLD anomalies were also computed for the EMS summers with respect to the non-EMS summers of 1987-2019 or that the writers simply created a climatology of the MLD for 1987-2019 and then computed the anomalies? Clarify. In any case, the comparison of anomalies between 1987-2020 and 1950-2020 is not advisable since the period before the 1987 encompasses a lot of the non-climate change signal. The writers should compute the anomalies of all the variable for the same period in order to be able to substantiate any claims on the link between the spatial distribution of wind and the rest of the air-sea heat flux variables. At the moment, there is no point in connecting these fields between them as their anomalies depict different things here.

**Discussion:**
- Currently there is no Discussion section in the paper and Results section contains comments which could belong to the discussion section of the paper. The entire paper needs restructuring as well as re-writing with the help from a native speaker.

**Minor Comments:**

**Abstract:**
**Line 6**: Suggestion to re-write:"*The Mediterranean Sea (MS) has  experienc significant surface warming,  during* **the** *summers when? Which years?]*  *,* **which was** *associated with devastating impacts*" to *(on what?where? Ecosystems? Society?)*"
**Line 9**: What does "*SST substructure*" mean here? For someone that reads this term the first time in the abstract it is not clear. Maybe write it differently?
**Line 10**: "*...identified in most of the basin…*". Do the writers mean that the EMSs appear more often covering a large area of the Mediterranean basin at the same time or that EMSs usually occur in many of the sub-basins of the Med.Sea (but not simultaneously necessarily)?. Better to rephrase this sentence.
**Line 11**: When the writer refers to "*ranked daily SST*", ranking can be anything and relative to different things. Here it is not explained what kind of ranking is used. It is a bit of a vague statement.
**Line 13:** "*..and more frequent than usual..*". Does "usual" mean relative to the climatology of the summer MHWs or the climatology of the summer period used to define the EMS (which by the way it is not mentioned and it need to be given as an information)? Also "*....mainly in* **the** *northern MS regions.*"
**Line 14:** "*However, the relative contribution of MHWs in EMSs is more pronounced in the central and eastern basin*". In the previous sentence the writers mention the characteristics of MHWs, which are more pronounced in the northern part of the basin during an EMS period. This sentence however, refers to a contribution (what kind of contribution is this?) of MHWs in the EMS, which is a bit confusing that is more pronounced in a different part of the basin. Also, by definition of "Extreme Marine Summers" one can understand that EMS is a different type of MHW and this sentence refers to a contribution of MHW to the EMS? It is a bit confusing. Consider rephrasing it.
**Line 17:** "*Upper ocean preconditioning is* **also important for** *the EMS formation..*"
Line 20: "*...regardless of* ***the*** *long-term trends*".

**Introduction:**
**Line 22**: "***The*** *global ocean..*". The end of this sentence needs a citation.
**Line 23-24**: "*..Environmental and societal implications –current and projected in the future– underline the need for continuous ocean monitoring and deeper understanding of the ocean climate, in terms of natural variability and anthropogenic climate change*" could be re-written as:
"The current e*nvironmental and societal implications highlight the need to improve our understanding of the anthropogenic forcing influence on the ocean climate through continuous ocean monitoring*".
**Lines 25-30***: These lines could be re-written and re-arranged as shown :*
"***The*** *Sea Surface Temperature (SST) is a fundamental climate variable and global climate change indicator. A rapidly growing literature has shown its key role in the intensification of atmospheric/oceanic events and processes, e.g., heavy precipitation events (Pastor et al., 2015), surface air temperature variations (Xu et al., 2019), MHWs under global warming (Frölicher et al., 2018), increase in global wave power (Kaur et al., 2021).* **More** *importantly, SST is the oceanic parameter that regulates air-sea energy exchanges, reflecting the role of the ocean's thermal inertia (Deser at al., 2010).*"
**Lines 31-44***:* I am not sure if such a long overview of the SST trends is needed in this introduction. I would better save the information for comparison with the paper's result in the discussion section. The writers could very briefly give a range of the documented SST trends from all the literature and continue with the paragraph of lines 45-49.

**Lines 47-49**: *The sentence could be re-written as: "The high-emission scenarios (SSP5-8.5) of the CMIP6 multi-model projections suggest an SST increase of 0.8° C to 3.5° C in the near- (2021-2040) and long-term (2081-2100) 21$^{st}$ century, respectively, relative to 1995-2014 (Iturbide et al., 2021).*

**Line 57-59:** *".. By that time, AMO has entered a declining phase, in agreement with the observed upper ocean cooling in the Atlantic that reversed the previous warming trends (Robson et al., 2016)".* Previous warming trends of the Atlantic or the Med.Sea?Not clear.

**Line 65**: Instead of "*...warm extreme oceanic events such as Marine Heatwaves (MHW), have attracted great research interest...*" better to write: "..extreme warm ocean temperature events, such as Marine heatwaves have gained great research interest.."

**Line 66-71**: These sentences could be re-written as: "***Over the past decades***, ***an*** *increased MHW intensity and frequency have been documented based on observational and modelled SST datasets, (Oliver et al., 2018; Holbrook et al., 2019; Darmaraki et al., 2019a; Juza et al., 2022; Pastor and Khodayar, 2023; Dayan et al., 2023). Further* ***increase of MHWs trends*** *is expected at* ***a*** *global and Mediterranean scale* ***over the 21$^{st}$ century*** *(Oliver et al., 2019; Darmaraki et al., 2019b; Plecha and Soares, 2019; Hayashida et al., 2020), due to anthropogenic forcing and especially under high-emission scenarios (Oliver et al., 2019).*"

**Line 73**: "*...warming trends and warm extremes is*  *motivated by their detrimental impacts...*"

**Line 79**: "*...Climate-related local extinctions..*" of what exactly?

**Line 84:** *"Summer periods are of particular interest as they are associated with greater surface warming both in present and future climate studies".* The greater surface warming is a given on summers of any time period (past or future). The reason behind the interest in summers is that this elevated warming has profound effects on marine ecosystems/communities and not because it is associated with elevated warming in current and future climate studies.

**Line 85**: "*...**The Mediterranean warming rates calculated...*"

**Line 86**: ""*..* *Indeed model projections suggest that that maximum SST increase is expected in summers*". Are you referring to the Med.Sea or to the ocean in general? Better to remain consistent and talk about the Med. basin.

**Line 87**: "*..Moreover, summer MHWs present the highest SST anomalies (Gupta et al., 2020) and are associated with a stronger ecological footprint (Oliver et al., 2019)*". Is this true on average for the global ocean or the Med. Sea? Better to build the argument behind the choice to investigate summer SST anomalies based on similar facts/studies about the Med.Sea.

**Lines 84-92**: Too many sentences with too many connecting words (e.g. Additionally, moreover, In addition etc). It is like similar arguments are being "thrown" one after the other but without forming a clear argument on the importance of summer SST. Re-write this paragraph, maybe starting with the reasons of summer SST importance step by step.

**Lines 94-95**: "*Considering the above, we propose the concept of extreme marine summer (EMS), from* ***an*** *ocean perspective, and we*  *explore* ***here*** *Mediterranean EMSs in a climatological framework.*"What do you mean by the climatological framework?

**Lines 96-105**: The entire paragraph here needs re-writing (preferably with help from a native speaker). Also it does not belong to the introduction, rather it belongs to Methods as it describes in detail the concept of EMs. Also when the writers talk about the *mean summer SST*, which years have examinated? It is not stated.

**Line 100:** "*..due to uniformly increased SST values throughout its duration..*". What do you mean by *uniformly increased SST* ?If we are in a EMS then it is a given that the SST values are going to be elevated throughout its duration,no?So what is the uniformly about?

**Line 101**: "*...due to warmer SSTs of extreme events alone. This relates to the fact that marine species have..*". What exactly is related with the marine species here? Not clear.

**Line 104:** "*..For instance, animals able to migrate to avoid anomalously warm water conditions lasting for several days may not be able to cope with longer lasting heat stress (Alexander et al., 2018)*". How much longer is the "longer lasting heat stress" from "a warm event whose duration is many days"? I do not understand this sentence. Is the duration of the EMS that differentiates it from the MHW definition and makes it more pertinent to the marine ecosystems? Or is it the change of the uniform SST change or the change in the SST substructures?

**Line 104-105**: "*Therefore, even in **the** absence of extreme SST values or extreme warm events, a summer season may present extreme mean conditions thereby strongly affecting marine life.*". How can a summer present extreme mean condition if the SST is not extreme and there is no extreme warm event?Confusing.

**Line 106**: "*An EMS may arise due to elevated SST anomalies  over the summer  period*". What do the writers consider here as summer duration? June-August? May- September? Not specified. Also elevated SST anomalies..don't already mean.."favoring of initial thermal conditions", since there is already a positive temperature anomaly?Otherwise the writers need to explain what they mean by "initial thermal conditions".Also the period examined is nowhere to be found.

**Lines 107-110**: "*The former may result from the interplay of atmospheric and oceanic factors, being the air-sea heat exchanges (turbulent and radiative fluxes), horizontal (Ekman and geostrophic currents) and vertical (entrainment, Ekman pumping) advection (Deser et al., 2010 and standard oceanography references therein)*". There is always an interplay between the atmosphere and the ocean, whether there is an EMS or not. Perhaps the writers meant the EMS emerged due to specific atmospheric and oceanographic conditions in the basin?Also if the former (SST anomalies) may be due to air-sea interactions, the latter (favoring initial thermal conditions) may be due to.. what?

**Lines 111-112**: "*The crucial role of **the** air-sea heat fluxes in **the** Mediterranean SST variability and observed warming trends in particular **has been shown in several studies** (e.g., Skliris et al., 2012; Shaltout and Omstedt, 2014)*". Is this paragraph devoted to the drivers of an EMS or to the motivation behind the investigation of the air-sea heat fluxes? This paragraph needs to be re-written and its content has to be more clear.

***Lines 113***:"*In this context,  this study **aims to**  understand the  role of air-sea heat fluxes in the formation of EMS*".

**Methods:**

**Line 129**: "*T*his **is a**  free  from diurnal variations SST product  derived from a combination of **the** HadISST2 and OSTIA datasets. Atmosphere  variables from the ERA5 product are also...*"

**Line 129-130:**"*... 10-meter wind speed, net short**wave, longwave** radiation as well as , latent and sensible surface heat fluxes, total cloud cover **and** specific humidity*.

**Lines 133:** "*All fields have a grid spacing of 0.25°x0.25° in longitude and latitude and are provided in hourly time intervals.*". Are you using hourly time interval in the study or not? If not, better to talk about the timestep you are using the datasets in the study. Also better to re-write the sentence and merge it with a previous sentence like this:" *This is a free from diurnal variations SST product derived from a combination of the HadISST2 and OSTIA datasets, with a 0.25°x0.25° spatial resolution, processed in a daily/monthly/hourly timestep*".

**Lines 134-136**: "**

*2019, at 0.05°x0.05° grid spacing, t*To cross check the quality of the reference dataset against a high-resolution observational SST dataset, **we also use the CMEMS L4 satellite SST product (EU Copernicus Marine Service Product, 2022c) for the period 1982-2019, at 0.05°x0.05° spatial resolution"**.

**Lines 135-138:** Is the CMEMS MLD different from the mixed layer thickness production extracted from the CMEMS reanalysis?Why are they mentioned twice?Confusing.

***Line 145:*** *"After identifying EMSs, we apply the methods of Röthlisberger et al. (2020) to assess the associated SST substructures at each..".* What is an SST substructure? Clarify.

**Line 154**: *"The example of Fig. 1a shows for a certain grid point that the summer of 2003 has been entirely warmer than.."*. When the writers say *"entirely warmer than"* do they mean that the entire SST distribution is anomalously warm (and not only the extreme percentiles)? Then they should define somewhere in the methods that they are looking at the SST distribution moments and their anomalies instead of using words like "entirely".

**Line 155-156**: *"RDAs in this example (Fig. 1c) are much higher for the higher rank days, suggesting that this summer is identified as extreme primarily due to the warmer summer days being warmer than usual"*. By looking at Fig.1c I see all the days of 2003 being anomalously higher than the average ranking between 1950-2020. All the days have ranking anomalies ranging from 1-3 C. Is there any criterion based on which you decide that the 3C ranking anomaly is the one primarily linked to the 2003 warming? I see all days having an anomalous contribution. So I am not convinced that the warming of 2003 was due to only the warming of the warmest days. The warmest days have the highest  anomalies but  does that mean that the anomalies of the rest of the days did not play a role in the 2003 warming? I do not understand.

**Section 2.2.2:** This section should be merged with Lines 143-145 since it is part of the EMS definition. Without this part the readers get confused with the EMS definition which is currently scattered here and there.

**Figure 2 caption**: Better to give the lat/lon coordinates used to split the Med. Into the different sub-basins instead of saying "Strait of Sicily up to 22E". The science must be reproducible, therefore, the exact coordinates of the subregions should be mentioned in the captions or in the description of the Methods. Also, the legend for the detrended SST timeseries in panel (b) says *"Med SST anomaly"* which is confusing, as usually the SST anomaly refers to the SST difference from the climatology. Better to just refer to the timeseries as "detrended summer SST". The zero line does not need a legend. It is evident.

**Line 171-172**: Better to decide to use either past or present tense in the study. Currently some sentences are in past and other in present which is confusing.

**Lines 175-179**: This paragraph may well belong to the introduction where the writers should give a brief overview of the EMSs and their added value. However, if the writers removed the signal of the multi-decadal SST oscillations how did they also remove "*any long-term variability of the Mediterranean SST*" apart from the multi-decadal oscillation one? Did you do an additional de-trending of the SST timeseries or does this refer to the multi-decadal signal? Clarify. This is confusing for the reader.

**Line 178**: *"...into the  warmest EMSs and..*."

**Line 191:** *"...Therefore, such discrepancies resulting from different climatological periods are not considered significant in the scope of the present study"*. What kind of discrepancies do you mean here?Do you test different climatology periods for the detection of MHWs?Why is this information relevant for the study?

**Lines 192-195**: *"...In addition, the availability of satellite SST data during the selected climatological period....to detrended SST data"*.All these sentences need to be re-phrased with help from a native English speaker. Also, what is the point of mentioning the comparison between the MHWs detected using ERA5 and those using satellite if a) the

comparison is not shown anywhere and b) the results from their comparison are not used anywhere else in the study?

**Line 204**: "*To investigate the role of surface heat fluxes during EMSs, we first compute the mean anomaly of EMS Qnet and its components with respect to their mean summer value over the study period...*". What do the writers mean here by computing the "*mean anomaly of EMS Qnet with respect to the mean summer value?* When one computes an anomaly and takes its average in time, then this process goes back to the mean value from where the anomalies were created. Clarify.

**Line 217**: "*Apart from the contribution of positive SST anomalies during the season, warmer than usual initial conditions may also favor...*". It is better to avoid using "initial conditions" since the writers are talking about upper ocean preconditioning in this section. The phrase "initial conditions" usually refers to initial conditions e.g. of a climate model etc.The entire section here, in fact, talks about the contribution of the ocean heat content to the EMS development whereas the previous is simply describing the atmospheric contribution. This is a common method to investigate drivers of extreme warm temperatures in the ocean, therefore the terminology should just refer to atmosphere and ocean heat contributions (or upper ocean preconditioning).

**Line 222**:"*... and the corresponding EMS anomalies (June before an EMS – mean June for all years) at each grid point..*". Better to write this as an equation instead of using words.

**Line 228:** "*To overcome any  OHC inter-comparisons due to different integration depths..*" Rephrase.

**Results:**

**Line 239:** What is the reason behind the choice of these particular 3 points whose SST substructures are investigated? Why not other points of the MS ?Is it random?Justify if not. Also, what is the period examined and represented by the grey lines? Currently we are only aware of the EMS years from the Figure legends. Caption needs to mention the entire period examined.

**Line 241**: "*Regardless **of** the variability..*"

**Figures 3 & 4**: What is the (K) in the titles and y axis label of Figures 3a) and 4a) respectively?Also panel 4 somehow is described as a whole before panel 3 of figures. I would suggest moving Figures 4 as Figures 3 and then describe the maps currently named as Figure 3, since anyway their description is now interrupted from the description of Figures 4.

**Figure 4.** The caption of the figure is missing the reference to plots d,e,f. They are only referred to as bottom row. They should be referenced together with the top row. e.g. a,d) southern of the Gulf of Lions (...) etc.

**Lines 266-267**: "*To gain insight into the SST substructures over the basin, Figs 3b and 3c show the fractional contributions of the warmest and the coldest half of the RDA to the formation of EMSs*". What do you mean by "*warmest and coldest half*"? Which ranks are the warmest and coldest ones out of the distributions shown here? Not clear.This is important in order to also understand the contribution maps of Figure 3. Also, what is the contribution of the middle ranks? The ones which are not too cold or warm?And how is the contribution calculated?Relative to what quantity? This is not clear.
- The description of Figures 3d-f comes later in Section 3.1.3 and after the description of Figures 4 & 5 which is confusing for the reader. It seems ad-hoc and the reader has to scroll up and down the manuscript. The description of the figures should be done sequentially. So section 3.1.2 should come after section 3.1.3

**Line 275**: If the variance of the RDA is calculated following the standard mathematical equation, Var = (RDA-mean RDA)^2/(number of summers). Are the writers sure that this quantity applied to the anomalous filed does not bring the Variance back to the mean? And probably this is why the maps of Fig5 resemble a lot the mas of Figure 3?Why did the writers choose the variance and not the standard deviation?

**Lines 279-280**: *"Results come in agreement with Shaltout and Omstedt (2014)  **that** examined the SST variability..."*

**Lines 279-281**: *"Results come in agreement with Shaltout and Omstedt (2014) who examined the SST variability in the basin showing that the maximum and minimum seasonal stability are found close to the southern Levantine sub-basin and the Gulf of Lions, respectively."*. This type of comments belong to the discussion and not in the result section.

**Lines 282-283**: *"We then calculate the fractional contribution of the coldest and the warmest half of the ranked summer days to the RDA variance for all 71 summers (Figs 5b,c)"*. Again which ones are the warmest and coldest half of ranked summer days, you need to name then. Otherwise it is vague. Also how do you calculate this contribution? Not clear.

**Lines 286-287**: *"In other words, the locations where the warmest (coldest) part of the rank day distribution has higher spread, experience EMSs primarily due to the contribution of the warmest (coldest) part of the EMS rank day distribution"*. In Figure 5a) I only see higher spread in northwest Med. Sea where there is also a high contribution from the warmest and coldest half of rank days (from Figure3). However there is also a high contribution from the warmest and coldest half or rank days in e.g. the north Aegean, where the spread (Fig5a) does not seem to be high. How do the writers explain this? Is their statement here true for all the areas of the Med. Sea?

**Lines 290**: *"At the location at 41.5° N-5° E (Fig. 4a), SST substructures appear particularly different among the EMSs."*.Although this is true I can also discern other years (grey lines) with the "warmest" part of the SST distribution contributing more to the summer temperatures (following the criteria of high values just like the writers). So what is the difference in this contribution with the contribution of the EMS RDA?

**Lines 292**: *"This is the part presenting the largest RDA spread in the rest of the years as well (6.7° C and 5.4° C for the warmest and the coldest part, respectively) (Fig. 4a)"*. Which years? The non EMS or the 3 other EMSs?Not clear. Also, what is the spread? Can you give the variance value?

**Line 293**: *"Similarly, at the location at 35°N-28°E (Fig. 4b), the highest rank days which contribute the most to the mean EMS RDA are also the most varying from summer to summer within the study period."*. There is no location 35° N-28° E indicated in Fig4b. The point 35° N-28° E is located in the central Levantine basin, which according to Fig5b) is characterised by an increased contribution from the high rank days. However, by looking at Fig5a) I do not see a particularly large spread in the area. In fact,the spread (variance) is much smaller than the one shown for the northwest basin.

**Line 295**: *"The RDA range of the warmest part here is 3.3° C (2.7° C for the coldest part)"*. When "range" is used then two numbers should be given that represent a *range* of what? What do the writers mean something else here. It is confusing.

**Lines 296-297**: *"In contrast, at the location 34.5° N-13° E (Fig. 4c), which exhibits greater RDA variability in the first rank days of the RDA distribution, RDA values of the coldest half display a range of 4.6° C (3.5° C for the warmest part)"*. Looking at Fig 4c) I see the RDA ranging from 1K – 3K (is this Kelvin or something else?) for the coldest half of the rank days. So where does the "range of 4.6 ° C" come from?Unless the writers mean something else.

**Line 299**: *"...the part of the RDA distribution that contributes the most to the EMS RDA is the one presenting the largest spread climatologically"*. Where is this climatological spread shown exactly? Not clear.

**Line 300**: *"When the multi-decadal trend is removed, smaller SST anomalies in EMSs are found in the entire MS. (Figs 3a,d)"*. The anomalies are smaller relative to the climatology but also in comparison with the non-detrended dataset.

**Line 306**: *"In both fields, RDA peaks are found in the Gulf of Lions and the Ligurian Sea, followed by the Adriatic Sea."* The Results section is where specific numbers should be

given. Consider mentioning the the RDA peak numbers you are referring to here. In general a range of values that represent big or small differences in the RDA between the left and right column of Fig3 should be given in the Result section.

**Lines 313**:"*In particular, in the detrended dataset, locations where the coldest rank days contribute to the RDA variability more than 50% are not restricted in the southern-central area north of the African coasts as in the original dataset. It is also the central Adriatic, part of the southern Levantine and the north Aegean Seas that exhibit a slightly enhanced contribution of the cool summer days to the RDA variability (Figs 5e,f).*". Rephrase the sentence and give more representative numbers. The result section is where values should be given, whereas the discussion section is used for commenting on the differences between the numbers of the results section.

**Lines 320**: "To better illustrate this, the SST substructures in the three example cases discussed above are very similar for the original and the detrended dataset, despite the actual EMSs being warmer (Fig. 4a-c and Fig. 4d-f, respectively)". The actual EMSs being warmer...than what? Not clear.

**Lines 320-324**: Re-phrase with help from a native speaker.

**Lines 326-329**: "In addition, SST substructures in EMSs seem to be independent of the selected study period. Sensitivity tests performed for different sub-periods show a consistent statistical behavior of the detrended SST dataset compared to the original one (not shown).". What kind of consistent statistical behavior do the writers mean here? Not clear.

**Line 331**: "…EMSs are formed based on a "background: SST substructure field, largely depending on the climatological ranked daily SST variability in the MS". I thought the SST substructures where represented through the RDA and not the climatological ranking of the daily SST. This sentence is confusing.

**Line 334**: "*Analyzing the EMS SST substructures in the previous section revealed that EMSs in the basin are commonly formed due to the warmer summer days being warmer than normal*". This is not true always, since the example case of the Central Med. Sea (Fig.4c) shows that the EMS develop mostly due to the..coldest rank of days becoming warmer.

**Line 336**: "*To complement these findings, this section investigates the role of MHW events during EMSs.*". Why does the role of MHWs acts as complementary to the EMS? Are the writers sure that there are EMSs without a MHW occurring?

**Line 336-337**: Re-phrase the sentences.

**Line 337-338**: "*Then, we examine the relative role of MHWs in EMSs by means of changes in MHW properties during EMSs (Sect. 3.2.2) with respect to mean MHW conditions*". Do the writers mean the non-EMS conditions when referring to "mean MHW conditions"?. Not clear.

**Lines 359 – 371**: These paragraphs belongs to a discussion section and not to the Results.

**Lines 269-371**: "*This difference is mostly related to our choice to focus on summer MHWs that begin and decay within the JAS summer period. Taking into account all events throughout the year, similar MHW duration values are reproduced (not shown)*".Do the writers mean that duration of the MHWs would change if the examined period would be between June – August or any other season of the year? What are the similarities of the MHW duration when taking into account all the events of the year? Similarities with the MHW duration of the current study or with previous studies?Not clear.

**Line 374-376**: "*Consequently, they represent the mean seasonal conditions of i) all summer MHWs in the study period and ii) the MHWs occurring in EMS, respectively, both not expected to be indicative of the most extreme MHW conditions.*".What do the writers mean here by "*both not expected to be indicative of..*"? The mean seasonal conditions are not expected to be indicative of MHW conditions. However the EMS are summers with

extreme warm conditions. So why are they not expected to be indicative of MHW conditions?

**Lines 377-378**: "*MHWs may promote the formation of an EMS either by being more frequent, of greater duration or of greater intensity than usual, thus, contributing in different ways –not necessarily bearing extreme characteristics– with positive SST anomalies to the seasonal SST being the focus of this study*". I am not sure the difference between the EMS and the MHWs is properly explained so far in order to understand why the MHWs *may promote EMS formation* instead of being part of/always present when an EMS is happening or how the EMS conditions are differentiated from MHW conditions, or why an EMS cannot promote a MHW instead.

**Section 3.2.2**

**Line 380**: "*To understand the role of MHWs in EMSs, this section presents the mean EMS anomalies of MHW properties with respect to their mean summer values.*". The way this sentence is written is confusing for the reader. Does that mean that this section presents the EMS anomalies of MHW properties relative to the mean EMS summer values or relative to the mean summer MHW values?

**Figure 6**: Does the computation of the mean MHW properties during 1950-2020 include the EMS years in the average or not? Because if the EMS years are included in this average what is the point in creating the anomalies of MHW properties only during EMS years (Figure 6d,e,f)? Clarify.

**Line 385**: "*In terms of mean basin values, mean summer MHW intensity equals 1.56° C while when isolating events during EMSs, it drops to 1.45° C (Table 1). This suggests either the suppression of MHWs in EMSs or the existence of alternative mechanisms potentially enhancing the MHW conditions in EMSs.*". Table 1 shows a basin-mean Imean of 1.41 ° C instead of 1.45° C. I am not sure I understand how the reduction of Imean during EMS-MHW conditions can mean either a suppresion or enhancement of MHW conditions. What do the writers mean here?What kind of mechanism are they referring to and how this differentiates when there is MHW suppresion from MHW enhancement?

**Lines 400-401**: "*MHW analysis of this part further suggests that enhanced MHW conditions commonly contribute to the observed SST anomalies in EMSs*".. I am not sure that the comparisons of Fig.6 shows this result (see comment of this Section in Major comments). But even if this is true, I do not understand why the MHWs contribute to the EMS conditions instead of being the reason why an EMS is formed in the first place?Also in Lines 407-408 the writers mention:"This suggests that in this location, the higher rank days which are the most responsible for the EMS formation (Fig. 3b), face more difficulty in producing distinct MHWs". Does the EMS create MHWs or the other way around in the end? These two sentences are a bit contradictory.

**Lines 399-415**: These two paragraphs are discussing the same thing and should be merged in one. Either more numbers of the indicative duration and intensities should be given here or move this sentences as part of the discussion.

**Section 3.2.3:**

Although this is a Results section the description of MHW properties gives no numbers. Rather it is limited in comparisons of "greater" or "smaller" magnitude. Either move these paragraphs to the discussion section offering possible mechanisms behind the results described or simply refer to the differences seen in MHW properties with the de-trended datasets using numbers.

**Lines 424**: Apart from the multi-decadal signal being removed is the trend value of the summer also removed here like in lines 170 – 173? Not clear.This information was never mentioned again in the manuscript after the beginning of the paper.

**Lines 426 – 42**7:"*This suggests the presence of similar event-driving mechanisms and reveals the positive contribution of the multi-decadal signal to the observed sea surface warming, this time specifically via the SST anomalies caused by MHWs.*". How exactly has the positive contribution of the multi-decadal signal to the observed SST warming and

(especially through the MHWs) been revealed here if the mechanisms have not changed at all in this and the original dataset? I do not understand. Clarify

**Lines 432-437**: These are arguments that belong to a discussion section and not to the results as they are commenting on literature and possible explanations.

*Line 432-433:* "During EMSs, MHW intensification is greater when using the original dataset. This is more pronounced in the eastern basin and particularly in the Aegean Sea". Apart from Table 1, Fig. 6 should be cited here as well.

**Table 1:** The categorization in "Orig" and "EMS" is not a good choice of names. Better to find some other names to distinguish between the two datasets used here.

**Line 439**: "*Moreover, we observe a similar behavior of MHW properties in the detrended compared to the original dataset*" I thought the writers were already talking about the comparison between the detrended and original dataset. So why repeat the argument here?

**Lines 440**: "'*...longer lasting events in EMSs (compared to the mean state) counterbalance the lower event intensity values in EMS (Table 1).*"This sentence is really unclear. I am not sure where to look in Table 1. Are you comparing the MHW days between the mean summer MHWs and the EMSs MHWs on the Orig dataset or the days between mean summer MHWs and the EMSs MHWs or the duration of events between the Orig and EMD datasets? Clarify. Better to rename the datasets of Table 1 and everytime you compare between them, refer to what datasets you compare. And also, how do you know that longer lasting events are "counterbalancing" the lower intensity? What do you mean by this? Not clear.

**Lines 442-446**: This paragraph refers to which Figure exactly? Not really sure where to look for these results.Is it if Figure 6? Very confusing.

**Line 442**:" *...in absence of long term trends....*". Why do you need to refer to the absent trends and what do you mean by that? That there are some trends that you have not shown or that there are no trends in the particular areas you are referring to? Not clear.

**Lines 444 – 445**: "*Among these regions, the Aegean Sea stands out as the one presenting a pronounced MHW contribution to EMSs both in the original and the detrended data...*". Again where am I supposed to look for these results? Is it figure 6 or not? Figure 6 does not include comparison between the detrended and the original dataset, no? These statemens here look like they come from nowhere since there is no reference to any figure inside the paper or any other result.

**Lines 446-447**: "*...Given that, the pronounced role of MHWs in EMSs in the eastern basin can be only partly attributed to the sea surface warming trend in the MS during the past few decades.*". How do the writers know that the pronounced role of MHWs in the MES in the eastern basin is only "partly" attributed to SST warming? Where do they show that it is "only partly" attributed? Have they compared any other variable such as trends of the MLD, winds?(e.g. trends of lower winds in the area or trends of shallower than normal MLD). I find these arguments a bit unfounded since there are no results to look at and compare at this point.

**Section 3.3**

The entire section here describes results on air-sea heat fluxes without giving any indicative numbers on the increased or decreased contributions presented in Fig.7. The description here is something between a Results section and a Discussion. The entire section should be re-written where some indicative numbers of the spatial distribution of the air-sea heat fluxes should be given, separately as Results and then in the Discussion section there should be a comparison with current knowledge and literature.

**Lines 449-458**: Is this analysis only for the original dataset and the detrended one is only shortly discussed in Section 3.3.5?Clarify in this paragraph.

**Lines 465 – 467**:" *Moreover, positive Qnet anomalies appear mainly in areas where enhanced MHW conditions in EMSs were detected (northern Mediterranean regions and particularly the Aegean and the Adriatic Seas; Fig. 7A vs Figs 4d-f)*". I am not sure what is

the point behind the connection between Fig.7a maps with the results from Fig.4d-f that refer to points. After all, the points that were (randomly?) selected from 3 regions (Fig.4) cannot represent the EMS profile of the entire area (from where they came for). In addition, we do not know if the EMSs of other points, e,g in the south-central Med. Sea exhibit the same behaviour as those in Fig.4,but are in areas characterised by negative Qnet. So I am not sure what is the argument behind this sentence. Is a cause-effect result implied here?

**Lines 468-469**: "This similarity is at least partly attributable to the widely explored driving role of air-sea heat fluxes in the formation of MHWs". How can a similarity between a map and 4 points (?) be attributed to "a widely explored role or air-sea heat fluxes"? What exactly is the similarity here and how is it related to the fact that we know a lot about air-sea heat fluxes? I do not understand the argument here. Clarify. Also, the studies cited here are global studies examining drivers of MHWs at a local scale. Better to cite studies that refer to the role of air-sea heat fluxes in Mediterranean MHWs and not on global events.

**Line 475**: Either 'above-mentioned' or 'aforementioned'.Replace.

**Lines 485 – 487**: This is a far-fetched claim to be made. Although there seems, indeed,to be a slight reduction of MLD across the Turkish coasts, I am not convinced that the reduced upwelling can be inferred solely from an MLD indicator. Especially because the MLD refers to extreme EMS summer which, at the moment, I am not quite sure which period exactly they refer to and if this period coincides with the exact period of the upwelling in the area. Also, the anomalies of the MLD apparently are computed with respect to a different baseline period, which cannot perhaps be directly compared with the anomalies of the rest of the variables of Fig.7, let alone to be linked with the impacts of EMS. In addition, is this reduction of MLD significant enough to suggest there is going to be reduced upwelling? Either compute trends of the upwelling index in the area or the authors should entirely remove this claim from the paper as it is not substantiated.

**Lines 487 – 488**; "SH flux (Fig. 7d) and net SWR anomalies (Fig. 7c) in the eastern Aegean and the central Levantine Seas are negative but of a much smaller magnitude.". How much is the magnitude here? Quantify this information wherever possible.

**Figure 7.** In the text shortwave radiation is referred to as SWR, whereas in Figure.7 as SW. Be consistent on the naming of variables everywhere.

**Lines 502 – 50**6: How can the Qnet anomalies in the Adriatic Sea be "*almost fully determined*" by shortwave radiation (line 503) when there is also a reduction of LH losses according to Fig.7b? I think the LH also plays a significant role here and it is not just the SWR.

**Line 504-505**: "*The increased LH loss in EMSs here is exceeded by the gain in net SWR, leading to the observed net heat gain*". This sentence does not make sense at all. I do not understand what the writers want to say here. Also, I do not see anywhere an increase or LH losses. I rather observe a reduction of LH losses.

**Lines 514 – 515**: "*MLD and wind speed anomalies tend to present the same sign, especially where large anomalies appear*.". Although this is true for some regions of the basin, I can see also other areas (e.g. southeast Levantine) where the windspeed is lower than normal and MLD is higher than normal. Again, at the moment there cannot be a direct comparison between MLD and any other variable of Fig.7 given the fact that the anomalous fields are computed based on different reference periods.

**Lines 509 – 520**. This paragraph again is a mix between Results (without any indicative numbers) and a Discussion which does not reveal anything novel. The writers try to claim a relationship between wind and MLD (which is already well-known), albeit on anomalous fields that are computed based on different periods. Again, this paragraph should be re-written separately for the Results section, where indicative numbers are given and the comments should be addressed in the Discussion section.

**Lines 531-532**: "*These fields suggest that LWR and SWR (standing for net values from now on) in EMSs work complementarily in the basin.*" What do you mean by complementarily? I do not understand.

**Lines 536**: The Figures of the supplementary material better be named as e.g. Fig.**S**3a instead of Fig.**B**3a. What does "**B**" stand for? I have never seen this in the literature so far. Also from Fig.B3a, I can see positive anomalies of cloud coverage in the central Aegean but also negative cloud coverage to the southern-southeast  Levantine, where the LWR anomalies are close to zero. So the statement "Indeed, positive EMS anomalies of total cloud cover derived from ERA5 are found in this area (Fig. B3a) leading to increased downwards LWR and the observed reduced net LWR loss from the sea surfaec (Fig.7e)". Is not entirely true for the Levantine basin.

**Line 538-539**: "Particularly in parts of the Adriatic and the northern Ionian Seas, the SWR surplus is responsible for the net heat gain observed in EMSs (Fig. 7a vs Figs 7b-e)". Since there is Section 3.3.3, where the contribution of each air-sea heat flux is quantified, better to combine these two section together. There is no point in describing qualitatively the air-sea heat contributions here, without giving any quantifiable information and then create a new section later (3.3.3) where the contributions are actually computed. Also, by looking the maps of Fig.7 I cannot deduce that it is only SWR surplus that is responsible for the net heat gain of the EMSs in the Adriatic and the Ionian Sea. I would argue that reduced LH losses contribute as well.

**Line 541**: "*Results suggest that Qnet anomalies (either positive or negative) in EMSs are primarily formed by LH fluxes in most areas, followed by SWR in the western and central basin and particularly in the Adriatic Sea.*". This contradicts the sentence just above, where SWR is responsible for the net heat gain!

**Section 3.3.2:**
As before, this section appears more as a Discussion section rather than a results section as no quantification is provided. This description (if needed) needs to be transferred to a discussion section.

**Line 550-552**:"*Differences in the behavior of surface heat fluxes are expected to exist among the locally detected EMSs and the mean EMS state presented in the previous Section. To shed light on this, Fig. 8 illustrates how common is a positive anomaly for each of the examined heat flux components during EMSs*". I do not see how the frequency of positive anomalies occurrence at each grid point, constitutes "*a mean EMS state*" while the analysis performed in the previous sections is on "*locally detected EMSs*". Since both analysis yield maps that means that both fields refer to locally-detected properties. I suggest to change the title of the entire section here.

**Figure 8**. I suggest changing the color pallete in this figure since red usually means higher and purple/blues lower. The red colour draws the attention a lot to the lower values of the frequency of occurrence.

**Lines 556-557**: "*The areas where positive Q net anomalies in EMSs are more frequently met are the same as where the largest positive mean EMS Q net anomalies were found (Fig. 7a vs Fig. 8a).*". I am not sure this is true. For example in Fig.7a, I can see higher Qnet anomalies in the Adriatic and the northeast Aegean, which do not exactly coincide with the highest frequency of Qnet anomalies of Fig.8a. For a better comparison between these two maps, it is better to use a pattern correlation coefficient instead of just a visual comparison. What is the point of showing both figures 7 & 8? I do not see any differences in the main message behind these two sections and the one coming after that (Section 3.3.3).

**Lines 566-567**: "*In the same locations, SWR radiation appears increased in most EMSs, reinforcing the mean EMS findings.*". What kind of EMS findings are reinforced here exactly?

***Lines 569 – 571***: "*Negative Qnet anomalies appear in every EMS in the Ligurian, the south-central as well as in a few specific spots in the WMED Sea, despite the*

*systematically increased SWR. In these areas wind appears always reduced in EMSs (although not followed by decreased LH loss), while suppression of vertical mixing is observed only in half (or less) of the EMSs (Fig. 8c)*". I am not sure where is the figure that shows negative Qnet anomalies as Fig.8 demonstrates maps on positive Qnet anomalies only. So I am not sure why Fig.8c is cited here.Confusing, Clarify.

**Lines 578 – 580**: "*…we expect that the responsible cool summer days (being warmer than usual) are mostly the early summer ones*.". The fact that the early summer days are…the cooler summer days with respect to the days during the heart of the summer .. is obvious. I do not understand the point behind this sentence.

**Lines 580 – 583**: "*In all EMSs in this area, the early summer (July) SST anomalies are the largest within the season. The increased mixed layer heat content in June is reflected in the positive SST anomalies of June while, also in May, SST in this area was found marginally larger than climatologically.*" Which figures shows all this? Confusing.

**Lines 584**: "*The temporal coverage of the modelled temperature and MLD used in the OHC calculation inevitably limits this task in using…*". Which task do you refer to? Not clear.

**Lines 586 – 587**: "*For this reason, a complementary task included the use of the detrended dataset only for locations that experience EMSs within the available period..*"? Which one exactly is the available period, not clear.

**Lines 589 – 590**:" *In the region discussed above (south-central basin), increased upper ocean heat content and spring SST values are observed before each EMS also in the detrended dataset, i.e., independently of the warming trend*". Isn't this something that one should expect given that as the summer approaches, the heat content of the ocean is ..bound to increase more and more? I am not sure why the writers expected that the de-trended dataset should show "no preconditioning" or no increase of the heat content of the ocean as we move to the summer. Also how much warming above average is the preconditioning considered as "preconditioning" and not just the normal increase of the period before the EMS? Not clear this.

**Lines 594 – 596**. "*Usefulness of this index is highlighted when examining single summers (e.g., the exemplary case of EMS 2015 in Sect. 3.3.4). Even if it is positive in most cases, its contribution is often enhanced when and where there is no causal link between the surface net heat balance and the observed surface warming thus revealing it constitutes an actually contributing EMS formation factor*". Does the "surface net heat balance" refer to Qnet? Keep consistent names throughout the paper. Generally it is not a very good practice to refer to a section and describe its results before the section itself is presented in the paper. It is disrupting for the reader. In addition, I have not seen anywhere so far in the paper **any causal link** between the surface Qnet (surface net heat balance?) and the SST warming. The only thing the authors have presented are visual comparisons between maps of physical variables, which are not supported by any good "statistical index" denoting the degree of similarity in their spatial distribution. So I would be careful not to use the words "causal" link, unless specific analysis has been done for specific events that prove the causal link.

**Line 597**: Apart from the fact that the entire paragraph does not belong to the Results section, as it mostly comments on the results, the writers suddenly go from Fig. 8d at the start of the paragraph, to referring Section 3.3.4 (that has not yet been presented) and then suddenly mention Fig.9d (also not presented yet) without having present figures Fig9.a-c before that.  In general the Figure references should be more structured in the text and should come with an order.

**Lines 598-599**:"*Although horizontal advection is not investigated in this study, this finding potentially suggests that EMS preconditioning hardly takes place in areas of intensified circulation.*". Which areas do you refer to exactly?Clarify.

**Section 3.3.3**

I am not sure how the results from this section differ from those in Sections 3.3.1 – 3.3.2. I understand the writers used an extra way to quantify this time the contributions from the different air-sea heat fluxes (apart from the quantification of the contribution that was necessary to begin with). But what is the added value of another yet way to see the contributions of the different air-sea heat fluxes to the EMS? I do not understand, qualitatively, what is the new knowledge we gained in Section 3.3.3 compared to Section 3.3.2. and 3.3.1. For example, in line 608, where the contributions are discussed, the Fig.7g is also cited to support the hypothesis that was also evident from the quantified contributions themselves. Probably this quantification metric is more meaningful than the descriptions from Section 3.3.1 and 3.3.2. However, at the moment the metric is not very clear in its explanation in the Appendix. It is quite difficult to understand how the writers constructed it, e.g. what is the difference between SSTA cum >0 (which is stated as SST anomaly in the Appendix), with the $\Delta SST'obs$? Is the second data anomalies from observations only? It is very confusing.

In addition, before the writers start describing the $P_{EMS}$ of Fig.9, they should give a qualitative definition at the start of the section to the readers as to what each positive or negative $P_{EMS}$ means and then continue with the description of their spatial destribution. At the moment the $P_{EMS}$ meaning is scattered here and there when the spatial distributions are analysed.

**Line 606**: "*In positive P EMS areas with 0 < P EMS <100%, other mechanisms are expected to work complementarily (i.e., towards greater SST anomalies)*". What does complementarily mean? What does it mean towards greater SST anomalies?

**Line 612**: "*..hence oceanic processes definitely drive EMSs in these areas*". I would avoid statements with "definitely" since the contributions from oceanic processes have not been actually proven here. They have just been inferred.

**Figure 9**: I am not sure what panels b), c) e) f) are showing exactly. What is the "warming phase" and "cooling phase" that are being subtracted here? How are they represented mathematically? Also I do not understand what is the use of an index where the warming and cooling phase are being subtracted. Clarify. In addition, in the citation of the figure, the writers should clarify the qualitative meaning of the $P_{EMS}$ metric and not just refer to it like : " $P_{EMS}$ *metric values (%) for the contribution....*". The citation is supposed to make the reader understand what are they looking at. At the moment I only understand that I am seeing some percentages of contribution.

**Lines 627, 639**: 1) "*These values indicate that oceanic processes are primarily responsible for the observed EMS SSTs in these areas*", "*Hence high SST anomalies leading to EMSs in these areas are formed despite the thermal energy deficit at the sea surface during the same periods.*" These sentences are confusing. You are saying that there are EMSs happening but..there is thermal energy deficit. How is this possible? Clarify.

**Lines 631 – 633**: "*The present section complements this finding by revealing that a thermal energy deficit at the sea surface is consistently observed in these areas even when focusing solely on highly effective (for the EMS formation) warming phases*". Rephrase.

**Line 650:** "*Results also reveal a strong link between MHW properties and surface heat fluxes (Fig. 9a vs Fig. 6d)*". I think this is a proven statement, multiple times already??? Why do you need to prove it again?

**Lines 654- 655**: "*The spatial similarity of MHW intensity and $P_{EMS}$ over the basin suggests that the crucial role of surface heat fluxes found for the northern MS is associated with their ability to drive high SST anomalies (thus intense MHWs)*". I still cannot understand the difference between EMS and MHWs in terms of high SST anomalies. They are both referring to high SST anomalies in general.

**Section 3.3.4**

The example of the Marine summer 2015 kind of suggests that there is not much of a difference between EMSs and MHWs. More specifically, Lines 569 – 661, are justifying the exploration of the EMS 2015, based on a citation about a MHW in the summer of 2015!. The writers need to either clarify what is the difference of EMS from the MHWs and thus, their usefulness compared to the MHWs or re-think the idea of EMS in total. In principle, these two definitions demonstrate two different ways to capture extreme warm temperatures in the ocean. Either, use one of them without comparison with the other or re-think the idea of the EMS.

**Lines 679-681**: "*On the other hand, the preconditioning index is clearly more pronounced here than in the positive part (Fig. 10d). Indeed, early summer days (July 2015) present…*". I thought the early summer days of the preconditioning phase was the month of June according to Section 2.2.5

***Lines 685-686***: "*Following 2015, other widely experienced EMSs such as 2003, 2012, 2018 (in descending order by means of spatial extent) are examined (not shown).*". These years also constitute MHW years in the Med. Sea. So I cannot see the usefulness/added value of the EMS definition in identifying extreme ocean periods when compared with the MHW definition.

**Section 3.3.5**

If the writers re-think the idea of the EMS and justify its usefulness as an index then the analysis of the contribution of air-sea heat fluxes in the detrended datasets deserves to be shown as a comparison to the equivalent results with the original dataset and its figures should be juxtaposed and commented in the main text.

**Conclusions and Summary:**

**Lines 794- 796**: "*Nevertheless, the multi-decadal signal contributes to the observed surface warming through more intense and slightly longer lasting MHWs, without affecting the MHW intensity spatial distribution, thus most probably also the event-driving mechanisms*". I do not understand how the multi-decadal signal contributes to more intense events without affecting the intensity spatial distribution and the event driving mechanisms. How do you infer that the event-driving mechanism is not affected? Clarify.

---

## Author Comment (AC1)

**Reply to Reviewer 1:**

We would like to thank the reviewer for their careful reading and constructive suggestions, which have significantly improved the presentation of our manuscript. We have revised the paper in line with the provided comments, and have also taken the opportunity to present some additional important insights following the provided recommendations. These actions have actually improved the quality of our manuscript.

Please, find below a point-by-point response to all comments. For ease of reference, we have assigned a number to each comment. The reviewer's comments are presented in blue font, while the authors' responses are presented in black font.

General review

C1. In this manuscript, the authors introduce the new (to my knowledge) and interesting concept of extreme marine summers. The manuscript presents a huge amount of work from the analysis of EMS to the different drivers/mechanisms involved in such EMS with interesting results.

But, in my opinion, there is a fundamental problem regarding the definition of EMS. When reading the manuscript, I could not get a clear idea of how EMS are defined.

Thank you very much for pointing out the need for a better explained definition of EMSs. It is essential that the reader understands the proposed concept.

We identify extreme summers within 1950-2020, separately at each grid point, on the basis of mean seasonal values. In particular, we define EMSs at each grid point as the summers (Jul-Aug-Sep) presenting an average summer SST above the locally computed 95$^{th}$ percentile of the mean summer SST values within 1950-2020. Considering the 71 summers of the study period, there are 4 summers exceeding this percentile threshold at each location.

An improved introduction of the new term has been included in the updated Introduction section, and a clearer and detailed definition has been included in the updated Methods of the revised manuscript, according to the clarifications provided also in our answers below.

C2. First, the authors define EMS as "define EMSs, separately at each grid point, as the four summers with the highest average JAS SST, i.e., exceeding the 95th percentile of the 71 available summer periods from 1950 to 2020". I would expect a basin definition for EMS, maybe coming after the grid point analysis. As we are talking about a season, I would expect that the categorization as extreme would affect the whole region (not with the same intensity in all grid points but with a big enough extent).

Thank you for your comment. The definition we propose applies to grid points, so the categorization of a summer as extreme does not affect the whole basin. We define EMSs locally, on the basis of mean summer SST values at each grid point. We basically treat extreme seasons as events, which may occur at a specific location. In this way, each grid point experiences its own locally detected extreme seasons. Of course, like with marine heatwaves (MHW), we expect some spatial continuity. For instance, summer 2015 (example included in the manuscript) is identified as extreme in a great part of the basin. Neighboring areas that do not experience this specific

summer as extreme (grey sea areas in Fig. 10) present a mean summer SST in 2015 that does not exceed the local (per grid point) threshold. These areas have potentially been warmer than usual, but not enough to satisfy the selected criterion, as expected with any local threshold-exceedance approach.

We agree that taking into account spatial extent is interesting and useful in the context of understanding extreme conditions affecting the basin. This is why we discussed an example case, summer 2015, being the EMS with the greatest spatial extent (i.e., with the greatest number of locations experiencing the same summer as extreme). However, this study primarily aims to treat EMSs in a statistical way rather than focus on specific summer cases. The current approach allows for investigating SST variability patterns during EMSs throughout the basin following the methodology of Röthlisberger et al. (2020). By applying a spatial extent criterion on top of the existent threshold, a different number or EMSs would be detected at different locations. In addition, locations experiencing extreme summer conditions that do not present a significant spatial extent, would then be excluded. The followed approach includes extreme seasons experienced at local scale allowing for a statistical analysis of their characteristics in a consistent way across the basin.

C3. Later in the manuscript the authors analyse different variables for the whole basin, calculate mean summer values,… maybe suggesting a basin-wide concept. But later there is a section named "Differences among local extreme marine summers".

Thank you for this comment. We do not suggest a basin-wide EMS concept. Each grid point experiences 4 locally identified EMSs. Therefore, the set of 4 EMS years for every location is not the same throughout the basin. The concept of extremity in this study relies on local SST variability. For the EMS detection we use a percentile-based threshold calculated at each location, rather than an absolute threshold value, or a threshold calculated for the whole basin. As a consequence, there are no locations with a different number of assigned EMS-years, or not experiencing EMSs at all. For this reason, there are no spatial gaps in the figures depicting variables during EMSs, except when we show results for a single summer (example EMS 2015; Fig. 10), as it was not experienced as extreme by every single location.

The Section "Differences among extreme marine summers" examines how robust is the concept of a mean EMS state, i.e., of averaging quantities over the 4 local EMSs at each grid point, not averaging over the basin. This is clearly stated in the beginning of this section in the revised manuscript.

C4. Another question that is not clarified is the choice of the 95% percentile.

Thank you for your comment. The $95^{th}$ percentile of mean summer SST values (at each grid point separately) was selected after performing sensitivity tests changing the threshold value. These tests aimed to check if important changes are observed in our results when using different thresholds, due to different characteristics of specific summers. These tests showed a consistent spatial distribution of SST anomalies and SST substructures in EMSs (as in Fig. 3a). Of course, when higher thresholds are used, the magnitude of the SST anomalies in EMSs increases, and vice versa.

Following these tests, we considered the 95$^{th}$ percentile (top 4 summers out of the 71 available summers within the study period) as a good compromise between the "extremity" of EMSs and a sufficient number of EMSs to be analyzed per grid point. A lower (higher) threshold would lead to the detection of less (more) "extreme" summers. On our concerns on the extreme character of EMSs (in relation to the threshold selection and the dataset's length), a larger dataset based on ensemble model data providing several realizations for the same period would certainly increase statistical confidence in our analysis (as Röthlisberger et al. (2020) did, in addition to the use of reanalysis data). For the moment, this remains a challenging idea for future work.

C5. Why not 90% as for extreme air and sea temperature to define extreme events?

The 90$^{th}$ percentile commonly used in MHW detection (also in this work) is applied on daily values. On the other hand, EMSs here are defined on the basis of mean seasonal values, i.e., without taking into account how daily SSTs are distributed during the season (see also our answer on C4).

C6. Why 4 summers? Are there only 4 summers exceeding the percentile? Is this needed for any single grid-point or as a basin mean? Are the 4 summers the same for the whole basin?

Thank you for this comment. As noted in C4, the selected 95$^{th}$ percentile threshold (top 5 out of 100 mean summer SST values) corresponds to the 4 warmest summers out of the 71 available summers of our study period. It is calculated separately for each location, based on the time-series of mean summer SST values of each grid point (we do not use mean basin values). Therefore, for each grid point, the 4 highest mean summer (JAS) SST values correspond to the EMS-years at this specific location. Consequently, the 4 EMS-years are not the same for the whole basin.

C7. Figure 4 shows results for three grid points where EMS occur in different years. I don't feel comfortable with a year being extreme in WMED but not in CMED. Or having a pixel suffering an EMS but not the closest one (figure 10 grey areas).

There is, of course, the possibility I did not properly understand the proposed definition. In any case, better explanation of the actual EMS definition is needed.

Thank you for pointing out the need for a better explanation. Please, see our answer in C2. We explain therein the proposed EMS definition and our choice to treat the occurrence of an extreme summer as a local event.

C8. I also have some major concerns about the analysis of MHWs. Please, see comments below.

Thank you very much for your detailed comments on the MHW analysis. Our answers are provided following your specific comments later in this document.

C9. The authors have done extensive, very solid and interesting work on the analysis of EMS drivers. A review of the above will add value to all this work and may lead to a really interesting publication. So, I encourage the authors to send a revised version of the manuscript and please understand these criticisms as a way to improve an already good work.

From the above comments, my recommendation is that the manuscript needs major revision before it is accepted for publication.

Thank you very much for this review. We believe your comments have greatly contributed in improving the paper. We have put our effort to provide a well revised version incorporating your recommendations, primarily focusing on providing a) a clearer understanding of the proposed concept of EMS and related methods and b) an improved presentation of the MHW analysis.

**2.2 Methods**

**2.2.1 Extreme marine summer definition and associated SST substructures**

C10. In lines 142-144 you define "as the four summers with the highest average JAS SST, i.e., exceeding the 95th percentile of the 71 available summer periods from 1950 to 2020". Why four? Are they the only ones exceeding the 95th percentile?

Thank you for your comment. Please, check our answer in C4. As the 95$^{th}$ percentile is computed separately at every grid point, the corresponding SST threshold value varies across the basin. By definition though, the same number of threshold exceedances exist at each location (4 in our case, considering the 71-year dataset).

C11. Why not another percentile?

Please, check our answer in C4.

C12. Have you checked if this percentile excludes some major MHW events?

EMSs are identified based on mean seasonal values, without taking into account how SST evolves within the season. If the SST anomalies during a major MHW are not large enough to make mean summer SST exceed the EMS detection threshold, this summer will not be identified as extreme at the examined location. In that case, such an event will not be included in the events occurring within EMSs.

We generally expect to meet higher MHW activity during EMSs compared to mean summer MHW conditions. This is simply because EMSs are defined on the basis of mean seasonal SST, and the elevated SSTs during MHWs are expected to contribute to the increase of the mean seasonal SST. Results in the MHW section generally confirm this assumption, also through the addition of a short analysis for MHW days following your relevant suggestion (please, see C22 for these results).

Going through the review, we understand that the presentation of methods needs improvements to increase clarity. It is important to clarify that only after detecting EMSs we investigate how they have become extreme. After detecting EMSs, the analysis of SST substructures provides information on SST distribution patterns during EMSs at different locations. Later on, the comparison of mean MHW conditions during EMSs against mean summer MHW conditions aims to further inform on the SST distribution during EMSs from an event-occurrence perspective. We have put our effort to update the relevant parts with a clearer explanation of the EMS concept and methods that depend on their detection.

C13. Please, explain clearly the process to select/exclude EMS/non EMS years. Do you choose the percentile first and then there are only four? Do you choose 4 years and look for a common percentile? I think I understand the methodology, and mostly agree, but please explain better to avoid confusion. EMS definition needs to be more robustly justified.

Thank you for your comment. Please, see our relevant answers in C4 and C6.

C14. What will happen when 2022 and 2023 are included in the study? 95th percentile will rise and maybe (probably) some of the 4 years will not meet this condition. Which impact can this have in the definition or the subsequent analysis. Please, if the data for 2022 are available, check this question.

Thank you for this comment. Having similar concerns when first applying the methodology, we had performed an additional test using different sub-periods (within 1950-2020) for the EMS detection and analysis of SST substructures. This was done to examine specific years significantly alter our results. These tests showed that, despite the expected differences in magnitude of SST anomalies, the spatial distribution of SST anomalies in EMSs and results for the revealed SST substructures are, to a great extent, independent of the study period.

In the same context, having concerns on the impact of summer 2003 (being particularly warm and associated with severe MHW conditions in a significant portion of the basin) on our results, we had repeated the analysis excluding this year. This naturally resulted in a lower SST threshold. However, results again showed very similar SST substructures, despite resulting from a different combination of locally identified EMSs.

On these grounds, we conclude that the SST substructures during EMSs primarily stem from fundamental characteristics of the climatological variability of summer SST in the basin, rather than being attributable to exceptional cases. This is further supported by our results based on detrended SST (already included in the manuscript). Specifically, despite the different combination of 4 EMS years and the absence of warming trend in the detrended dataset, results highly resemble the ones based on the original one. Therefore, we believe that the 71-year study period, even if not updated with the very latest data, is able to capture the SST substructures during anomalously warm summer conditions in the Mediterranean Sea in a present climate context.

2.2.2 Detrending time series

C15. In this section Figure 2a is mentioned. In this figure time series for sub-basins are shown and sub-basins are defined in the figure caption. It would be better to explain the sub-basins, if they are used throughout the paper for analysis, in the methodology section and it would be good to add (up to you) a figure showing them or some indication in one of the actual figures. You should also explain how you decided the limits of the basins (22º, Sicily strait,…) Is the Thyrrenian sea included in the WMED or CMED? Please explain.

Figure 2a is included in the manuscript to provide the reader an idea on the SST temporal evolution within the study period. Neither the mean basin values nor the ones for the considered sub-basins in this figure are used in our analysis, as the entire work is based on analysis per grid point.

Time-series for the different sub-basins derived from the ERA5 dataset are included only to provide an idea on the non-uniform warming trend across the basin based on the dataset in use, also showing consistency with literature. The choice of the 3 sub-regions is only indicative, and area-averaged values for these sub-regions are not used in any part of the paper. As regards the Thyrrenian Sea, it is included in WMED. By stating *"Strait of Sicily up to 22°"* in the legend of Fig. 2, we meant the strait itself being the boundary separating the 2 sub-regions, as shown in Fig. Rev 1. This has been rephrased in the revised document for clarity.

[Figure]

*Figure Rev 1 Western, Central and Eastern Med sub-regions used to compute the example time-series of Fig. 2 in the manuscript*

**2.2.3 Marine heatwave identification**

C16. For the detection of MHW events you use the fixed climatology 1983-2012, which selection is adequately explained. In a recent work Rosselló et al suggest the use of "a moving baseline covering the 20 years prior to the year under study". As you try to remove the effect of multidecadal trend, the use of a moving climatology it could also be useful. This is just a suggestion you can try or at least keep in mind for future work.

Rosselló, P., Pascual, A., & Combes, V. (2023). Assessing marine heat waves in the Mediterranean Sea: a comparison of fixed and moving baseline methods. Frontiers in Marine Science, 10. https://doi.org/10.3389/fmars.2023.1168368

Thank you for this suggestion. The use of shifting climatologies in MHW detection is indeed the most recently suggested approach for investigating MHW properties in a changing climate. A 20-year length moving baseline would remove the effect of long-term trends and would not require selecting a specific climatological period which is, to some extent, arbitrary. However, for consistency reasons, we used the same detrending method (Empirical Mode Decomposition) in any task of this work where we aimed to remove the multi-decadal trend. For MHWs, we aimed to perform a direct inter-comparison of event properties derived from the original vs the detrended dataset. For this reason, minimizing the methodological discrepancies between the two runs was

considered adequate. We certainly keep in mind this robust method for removing the effect of long-term warming trends in MHW detection, for future works.

C17. Lines 195-196: "Finally, to focus on summer MHWs, we isolated events with their onset and end day falling within the JAS summer period." Have you checked how many events are removed by using this condition? How many MHW days are removed? In the case an event started on June 28 and lasted for 30 days, would it be considered? Do you apply any spatial extent filter to detect MHWs? Just to avoid single pixel heat spikes.

Thank you for this comment. The issues you raise have also been the authors' concerns. The fixed temporal length of a season inevitably leads either to the underestimation of the number of MHWs by including only events beginning and ending within the season (approach 1), or its overestimation by including also those that fall partly within the season (approach 2). We understand that mean frequency values computed based on both approaches generate similar concerns (related to under- and over-estimation of MHWs count, respectively). However, the aim of this section is to inter-compare MHW properties between extreme and non-extreme summers over the 71 years. For this reason, we believe that the followed approach (1) is able to serve this objective despite the discussed uncertainty.

We performed two tasks in order to a) examine how many events are excluded using the followed approach and b) provide an additional way for assessing the role of MHWs in EMSs, that does not suffer from this uncertainty. The latter was done by using MHW days, and results are included in C22 (and in the revised manuscript) following your relevant suggestion thereinx.

As regards (a), we took into account also events starting before 01 Jul or ending after 30 Sep, with at least half of their duration falling within the JAS period (to avoid including events occurring mainly during spring/autumn). The number of additional events when following approach 2, is mapped in the following fig. Rev 2:

[Figure]

*Figure Rev 2 Additional summer events within 1950-2020 when using approach 2 (including also events with at least half of their duration within JAS period)*

The total MHW count within 1950-2020 and the corresponding annual frequency for the 2 approaches are presented in Fig. Rev 3: left, same approach as in the manuscript; right, for the new test including extra events:

**Approach 1:**                                                          **Approach 2:**

[Figure]

*Figure Rev 3 Left column: Total number of summer MHWs within 1950-2020 and mean frequency (events/year) using Approach 1; Right: Same as left but using Approach 2*

The additional summer events within 1950-2020 when using approach 2 are less than 8 in most of the basin (Fig. Rev 2). The increased number of events (and the corresponding increased frequency) present a quite similar spatial distribution compared to the previous results (Fig. Rev 3).

We consider that either of the two approaches is appropriate in the context of this analysis, as it solely aims to compare events among extreme and non-extreme summers. However, as intensity and duration of events (apart from event frequency) are also inter-compared between extreme and non-extreme summers, we believe that the followed approach (1) may be preferable because it incorporates information derived strictly from JAS periods.

Based on the above, we believe that it is adequate to keep the current approach, enriching the corresponding part of the methodology. In particular, we will discuss the uncertainty associated with counting summer events and computing mean summer frequency in the revised text, briefly explaining that results from different approaches do not significantly alter our conclusion on whether and where increased MHW occurrence is met. Importantly, we will also include an additional way for assessing the role of MHWs in EMSs, free of this limitation, based on MHW days. In this respect, please see our answer including relevant results in C22.

As regards spatial perspective in MHW detection, we have not applied any filter for spatial extent in this work. The entire paper is based on analysis per grid point, as explained in the clarifications provided for the proposed EMS definition. Similarly, we do not advance the Hobday et al. (2016) per pixel methodology for detecting MHWs, as the initially identified EMSs at each grid point (without any spatial filtering) are then used in the MHW analysis. We acknowledge studies suggesting spatial limitations to retain the most important MHW cases and exclude potential spikes. However, considering the objectives of this section, along with the fact that we are not targeting impacts, which would make spatial limitations of great relevance (e.g., Pastor and Khodayar, 2023), we consider that potential downsides of the employed per-pixel approach are not major in the context of this task.

**3.1.1 Decomposing the extreme marine summers**

C18. Lines 242-243

"the largest contribution to the mean EMS RDA at this location comes from the warmest part of the SST distribution (higher rank days), with the exception of EMS 1999 where the middle part contributes the most." Has this been calculated? Is there a numerical value? I can understand from looking at figure 4a but, what exactly is "the middle part"? Do you divide the rank daily anomaly in 3 periods of 30 days? Do you divide the RDA by some anomaly threshold? Line 244: Why "the 60 warmest rank days"?

I suppose you are following the methods of Roethlisberger (2020) but a clearer explanation would be good.

In this section you comment figure 3 with contributions of the warmest/coldest half to the EMS but in the analysis of the three locations you refer to "the middle part". It would be better to use the same process for EMS analysis/characterization. Or explain what you mean as "the middle part"? Is it a percentile interval of the rank days? Is there an anomaly threshold to define the RDA parts? You also mention "the 60 warmest rank days" for 2003. Please clearly explain the methodology/process to analyse rank days series.

Thank you for pointing this out. We agree that mentioning other than the 2 parts introduced in Methods (coldest/warmest half of 92 ranked SST values) confuses the reader. Although basic characteristics of other parts ("*the middle part", "the 60 warmest rank days"*) may be visually easy to observe in the discussed examples, these sentences should be rephrased for clarity. We have modified them without referring to finer parts of the distribution, to be consistent within the text and avoid confusions.

Regarding your question on how we divide the SST distribution, we do not employ any anomaly threshold or SST percentile-based intervals. SST values for each summer have been ranked before their division into different parts. Therefore, by dividing the distribution in any count of equal parts (two in our case) from the lowest to the highest rank days, we get colder towards warmer parts, of equal temporal length.

Then, following Röthlisberger et al. (2020), the contribution from the two parts (three in their study) of the SST distribution to the seasonal SST anomaly is calculated for a specific EMS $s$ as follows:

$$Contr_{coldest}(s) = \left( \frac{1}{D} \sum_{d=1}^{\frac{D}{2}} RDA_{d,s} \right) / SA_s \quad (Eq.1)$$

$$Contr_{warmest}(s) = \left( \frac{1}{D} \sum_{d=\frac{D}{2}+1}^{D} RDA_{d,s} \right) / SA_s \quad (Eq.2)$$

, where D=92 summer days, $RDA_{d,s}$ is the SST anomaly of the ranked day $d$ of the EMS $s$, with respect to the mean daily SST of the specific rank day over the 71 summers, and $SA_s$ is the seasonal SST anomaly of the EMS $s$ with respect to the mean SST over the 71 summers:

$$SA_{EMS} = \frac{1}{92} \sum_{d=1}^{92} SST_{d,EMS} - \frac{1}{71} \sum_{s=1}^{71} \frac{1}{92} \sum_{d=1}^{92} SST_{d,s}$$

Figures 3b-c in our manuscript map these contribution percentages considering all 4 EMSs at each grid point, as follows:

$$Contr_{coldest} = \frac{1}{4} \sum_{EMS=1}^{4} \frac{1}{92} \sum_{d=1}^{46} RDA_{d,EMS} / MA \quad (Eq.3)$$

$$Contr_{warmest} = \frac{1}{4} \sum_{EMS=1}^{4} \frac{1}{92} \sum_{d=47}^{92} RDA_{d,EMS} / MA \quad (Eq.4)$$

, where $MA$ is the mean SST anomaly of the 4 EMS:

$$MA = \frac{1}{4} \sum_{EMS=1}^{4} \frac{1}{92} \sum_{d=1}^{92} SST_{d,EMS} - \frac{1}{71} \sum_{s=1}^{71} \frac{1}{92} \sum_{d=1}^{92} SST_{d,s}$$

Similarly, we could split the ranked SST distribution in more than 2 parts, e.g. in 3 parts of equal length (coldest-middle-warmest) by changing the rank day summation limits (d = 1 -> D/3, d = D/3+1 -> 2D/3, d = 2D/3+1 -> D, respectively). This was actually our initial approach. However, after checking that the same main conclusion is obtained through the two-part analysis, we chose to present the latter, being probably more easy to follow.

Regarding the quantification of the relative contributions, numerical values (%) for the mean EMS state are already provided in Fig. 3b-c, (computed for the 2 parts based on Eq. 3-4). Discussing SST substructures at the example locations of Fig. 4 mainly aims to present the methodology by

showing in more detail (looking into the different local EMSs) what Fig. 3b-c provides for the mean EMS state. We have however included in the revised text these values (contribution percentages of Eq. 3-4) for the three example locations.

In addition to the aforementioned modifications, we have adjusted Methods (Sect. 2.2.1) including a clearer explanation on the process to apply the Röthlisberger et al. (2020) methodology in our case (including also the Eq. 3-4).

C19. If I understood well, there are no relevant differences in the analysis from detrended SST data.

Yes, despite the smaller magnitude of SST anomalies in EMSs resulting from the detrended dataset due to the removal of any long-term warming trend, SST substructures in EMSs throughout the basin are found to be remarkably similar with the ones based on the original one.

3.2.2 Marine heatwave properties during extreme marine summers

C20. Line 384 "EMSs exhibit lower intensity relative to the mean summer conditions". Do you mean "mean summer MHW conditions"?

Yes, thank you for pointing this out. Anomalies presented in this section are mean values for each MHW property among the 4 locally detected EMSs, with respect to the mean value of the MHW property over all 71 summers (always per grid point).

C21. Lines 386-387 "This suggests either the suppression of MHWs in EMSs or the existence of alternative mechanisms potentially enhancing the MHW conditions in EMSs". Please, rephrase in a clearer way (can't understand).

Thank you for this comment. We agree that this sentence is not clear. It refers to the negative MHW intensity anomalies in the southern parts of the basin. These anomalies suggest that MHWs are less intense during EMSs compared to mean MHW conditions in these areas. However, mean MHW intensity is only one way to describe MHW conditions. Considering the other two examined properties (frequency and duration), we see that although being less intense, MHWs in these areas occur more frequently and last longer during EMSs compared to mean MHW conditions.

We have removed this sentence, along with the previous one referring to mean-basin values. The aforementioned findings are supported in the revised manuscript without any reference to quantities averaged over the basin. For this reason, we have also removed the few extra sentences mentioning mean-basin metrics along with Table 1 that includes such values, as explained in our answer in C26. Reformulation of this section also involves the addition of our results for MHW days (according to our answer in C22), as an additional means to compare MHW conditions between extreme and non-extreme summers.

C22. Line 389: You say MHWs in EMS are more frequent in a just 4 summers series. I'm not sure this can be statistically sound. Please, consider the count of summer days meeting MHW conditions as an alternative variable to analyse.

Thank you for this comment. This sentence means to report that MHWs occur more frequently during EMSs compared to non-extreme summers.

In case your concerns originate from an unclear explanation of the way we compute EMS anomalies based on the 4 EMSs, please consider the following clarification. Our step-by-step process is as follows: Firstly, we detect all summer MHWs and compute their properties (intensity, duration, frequency) at each grid point. Secondly, we compute the mean value of each property over all summers. This is what we consider as mean MHW conditions (in terms of this property) for the summer season. Then, we compute the mean value of this property over the 4 EMSs. This is what we consider as mean MHW conditions in EMSs. EMS anomaly values for the examined MHW quantities are computed at each grid point, by subtracting the mean value over the 71 summers from the mean value over the 4 locally detected EMSs (mean EMS value – mean summer value). Importantly, to compute EMS anomalies of frequency, we do not subtract number of events, but events per year. Therefore, the sentence in Line 389 reporting that MHWs occur more frequently during EMSs compared to non-extreme summers, means that the number of events per EMS is greater than the normally expected number of events per summer.

In case your concerns are related to the limitations of the way we count summer events and compute mean frequency for EMSs, please see our clarifications on the relevant comments C17 and C26.

In any case, we agree on the added value of using MHW days. Following this request, MHW days during each summer within 1950-2020 were counted, taking into account also events that fall partly within the JAS period. The relative contribution of MHWs during EMSs with respect to mean summer MHW conditions, can be expressed in terms of MHW days, through the percentage of MHW summer days occurring within EMSs (Fig. Rev 4). Results show greater contribution percentages in the central and eastern Mediterranean, suggesting a more pronounced role of MHWs during EMSs in these areas, this time by means of temporal coverage of MHW conditions.

Based on the above, we have added Fig. Rev 4 in the Appendix, and have updated the MHW methods and relevant discussion part accordingly. This addition complements the existent results of Fig. 6 providing an additional metric for the role of MHWs in EMSs, free from the previously discussed (C17) uncertainty associated with counting events within a fixed period.

[Figure]

*Figure Rev 4 Percentage of MHW days falling within EMSs (with respect to total count of MHWs within 1950-2020)*

C23. Lines 395-397: Please, reconsider the frequency analysis.

Thank you for this comment. Please, see our answers in C17 and C22.

C24. Line 400: Have you checked that MHWs occur in the "warmer summer days" during the 4 EMS? You can indeed have MHW events in the "colder summer days". How do you attribute MHWs to the different parts of the RDA?

Thank you very much for this important comment. It motivated an actually fruitful analysis with interesting results. This paragraph (Line 400 onwards) is indeed not sufficiently supported. It relates results from the SST substructures and MHW analysis by attributing MHWs to specific parts of the SST distribution based on assumptions.

We conducted an additional analysis using MHW days, as follows: Before ranking SST values, we created a binary dataset storing if there is an active MHW at each summer day, or not. After ranking SST for each summer, we paired the ranked SST values with the above information (MHW day or not), creating a $2^{nd}$ binary dataset free of temporal order. The count of MHW days during the $1^{st}$ and $2^{nd}$ half of this dataset corresponds to the MHW days falling within the warmest and the coldest half of the rank day distribution, respectively. In this way, for each summer we were able to compute the relative contribution (%) of MHW days of the warmest and the coldest half with respect to the total count of MHW days during the summer. Figure Rev 5 shows the percentage of MHW days falling in the warmest part of the SST distribution, considering all (4 locally identified) EMSs. The corresponding percentage for the coldest part is simply the remaining percentage.

Results show that the majority of MHW days during EMSs take place within the warmest half of the SST distribution in the entire basin (Fig. Rev 5). Particularly in northern regions (northwestern Med, Adriatic, Aegean Seas), very large percentages (locally exceeding 95%) of MHW days fall within the warmest part of the SST distribution, while high percentages are also encountered in the Alboran and the Ionian Sea to the southwest of Crete. The lowest values, of about 50%, are observed in certain spots in the eastern Levantine Sea, showing that MHW days during EMSs tend to be uniformly distributed over the warmest and the coldest part of the SST distribution at these locations.

Importantly, the northern sub-regions and areas in the southeastern Ionian Sea where the vast majority of MHW days in EMSs occur within the warmest part of the summer, are also the areas presenting the higher EMS anomalies for all MHW properties (see Fig. 6d-e in the manuscript). This finding suggests that the more intense, more frequent and longer lasting events observed during EMSs occur within the warmest part of the SST distribution.

[Figure]

*Figure Rev 5 Percentage of MHW days falling within the warmest part of the SST distribution during EMSs*

Based on the above, we have reformulated this part of the manuscript, including basic results from this additional analysis and Fig. Rev5. Specifically, we suggest the inclusion of this figure (Fig. Rev5) in the Appendix, merged with the relevant Fig. Rev4 presented in C22, as follows:

[Figure]

**Suggested figure for Appendix:** *Percentage of MHW days falling within EMSs, with respect to total count of MHWs within 1950-2020 (top); Percentage of MHW days falling within the warmest part of the SST distribution during EMSs (bottom)*

C25. Lines 414-415 "Given, however, the smaller variability of daily SST values during summers in the EMED, we conclude that the relative role of MHW events in the formation of EMSs is more pronounced in the EMED". I'm not sure this is strongly supported from the previous lines. It seems logical but it needs a more robust support. Please, extend this paragraph to support this affirmation.

Thank you very much for noting this. This part should have been more thoroughly examined. To do so, here we examine the percentage difference of MHW intensity in EMSs relative to mean MHW conditions.

The produced normalized anomalies (Fig. Rev6-right) are very similar to the absolute anomalies of Fig. 6d of the manuscript (included herein in Fig. Rev6-left). The absolute increase in mean intensity during EMSs in the Adriatic, Aegean and parts of the Ionian Seas is slightly smaller than the increase observed in the northwestern basin (Fig. Rev6-left). However, normalized anomalies (Fig. Rev6-right) suggest that these intensity changes in EMSs are of approximately equal importance across the aforementioned areas. For instance, mean MHW intensity in the NW Med reaches 2.5 °C, being significantly larger than in the Aegean Sea (1.7 °C). The corresponding mean intensity anomalies in EMSs are 0.35 °C and 0.25 °C, for the NW Med and the Aegean, respectively. These anomaly values correspond to a relative increase in MHW intensity of about 14% and 14.7% for the two areas, respectively. Such small differences reveal that our initial statement regarding the relative importance of the observed changes is not supported by these results and thus we modified this part accordingly.

Nevertheless, from the perspective of temporal coverage of MHW conditions, the relative contribution of MHW days in EMSs reveals a more pronounced role of MHWs in EMSs in areas of the central and eastern Mediterranean (C22, new Fig. Rev4).

Based on the above, we have modified this sentence considering the new results from the MHW days analysis included in C22.

[Figure]

[Figure]

*Figure Rev 6 Left: MHW intensity anomaly in EMSs; Right: Normalized intensity anomaly (percentage difference (%) of MHW intensity in EMSs relative to mean intensity)*

C26. Figure 6 (d,e,f). These figures show some MHW anomalies in the EMS. I see all grid points in the Mediterranean present MHW anomalies. Do you mean that any place in the Mediterranean experienced an MHW in EMS, at least one time? I'm not sure if these maps make sense, unless you can state that there have been MHWs in all grid points during EMS.

Thank you for this comment. We agree that the methodology or the discussion of results currently do not explain why all grid points in the basin present MHW anomalies. MHWs do not occur every summer at each grid point. However, they occur at every single grid point during EMSs, as shown in the following figure (Fig. Rev 7) mapping their count. A relevant note has been included in the revised manuscript.

[Figure]

*Figure Rev 7 Number of MHWs falling within the locally identified EMSs*

C27. Table 1: In caption you say "EMS identified in 1950-2020". Please, state that this are only 4 years. This whole section needs to be reconsidered. I don't think that the way the four EMS are compared to the mean 1950-2020 series is appropriate (see previous comments and figure 6) comments. Consider that the four EMS experienced strong and/or long events. You should also analyse which Med areas were affected to properly decide if basin averages are adequate. It is possible that MHW events did not affect the majority of the basin so spatial MHW averages could not be appropriate, please check and justify your assumptions.

Thank you for this comment.

Regarding the comparison of extreme vs non-extreme conditions:

For any variable in the paper, we consider that the mean EMS conditions are represented by the mean value of that variable over the 4 EMSs, at each location. The corresponding EMS anomalies are computed by subtracting the mean value of the variable over the 71 summers from the EMS-mean. This is how EMS anomalies are computed per grid point throughout the paper. In the same way we compute the EMS anomalies also for MHW properties, as explained in C22.

On any concerns about the event occurrences across the basin in relation to the computation of EMS anomalies, we show in C26 that all grid points experience MHWs during EMSs. This has been noted in the revised text when introducing EMS anomalies for MHW properties. Clarifications on relevant concerns provided in C17 and C22, as well as the new analysis of MHW

days, have been included in the revised manuscript (please, see our answers in the above comments).

In case your concerns are related to the count of EMS-years (4), please consider our explanations in C4, C6, C22. In the context of comparing extreme and mean conditions, we consider that 4 out of 71 summers is an adequate sample to represent extreme cases.

Regarding the averaging over the basin:

Using mean basin values in this Table may indeed not be appropriate, considering the spatial differences "masked" in a single value (such as the smaller than usual intensity found over southern regions during EMSs in Fig. 6d). We also understand that no valuable insight is added through this Table, as the principal conclusions, in the context of our objectives for this section, can be derived through the maps of Fig. 6. Separating the basin into different sub-regions based on similar MHW behaviour during EMSs and then compute regional means would be adequate, still not providing additional information on top of Fig. 6. Considering also that no mean-basin approach is used in any other part of this paper, we have removed this table and the short discussion parts that refer to these mean-basin values.

3.3.4 Illustrative example of extreme marine summer 2015

C28. Lines 658-659: What does "Summer 2015 has been the most widely experienced EMS in the basin" mean? Is it possible that part of the basin experiences EMS but not the whole basin? Does EMS refer to extreme summer for the whole basin or not?

Thank you for your comment. Please, check our clarifications on the definition of EMSs provided for C2. "Summer 2015 has been the most widely experienced EMS in the basin" means that this was the summer with the greatest number of locations experiencing it as extreme. Although its spatial extent is the greatest among all detected EMSs, EMS 2015 (like every EMS) does not refer to (and does not imply extreme conditions for) the whole basin. Following the proposed definition, it is possible that only a part of the basin experiences a specific EMS.

C29. Line 659: "non-grey locations in figure 8". Does this refer to figure 10?

Thank you very much for this correction. Yes, it is a typo, the correct reference is Fig. 10.

C30. Figure 10: What are the grey areas? Non-EMS points?

Yes, grey areas are the ones that have not experienced summer 2015 as extreme.

C31. Figure 10: "Non-coloured sea grid points in the Mediterranean Sea stand for locations that did not experience summer 2015 as extreme". Does your definition apply only to grid points? Is there any EMS for the whole basin? Is there an spatial extent threshold to declare a summer as an EMS for the basin?

Thank you for your comment. Please, see our answers to these questions in the relevant comments C2 and C28. A clearer and more detailed explanation of the definition has been included in the revised version of the manuscript, based on these answers.

**3.3.5 Air-sea heat fluxes using detrended data**

C32. Do you detrend all datasets? Wind, SWR,… ?

Yes, we detrended all datasets following the same methodology as with SST to remove the multi-decadal trend (mentioned in Methods, lines 172-174).

---

## Author Comment (AC2)

**Reply to Reviewer 2:**

We would like to thank the reviewer for the valuable feedback on our manuscript. We have revised the paper in line with the provided comments and recommendations.
Please, find below a point-by-point response to all comments. For ease of reference, the reviewer's comments are presented in blue font, while the authors' responses are presented in black font.

The paper untitled "Investigating extreme marine summers in the Mediterranean Sea" - by Dimitra Denaxa Gerasimos Korres, Emmanouil Flaounas, and Maria Hatzaki – investigates Marine Heat Waves (MHW) during "extreme marine summers" in the Mediterranean Sea. The authors use a reanalysis to identify MHW characteristics and understand the main drivers explaining regional differences during particularly hot summers.

In general, I found the paper is well-written with a straightforward general structure. I really appreciated efforts of authors to clearly describe their results, even if some parts could be shortened to reduce the long length of the manuscript.

However, I have several concerns from the beginning of the paper regarding methodological choices:

1. I do not understand why the authors decide to only consider 4 events as "extreme marine summers". Even if Figure 4 gives an idea about the distribution of summer SST, this is only at three locations. The authors need to give a more precise definition of what they call "extreme marine summer" regarding the distribution.

Thank you for your comment.

As regards our choice to consider 4 extreme marine summers (EMSs):

EMSs at a particular location are the summers presenting a mean summer (Jul-Aug-Sep) SST above the local 95$^{th}$ percentile of mean summer values within 1950-2020. This percentile threshold (top 5 out of 100 mean summer SST values) corresponds to the 4 warmest summers out of the 71 available summers of our study period. It is calculated separately for each location, based on the time-series of mean summer SST values of each grid point. It should be noted here that it is after detecting EMSs on the basis of mean seasonal SST, that we investigate daily SST distributions and MHW characteristics during these extreme summers.

The 95% percentile was selected after performing sensitivity tests changing the threshold value. These tests aimed to check if important changes are observed in our results when using different thresholds, due to different characteristics of specific summers. These tests showed a consistent spatial distribution of SST anomalies and SST substructures in EMSs (as in Fig. 3a). Following these tests, we considered the 95$^{th}$ percentile (top 4 summers out of the 71 available summers within the study period) as a good compromise between the "extremity" of EMSs and a sufficient number of EMSs to be analyzed per grid point. A lower (higher) detection threshold would increase (decrease) the number of EMSs, leading to less (more) "extreme" summer conditions.

As regards Figure 4:

Discussing SST substructures at the example locations of Fig. 4 aims to present the methodology by showing in more detail what Fig. 3 shows for the entire basin using averages over the 4 EMSs. In this way we obtain a closer look into EMS conditions, as we may see the SST distributions during the different EMSs at the same location, and how the different parts of each distribution may contribute to the EMS anomalies. Please, see also our answer in minor comment for L239.

As regards the *"more precise definition of what they call "extreme marine summer" regarding the distribution":*

EMSs are identified based on mean seasonal values, without taking into account how SST evolves during the season. It is after detecting EMSs that we investigate how they have become extreme. Then, it is the analysis of SST substructures that provides information on SST distribution patterns during EMSs at different locations. Later on, the MHW analysis aims to further inform on the SST distribution during EMSs from an event-occurrence perspective.

To conclude, a summer may get unusually warm (in terms of mean seasonal SST) under different daily SST distributions during the season. But for the identification of EMSs, we do not consider any criterion relevant to the shape of the SST distribution.

We agree that the presentation of the proposed definition for EMSs needs improvements to increase clarity. We have updated the relevant parts with a clearer explanation of the EMS concept and detection method in the revised manuscript, based on the above clarifications.

> 2. My second concern is a lack of justification about applying methodologies for each "grid point". Results gives a very statistical point of view. Is there any criterion about spatial scales? This is confusing as they give physical interpretation associated with regional patterns. I would at least explain why they are not considering EMS composites.

Thank you for your comment. The definition we propose applies to grid points, so the categorization of a summer as extreme does not affect the whole basin. We define EMSs locally, on the basis of mean summer SST values of each grid point. We treat extreme seasons as events, which may occur at a specific location. In this way, each grid point experiences its own locally detected EMSs. Of course, like with marine heatwaves (MHW), we expect some continuity across the basin. For instance, summer 2015 (example included in the manuscript) is identified as extreme in a great part of the basin.

We agree that taking into account spatial extent is interesting and useful in the context of understanding extreme conditions affecting the basin. This is why we discussed an example case, summer 2015, being the EMS with the greatest spatial extent (i.e., with the greatest number of locations experiencing the same summer as extreme). However, this study primarily aims to treat EMSs in a statistical way rather than focus on specific summer cases. The current approach allows for investigating SST variability patterns during EMSs throughout the basin following the methodology of Röthlisberger et al. (2020). By applying a spatial extent (or continuity) criterion, or using composites, a different number or EMSs would be detected at different locations.

Moreover, locations experiencing extreme summer conditions that do not present a significant spatial extent, would then be excluded. The followed approach captures extreme seasons experienced at local scale allowing for a statistical analysis of their characteristics in a consistent way across the basin. We have included a brief explanation of this choice in the revised Methods.

On your concerns on the physical interpretation associated with regional patterns, please see our answer in the following comment (3). We explain therein the pre-processing tests we initially performed to ensure that the per-grid-point analysis and the fixed number of considered EMSs do not significantly alter the resulting spatial patterns.

3. If I understand Figure 3 correctly, it needs more explanation since patterns might results from one "very extreme" event for instance, and might not be representative of the 4 events. This also might be an issue in other figures.

Thank you for this comment.

Having similar concerns when first applying the methodology, we had performed an additional test using different sub-periods (within 1950-2020) for the EMS detection and analysis of SST substructures. This was done to examine if specific years significantly alter our results. These tests showed that, despite the expected differences in magnitude of SST anomalies, the spatial distribution of SST anomalies in EMSs and results for the revealed SST substructures are, to a great extent, independent of the study period.

In the same context, having concerns on the impact of summer 2003 (being particularly warm and associated with severe MHW conditions in a significant portion of the basin) on our results, we had repeated the analysis excluding this year. This naturally resulted in a lower EMS detection threshold. However, results again showed very similar SST substructures, despite resulting from a different combination of locally identified EMSs.

On these grounds, we conclude that the SST substructures during EMSs primarily stem from fundamental characteristics of the climatological variability of summer SST in the basin, rather than being attributable to exceptional cases. This is further supported by our results based on detrended SST (included in the manuscript). Specifically, despite the different combination of 4 EMS years and the absence of warming trend in the detrended dataset, results highly resemble the ones based on the original one.

4. It misses a discussion about the interest for the community of this new term "ESM".

Thank you for pointing this out. Extreme marine seasons constitute a new concept and thus require a clear definition as well as explanation of our motivation and scientific interest.

The idea of studying extreme seasons was based on the need to better understand elevated SST conditions on the seasonal timescale. As a lot of research has focused on long term warming trends and warm ocean extreme events, we believe that investigating extreme seasons in the Mediterranean Sea provides interesting insights from a different perspective.

First of all, the summer season was selected due to the greater summer SST trends (in past and future periods) compared to other seasons. After defining extreme marine seasons, on the basis of mean seasonal SST, we aimed to investigate their characteristics. The first question we begun with, is how the warmest marine summers in the basin have become such. Here comes the analysis of SST substructures during extreme summers, providing information on these summers in relation to climatological summer conditions. Such analysis of extreme seasons has not been performed (to our knowledge) for the Med. Sea and is potentially relevant to impacts on marine life, as marine species do not experience the same thermal stress in any case of elevated summer temperatures, e.g., when the coldest, or the warmest summer days are warmer than normal.

We understand (thanks to all reviewers) that it is important to prevent potential confusion between EMSs and MHWs, as EMSs are obviously related to warm events. We naturally expect a higher MHW activity during EMSs compared to mean summer MHW conditions, because MHWs contribute to the increase of the mean seasonal SST. However, MHWs are events that typically last for days and may reach several weeks according to their causing factors and the concurrent atmospheric and oceanic conditions regulating their duration. On the other hand, an extreme marine season is, by definition, of fixed duration. Extreme summer analysis therefore constitutes a different perspective for investigating anomalously warm surface conditions in the basin.

The revised manuscript explains the scientific interest behind the concept of EMS, based on the above. Relevant additions/updates (on the EMS definition, difference with MHWs, motivation and scientific interest) have been included in the Introduction and Methods of the revised manuscript.

I think these points need to be addressed before introducing this new concept. A major revision is required to clarify these points.

Minor comments:

L128 – "derives from a combination of HadISST2 and OSTIA datasets" – ERA5 SST is based on a reanalysis which includes model and observations datasets. I would reformulate.

Thank you for this suggestion. This sentence has been rephrased in the revised manuscript.

L133 – "In addition to ERA5 (reference SST dataset), we use the CMEMS L4 satellite SST product (EU Copernicus Marine Service Product, 2022c) for the period 1982-2019, at 0.05°x0.05° grid spacing, to cross check the quality of the reference dataset against a high-resolution observational SST dataset." What did you compare? What did you conclude?

Thank you for this comment. We inter-compared mean annual/seasonal SST values and corresponding linear trends for the Med. Sea and sub-basins using the 2 datasets (ERA5 vs CMEMS SST) for a common period. Indicatively, we show some basic inter-comparison results:

[Figure]

| Linear trends in Med regions (deg/year) 1982-2018 | ERA5 | CMEMS |
|---|---|---|
| **Summer JAS** | 0.039 | 0.048 |
| WMED | 0.031 | 0.040 |
| CMED | 0.036 | 0.044 |
| EMED | 0.049 | 0.060 |

The ERA5 dataset slightly overestimates SST against CMEMS observations, with a larger overestimation observed during the first years of the common period (1982-2018) and an improved match over the most recent years, as shown through the mean summer time-series for the 2 datasets presented above (top-right). ERA5 SST presents smaller increasing trends across the basin (see indicative trend values in the Table for the 3 yellow-colored sub-regions).

Satellite SST was additionally used in many tasks performed to investigate EMSs (e.g., rank day anomalies methodology, MHW detection), in the context of ensuring that the reanalysis produces similar results with the observations. As examples of such inter-comparisons, we show below the contributions from 3 different parts of the SST distribution (coldest-middle-warmest) to the rank day SST anomalies of EMSs (top left for ERA5; top right for CMEMS), and how an extreme MHW is captured by the 2 datasets (bottom).

[Figure]

[Figure]

Differences fall within acceptable ranges, considering that we inter-compare modeled against observational SST. Although we considered such tests a necessary pre-processing task, we opted

not to include any relevant results as they fall out of the paper's objectives and also to avoid increasing the length of the manuscript. We agree however on reporting in Methods that inter-comparison of ERA5 against CMEMS SST revealed a satisfactory level of similarity among the two datasets. This information has been included in the revised manuscript.

L143 – "We then define EMSs, separately at each grid point, as the four summers with the highest average JAS SST, i.e., exceeding the 95th percentile of the 71 available summer periods from 1950 to 2020. " As the authors want to introduce a new concept, it needs more explanation and justification.

Thank you for pointing out the need for a clearer definition for extreme marine seasons, as they constitute a new concept.

We define EMSs, at a particular location, as the summers presenting a mean seasonal SST above the $95^{th}$ percentile of mean summer SST values within 1950-2020. The selected $95^{th}$ percentile threshold corresponds to the 4 warmest summers out of the 71 available summers of our study period. It is calculated separately for each location, based on the time-series of mean summer SST values of each grid point. The period July-August-September (JAS) has been considered as the marine summer season, being the warmest part of the SST annual cycle in the Med. Sea (Pastor et al., 2020).

The EMS definition in Introduction and Methods has been rephrased in a clearer and more detailed manner in the revised manuscript, based on the above explanation and including clarifications requested by the three reviewers.

L239 – "we provide examples of different patterns for three grid points of the domain ». Why these points? Are they representative a regional patterns?

Thank you for your comment. The main spatial pattern of SST substructures is the increased contribution from the higher rank days in the greatest part of the basin, with main exception an area in the central-southern basin along the African coasts (as appears in Fig. 3). The reason we included Fig. 4 is to provide examples of SST distributions for different EMSs, as such information is "hidden" in the fields of Fig. 3 that include averages over the 4 local EMSs.

In particular, the location in the Levantine Sea shows a greater contribution of the warmest part to the seasonal anomaly, with SST distributions similar among the local EMSs. At the location in WMED the warmest half again contributes the most to the seasonal SST anomaly in all EMSs, but the shape of the SST distribution here varies significantly among the EMSs. This difference is related to the greater summer SST variability in the WMED (as discussed in the manuscript based on variance and contributions from different parts to variance). In contrast, the location in the southern-central basin shows a greater contribution from the lower rank days of the SST distribution during every local EMSs.

In other words, the examples of Fig. 4 confirm the information on SST substructures presented in Fig. 3, and serve to illustrate the rank anomalies methodology, by showing local rank day

distributions of SST for every EMS along with the local rank day variability of SST over the study period.

Figure 4 – This figure is interesting, but this arises questions. I would imagine with such "Extreme Marine Summer" concept, that the spatial scales would be close to those of the basin. Maybe, the authors should show several plot with seasonal SST anomaly during two or three typical ESM.

Thank you for this comment. Please, see our answer in Major Comment (2). We support therein our methodological choice to define and study extreme seasons separately at each grid point of the domain. The followed approach aims to capture extreme summers at local scale. Such EMSs may not always be significant from a basinwide perspective. Although there are EMSs experienced by a great portion of the basin (as the example provided for EMS 2015), this is not a given.

L627 – "These values indicate that oceanic processes are primarily responsible for the observed EMS SSTs in these areas " – You often refer to oceanic processes, but can you give references that support this result.

Thank you for your comment. It is important to first clarify that when we refer to the role of oceanic processes in driving EMSs we refer to the contribution of the residual term $R$ in Eq. A1. The metric we constructed to quantify the role of air-sea heat fluxes in the formation of EMSs provides a percentage contribution at each location. The remaining percentage is therefore attributed to non-heat flux (i.e., oceanic) factors. The role of oceanic factors is therefore supported by this equation, despite any uncertainties relevant to our methodological approach. It is clear that this topic is neither investigated nor discussed based on literature. However, we doubt on the added value of the latter, as this metric is first introduced in this paper and our findings cannot be easily inter-compared with literature. Investigating the role of horizontal advection in the formation of EMSs would provide valuable insights into the relevant role of oceanic processes supporting our analysis. Our choice however for this paper has been to focus only on the role of air-sea heat flux. For this reason, future work recommended by the authors within the manuscript includes a broader assessment of physical mechanisms behind the development of Mediterranean EMSs.

---

## Author Comment (AC3)

**Reply to Reviewer 3:**

We thank the reviewer for the effort and care with which they assessed this manuscript. We deeply appreciate the time they have dedicated to thoroughly evaluating our work. We have carefully addressed all comments and we have made substantial revisions to the manuscript. We sincerely believe that the provided recommendations have greatly improved the quality of the manuscript.

Please, find below a point-by-point response to all comments. For ease of reference, we have assigned a number to each comment. The reviewer's comments are presented in blue font, while the authors' responses are presented in black font.

Review: **Investigating extreme marine summers in the Mediterranean Sea**

**Major Comments:**
**C1. Abstract:**
In general the abstract quotes the main results of the paper. However, the way it is currently written does not form a clear and concise story, rather it presents a series of results concerning the EMSs, without showing how are these connected between them. The entire abstract needs to be re-written in a more cohesive manner, forming an "argument", which shows what is known for the SST/MHW/EMS of the Med. Sea, what is the current problem faced and what is the current study's aim/methodology/added value on the subject tackled.

Thank you for your comment. The abstract has been re-written in a clearer and more cohesive manner, taking also into account your specific suggestions (per-line minor comments) provided later in this review.

**C2. Introduction:**
The introduction is very long and currently structured as:
- Importance of SST (Lines 22-30)
- Past Med. Sea SST trends (Lines 31-43). Such a detailed information on the SST variability is not necessary in the Introduction of the paper. Rather on the discussion section where the writers can compare their results with previous literature.
- Future Med. Sea SST trends (Lines 45-48)
- Decadal variability of Med. Sea SST (Lines 50-64)
- MHW trends (Lines 65 - 71)
- MHW impacts on marine life (Lines 73 - 82)
- Motivation for focusing on summer Med.SST (Lines 84 - 92)
- Explanation of the EMS concept (this belongs to Methods section) (Lines 94 -105). This paragraph describes the concept of EMS very vaguely and the reader can get confused. Is the EMS based on the MHW drivers or something else? Better to describe this in Methods and not here.
- Drivers of EMS. This again belongs to Methods (Lines 107 – 114) and it needs to be clarified if it is a paragraph on the drivers of EMS or on motivation for the investigation of the air-sea heat flux effect on EMS. See comments below
- Paper objectives (Lines 116-120). This is the only clear paragraph of the Introduction so far. However, the language needs to be improved. In addition the 4 objectives of the

paper can be re-arranged to:
- Structure of daily SST in EMTs
- Role of SST multi-decadal variability of in the EMS characteristics
- Role of MHWs in EMS
- Drivers of EMS

Once the objectives of the paper are clear then the Introduction structure should describe brief background knowledge on these aspects. Also the writers should define better and briefly the EMS in the Introduction, making clear that EMS is something different from the MHWs, because currently it is not clear. The reader is confused between MHWs and EMS, and whether qualitatively and quantitatively they are the same thing or not or if one is driving the other. In general, it is not clear what the relationship between the two is and what is the added value of the EMS (or what is an EMS in general) compared to MHWs. The writers need to remove some sections from the Introduction that belong mostly to methods and should devote a brief paragraph on explaining the idea behind the EMS and its substructures here.

Additionally, in many paragraphs it is not always clear whether the writers refer to MHW/SST characteristics in the global ocean or in the Med.Sea (for example in lines 84-92), whereas the Introduction section should be more devoted to conditions relevant for the basin, as it progresses. The relevant information from the global ocean (SST) characteristics currently are given here and there but should be used less as a motivation for a study on the basin. Also the period examined should be briefly mentioned in the objectives paragraph.

Given the entire paper needs restructuring (see general comments to the editor ), the writers should more clearly define the objective of their paper and then write an introduction providing information relevant for the paper and not just general characteristics of Mediterranean SST.

Alternative structure of the Introduction could be:
1. Brief description of Mediterranean SST, trends, decadal variability, future trends & its importance
2. Briew review of MHWs in the Med.Sea
3. Brief introduction of EMS idea and how this differs from MHWs
4. Motivation of the study/objectives.

Thank you very much for this detailed analysis and suggestions. The introduction has been revised taking into account the points you raise here, as well as your line-by-line comments provided later on. The information provided on SST, MHWs and their past/future trends has been shortened and is now only relevant to the Med. Sea. More importantly, the EMS concept has been introduced in a clearer manner: our motivation to investigate anomalously high SST conditions on the seasonal scale, the proposed concept of EMSs, and differences with MHWs are clearly communicated within the revised Introduction, as recommended. Relevant explanations are provided under specific comments within this document.

**Methods:**

The definition of the EMS has fundamental problems:
**C3.**
1) "*We then define EMSs, separately at each grid point, as the four summers with the*

*highest average JAS SST, i.e., exceeding the 95th percentile of the 71 available summer periods from 1950 to 2020*". If the four summer periods (JAS) where the SST exceeds the 95th percentile are selected then it is likely these 4 summers will be towards the end of the 1950-2020(This is the first impression the readers get when they start reading the manuscript since the de-trending of the SST comes as an explanation much later in the Methods, which is confusing). In addition why the four warmest summers and not 5 or less summers? Or why not all the summers which are above the 95th percentile in the 1950-2020 period?Is there a specific reason for this choice? Have any sensitivity tests been performed for this choice? The choice of the 4 summers has to be well justified. Is it based on any physical quantity or previous study?

Thank you for your comment. We define EMSs at a particular location as the summers presenting a mean seasonal (Jul-Aug-Sep) SST above the locally computed $95^{th}$ percentile of mean summer SST values within 1950-2020. Considering the 71 summers of the study period, there are 4 summers exceeding this percentile threshold at each location.

The threshold was selected after performing sensitivity tests, to check if important changes are observed when using different percentiles (as different sets of summer-years were identified as extreme in each case). Apart from the expected increase (decrease) in the magnitude of SST anomalies when higher (lower) EMS thresholds are used, these tests showed a similar spatial distribution of SST anomalies and SST substructures in EMSs (as the ones in Fig. 3). Based on these tests, we considered the top 4 summers as a good compromise between the "extremity" of these summers and a sufficient number of summers to be analyzed per grid point. On our concerns on the extreme character of EMSs (in relation to the threshold selection and the limited dataset's length), a larger dataset based on ensemble model data would increase statistical confidence in our analysis (as Röthlisberger et al. (2020) did, in addition to the use of reanalysis data). For the moment, this remains an idea for future work.

The EMS definition in Methods has been rephrased in a clearer and more detailed manner in the revised manuscript. The choice to perform the same analysis also using detrended data, is now justified within the EMS definition in Methods, as recommended.

**C4**
2) "*By splitting the summer rank days into different parts of the SST distribution (e.g., the coldest and the warmest half), we quantify the relative contribution of each part to the observed EMS RDA.*" I do not see that kind of processing anywhere in Figure 1 so I am not really sure I understand what do the writers mean here. What do you mean by "*contribution of the observed EMS RDA*"? I only see Rank Day anomalies in Fig.1c.

Thank you for pointing out the need for further explanation of our methods.

Figure 1c shows an example of how rank day anomalies are distributed during a specific summer. We opted not to provide a detailed description of the rank anomalies methodology as it is thoroughly presented in Röthlisberger et al. (2020). We understand however that the current version of the manuscript is not sufficiently self-explanatory. For this reason, we have updated

Methods including a detailed description of the process to apply this methodology, also adding the equations used for computing the contributions. Hereafter we clarify the parts you mention:

The distribution of rank day anomalies during every EMS (as the example in Fig. 1c for summer 2003) is divided into two equal parts: the coldest and warmest part, ranging from the first to the 46th rank day and from the 47th to the 92nd rank day, respectively. Then, the methodology of Röthlisberger et al. (2020) is applied to quantify the contribution of each of the two parts of the distribution (three in their study) to the seasonal SST anomaly. These contributions are calculated for the 4 EMSs at each location (resulting in the maps of Figs 3b-c, d-e) based on the equations:

$$Contr_{coldest} = \left( \frac{1}{4} \sum_{EMS=1}^{4} \frac{1}{92} \sum_{d=1}^{46} RDA_{d,EMS} \right) / MA \quad (Eq.\,1)$$

$$Contr_{warmest} = \left( \frac{1}{4} \sum_{EMS=1}^{4} \frac{1}{92} \sum_{d=47}^{92} RDA_{d,EMS} \right) / MA \quad (Eq.\,2)$$

, where the rank day anomaly for each rank day $d$ during an EMS is obtained by subtracting the rank day mean (mean SST over the 71-summers for rank day $d$) from the SST of rank day $d$ of the examined EMS.

$$RDA_{d,EMS} = SST_{d,EMS} - \frac{1}{71} \sum_{s=1}^{71} SST_{d,s}$$

, and $MA$ is the seasonal SST anomaly averaged over the 4 EMS:

$$MA = \frac{1}{4} \sum_{EMS=1}^{4} \frac{1}{92} \sum_{d=1}^{92} SST_{d,EMS} - \frac{1}{71} \sum_{s=1}^{71} \frac{1}{92} \sum_{d=1}^{92} SST_{d,s}$$

The computed contributions ($Contr_{coldest|warmest}$ x $100\%$) represent which portion of the mean SST anomaly during EMSs is attributed to each part of the distribution.

**C5. Lines 170-173:** "*The trend value of each summer season is also removed from all days belonging to this summer to obtain a detrended summer dataset of daily SST values as well*". On top of the de-trending of the multi-decadal SST you also remove the summer trend: Only from the already detrended dataset or from the non-detrended dataset as well? Why is this? What does that summer detrending offer in the study? Also, which months represent the summer detrending here? Jun-August or the Jul-Sep months used to define the EMS? How do you calculate the summer trend? Is it based on the daily summer SST of each year or the trend of the yearly-averaged SST?It is not clear. And what are the days that "*belonging to this (which) summer*" the de-trending is happening on? The choice of the summer detrending has to be justified and also the method by which the

summer detrending happened has to be mentioned somewhere. At the moment this step
of the method is not clearly explained.

Thank you for this comment. We clarify the points you raise in the following: EMSs are detected
on the basis of mean summer values. To detect EMSs in a dataset free of multi-decadal trend, we
first created a detrended dataset of 71 mean summer values (1950-2020). This was done by
removing the multi-decadal variability from the actual mean summer SST time-series. Then, to
analyze daily SST free of multi-decadal variability, we also created a detrended dataset of daily
values. To create this dataset we used the multi-decadal signal computed above, which is a time-
series of 71 values (red dashed line in Fig. 1a). Each of these 71 values was removed from all daily
SSTs of the respective summer. Only months Jul-Aug-Sep (JAS) have been considered as marine
summer period throughout the manuscript and all processing has been applied per grid point.
Methods have been updated in the revised manuscript considering the additional information
provided above.

**C6.** The term "SST substructure" could be better replaced by "SST distribution" since the
ranking of daily SST resembles an SST distribution.

SST substructure of an extreme summer is a statistical characterization for the way daily SST
values are organized over the duration of the summer. We use this term closely following
Röthlisberger et al. (2020). We have revised manuscript introducing the term in a clearer manner,
explaining that investigating SST substructures of EMSs in this work is done by investigating how
the rank day SST distributions form the mean seasonal SST anomalies of these summers.

**C7.** Alternative structure of Methods section is
**2.1** Datasets
**2.2** Definition and Detection of Extreme Marine Summers (Currently named subsections
2.2.1 and 2.2.2 belong to the same section). This is also because the detrending of the
SST timeseries should be explained along with the definition of the EMSs. Otherwise the
readers get the impression that the EMS are the four warm summers towards the end of
the period. These two subsections need to be re-written and merged in the right order so
that the reader undestands the SST warming trend does not affect the EMS definition.
Currently the definition of the EMS is scattered a bit here and there in these paragraphs.
Also, justification of the choice of definition/examples on the SST substructures & SST
detrending

Thank you for these comments and suggestions.
Regarding detrending: We agree that the EMS definition and our choice to use both the original
and detrended datasets should be explained within the same section. In this way we make clear
from the beginning that we aim to investigate both the actually warmest EMSs (falling towards
the end of the study period) and EMSs resulting from a dataset free of multi-decadal variability.
This change has been included in the revised manuscript. However, the detailed description of the
detrending method has been kept within a separate section to maintain the flow of the text, allowing
the reader to focus on the EMS definition and methodology of SST substructures without being
interrupted by detailed methodological explanations.

Regarding our choices in the example on SST detrending: Figure 2a is included in the manuscript to provide the reader an idea on the temporal evolution of mean summer SST values in Med within 1950-2020 as derived from the ERA5 dataset. Neither the mean basin values nor the ones for the considered sub-basins in this figure are used in our analysis, as the entire analysis is performed per grid point. Figure 2b is only an example on applying the Empirical Mode Decomposition on a single time-series. We chose to show the detrended time-series for the entire basin, as the actual time-series for the basin is presented right above (Fig. 2a). These mean-basin values are not used in any part of the manuscript; they only serve as an example for detrending mean summer SST.

Regarding our choices for the example on SST substructures: Figure 1 shows the steps in computing the rank day anomalies at a grid point. We chose to show daily SST time-series for summer 2003 at a location in WMED, as 2003 was a remarkably warm summer especially in WMED.

**C8.**
**2.3** If the idea of the EMS idea is retained then it has to be clearly differentiated from the MHW definition and then it would merit giving a brief descriptions of the MHWs.

Thank you for your comment.

The idea of studying extreme seasons was based on the need to better understand elevated SST conditions on the seasonal timescale. As a lot of research has focused on long term warming trends and warm extremes, we believe that investigating extreme seasons in the Mediterranean Sea provides interesting insights from a different perspective.

On our decision to focus on the summer season in the Mediterranean Sea, this is mainly because summers present (and are projected to present in future periods) greater SST trends compared to other seasons, which is highly relevant to impacts on marine life.

After defining extreme marine seasons, as the 4 warmest seasons at each location based on mean seasonal SST, we aimed to investigate their characteristics. The first question we began with, is how the warmest marine summers in the basin have become such. Here comes the analysis of SST substructures during extreme summers, providing information on these summers in relation to climatological summer conditions.

EMSs are related to warm events. We naturally expect to meet higher MHW activity during EMSs compared to mean summer MHW conditions, as MHWs contribute to the increase of the mean seasonal SST. There are, however, fundamental differences. MHWs are events that typically last for days and may reach several weeks according to their causing factors and the concurrent atmospheric and oceanic conditions regulating their duration. On the other hand, an extreme marine season is, by definition, of fixed duration. A JAS period may be detected as extreme due to the highest (lowest) rank days, i.e., warmest (coldest) days, being anomalously warm, or due to a uniform shift of the SST seasonal distribution. Such analysis of extreme seasons has not been performed (to our knowledge) for the Mediterranean Sea.

We agree that extreme marine seasons constitute a new concept and thus require a clear definition. The revised manuscript includes an improved presentation of the extreme summer concept and motivation for this study, a clear definition of EMSs, and a short discussion on differences with MHWs based on the above clarifications.

**C9.**
**2.4** Description of EMS drivers a) Air-sea heat fluxes, b) Upper ocean preconditioning (Sections currretly named as 2.2.4 and 2.2.5 could be merged)
- Description of the PEMS metric.

Thank you for this suggestion. Both the seasonal anomalies of surface fluxes and the new metric are used to investigate surface fluxes during EMSs, we therefore believe that description of the relevant methodologies should remain within the same section (2.2.4). On the other hand, the upper-ocean preconditioning index is presented in a separate short section 2.2.5 as there is no direct link or overlapping with other methods.

**Results:**
**C10.** The Results section should be differentiated from the Discussion section (that needs to be created) by providing solely indicative numbers on the variables described. There has to be a quantification of the results and not a qualititative description based on visuals.

Thank you for very much for bringing up this concern. We chose to provide results along with their discussion taking into account characteristics of the specific tasks performed to investigate extreme summers (SST substructures, MHWs, drivers). Considering the introduction of a new concept (EMS), the use of many, occasionally non-familiar, methodologies and the length of the manuscript, we concluded that presenting the aforementioned tasks independently and discuss results within the corresponding sub-section (3.1, 3.2 and 3.3, respectively) would be easier to follow. In the section for drivers in particular (3.3), our choice to provide results along with their discussion is further related to the content of this section. We considered important to first present and qualitatively discuss basic findings for EMSs (seasonal anomalies for drivers) and then move to quantitative results from the proposed metric (please, see also C13).

Nevertheless, we have included some informative numerical values in all Results sections 3.1, 3.2 and 3.3 in the revised manuscript, as recommended. Please, see our relevant answers in C15, C16, C79, C81**.**

Re-structuring of Results section could be:
**C11.**
1. Characteristics of EMSs (in original and detrended dataset). Not sure what is the essential difference between Sections 3.1.1 and 3.1.2 but they could be merged and shortened into one clear message.

We agree on merging Sections 3.1.1 and 3.1.2. This has been done in the revised text.
As regards the separate sections for detrended data: we had initially followed this approach consistently in the manuscript, dedicating a short sub-section for results using detrended data wherever this analysis has been performed. This was done intentionally to prevent any potential

confusion between results for the two cases. However, we acknowledge that for the sake of reader comprehension, continuity and overall document's flow, merging these sections would likely enhance the manuscript's coherence. We have performed both merging suggestions in the revised manuscript.

**C12.**
2. Role of MHWs in EMS (?) ( if EMS are clearly differentiated from MHWs)
- MHW detection
- MHW properties (original & detrended datasets)

Thank you for this suggestion. We have differentiated EMSs from MHWs in the updated EMS definition in Introduction and Methods (please, see our clarifications in C8). We have also merged results from the original and detrended datasets following your suggestion (as in C11), also shrinking the latter to a single key message.

**C13.**
3. Drivers of EMS. Merging of Sections 3.3.1, 3.3.2,3.3.3 where one quantifiable method
of air-sea heat fluxes contribution should be chosen and described with indicative
numbers.
- original dataset
- detrended dataset
- Case study the MHW/EMS 2015.

Thank you for your comment and suggestions. Please, see our previous answer in C10 on the content and structure of Sect.3.3 for drivers. We choose to provide results along with their discussion taking into account the nature of the results of this section. We considered adequate to first present seasonal anomalies for air-sea heat fluxes in EMSs. We believe that these results constitute basic findings for EMSs and deserve to be presented, despite not quantifying the role of the examined variables in forming EMSs. After presenting these findings in the manuscript, we support the need for constructing a metric able to answer "what drives EMSs", also providing quantitative results. The method to quantify the role of air-sea heat fluxes in the EMS formation in the manuscript is the one for the proposed metric.

We have included some informative numerical values to support the visual comparisons in 3.3.1 following your relevant comment (please, see details in C16).

As regards the detrended dataset, we agree that this short section should be moved right after the corresponding results for the original, so that the flow is not interrupted by other results.  We have rearranged the sections accordingly.

**Section 3.1.1:**
**C14.**
- I understand the rational behind the focus on the contribution of the warmest and
coolest half summer days in Figures 3, however, what about the days that are ranked in
the middle of the summer? What about their contribution? At the moment the method
sort of examines the SST distribution by only looking at its extreme statistical moments

(the warmest and coolest quantiles of the distribution). But how about the mean of the SST distribution? Does this change too apart from the warmest days getting warmer? It is well known that there is a mean SST shift which drivers warm days to become warmer in the basin.
- More importantly, how is the contribution calculated? The writers simply refer to a contribution to an EMS. How is the EMS represented here as a quantity whose contribution is split into warmest and coldest days? Not clear.

Thank you for this comment. We could have divided the distribution into more than 2 parts, e.g. in 3 parts of equal length (coldest-middle-warmest). This was actually our initial approach. Although it provides more detailed information on the SST distribution, we considered the two-part analysis more easy to follow, while providing new information and the same key message: the warmest part of the distribution accounts for most of the seasonal SST anomaly of EMSs in the greatest part of the basin (with main exception being the southern-central region discussed in the manuscript).

To provide an example of this three-part analysis that separates the middle ranks, we show below the contributions from the coldest-middle-warmest third of the SST distribution to the seasonal anomaly in EMSs: Left from ERA5, right from Copernicus Marine SST L4 data, for a common period 1982-2018 (results on the right are produced just to cross check results with observations for the common period). The contribution percentages are computed as described in Eqs (1) and (2) provided above (C4), but using different summation limits in order to divide the ranked daily SST distribution into 3 parts, from the lowest to the highest rank day.

[Figure]

As regards your question on the mean of the distribution: It is based on the mean value of the SST distribution that we identify EMSs (a summer must present a mean summer SST above the

detection threshold to be detected as extreme; see C14). Therefore, the "well known mean shift which drives warm days to become warmer in the basin" is taken into account when investigating EMSs, in the sense that the mean of the distribution in EMSs is, by definition, greater than climatology. This applies also to the detrended data, where no long-term warming trend exists and EMSs do not fall within the recent years, but the means of the EMS distributions are, by definition, greater than the corresponding climatology.

**C15. Section 3.2.2**
**Line 389**:"...*Supporting the latter scenario, MHWs in EMSs appear longer lasting as well as more frequent..*".Although increased frequency and duration of MHWs is a very commonly highlighted result in most studies,i am first curious about a) the increased frequency. In the selected examples of Fig.4 the EMS years were only 4 out of the entire period. How can MHWs be more frequent when the EMS years are only 4 out of 70 years (1950-2020)?Or how can you compare the MHWs during EMS years which (drawn from the example of Fig.4) are less in number compared to the entire period of 70 years? And how can this comparison yield an increased frequency? How can there be an increased frequency of events that are happening in a limited number of years? Unless the writers mean something else here. The only way for a particular point to present more MHWs during EMS period compared to the normal summer periods is if these events are very short and somehow they happen very often, much more often than the normal summer. This however, contradicts the increased duration found for most points in the Med.Sea. In general this comparison of MHW characteristics during EMS and non-EMS years does not seem to have a solid foundation in a sense that the sample of EMS years is very small compared to the average summer period examined. To this end, I do not see the point of comparing the frequencies of MHWs. I am not sure what kind of quantities are compared here.

Hereafter we clarify which quantities are compared when we compute EMS anomalies for MHWs: First, we detect all summer MHWs at each grid point. Next, we compute the mean value of the examined properties (intensity, duration, frequency) over all summers. For frequency for example, this means that the total count of events is divided by the count of summers (71). This is what we consider as mean MHW conditions (in terms of each property) for the summer period. Then, we compute the value of the examined property averaged over the 4 locally detected EMSs. Similarly, frequency here is the number of MHWs falling within EMSs, divided by the number of EMSs (4). This is what we consider as mean MHW conditions in EMSs. EMS anomalies are then computed by subtracting the mean summer value from the mean EMS value. Therefore, frequency anomalies are computed by subtracting the mean summer frequency (count of MHWs per summer) from the mean frequency over the 4 EMSs (count of MHWs per EMS). We do not compare the number of events, but events per year, between all and extreme summers.

In case there are further concerns on the sample size of EMSs (4): MHW detection in this work is based on the commonly used 90[th] percentile threshold which is relatively easy to exceed and thus not expected to capture only the most extreme events. During the 4 EMSs, the total number of detected events ranges from 4 to 16, as shown in the following figure, which is a significant number

considering the number of EMSs and the normally expected event frequency (events per summer in Fig. 6c of the manuscript).

[Figure]

The produced frequency anomalies show a noticeably higher event occurrence during EMSs (see Fig. 6f from the manuscript below). This anomaly field shows that in approximately half of the basin, every EMS experiences at least 2 additional events compared to a normal summer. We believe that the consistently positive anomalies across the entire basin and their significant values in most areas constitute a worth noting result.

[Figure]

Nevertheless, we have additionally examined MHW days during EMSs, following the relative request of Referee 1. First, MHW days during each summer within 1950-2020 were counted taking into account also events that fall partly within the JAS period. The relative contribution of MHWs during EMSs with respect to mean summer MHW conditions, can be expressed in terms of MHW days, through the percentage of MHW summer days occurring within EMSs (Fig. below). Results show greater percentages in the central and eastern Mediterranean, where almost 50% of the MHWs occur within EMSs. This finding suggests a more pronounced role of MHWs during EMSs in these areas, this time by means of temporal coverage of MHW conditions.

[Figure]

*Percentage of MHW days falling within EMSs (with respect to total count of MHWs within 1950-2020)*

Based on the above, we have included this figure in the Appendix (also considering the relevant request of Reviewer 1), to enrich the existing analysis through an additional metric for the role of MHWs in EMSs, free from the discussed uncertainty.

**Section 3.3.1**
**C16.**
- Any kind of attempt to correlate the spatial distributions (maps) between the different air-sea heat fluxes should be accompanied by at least a pattern correlation coefficient. At the moment the comparison are described solely as visuals, without any indicative numbers or coefficients denoting correlation between the different variables. Otherwise the claims made by the writers are not substantiated. I suggest the writers re-write the entire section, at least performing pattern correlations coefficient between the maps of this section. I feel this section belongs better to a Discussion section where the writers need to be careful not to repeat the results from section 3.3.3, where the contributions od the different air-sea heat fluxes are actually quantified. Section 3.3.3 belongs better to the Results section.

Thank you for this comment.

We use the visual comparison of EMS anomalies of the different variables for specific regions as a first means to inform on basic spatial patterns. The patterns we see in the EMS anomaly fields are not similar for the different variables across the entire basin. Given the localized nature of these anomalies, a pattern correlation coefficient (a single value for the entire basin) would not yield meaningful insights in the context of this comparison.

Following your recommendation, we considered useful to provide numerical values for the contribution of each component to the Qnet anomaly in EMSs, rather than only qualitatively describe these contributions while presenting these maps. The draft figure below shows the percentage contribution of LH, SH, SWR and LWR to the EMS anomaly of Qnet. These percentages are computed by dividing the LH/SH/SWR/LWR anomalies of the existing Fig. 7, with the Qnet anomaly (placed on the right below). Note here that 1) positive percentage at a particular location means that the anomaly for the examined component is of the same sign as the

anomaly of Qnet (and vice versa) and that 2) the sum of contributions (positive or negative) from all components equals 100%.

[Figure]

[Figure]

This quantification confirms the dominant role of LH flux, as in most of the basin this component explains from 80% up to more than 100% of the Qnet anomalies (either positive or negative). The Adriatic, parts of the Ionian Sea (and some extra sites) are exceptions on this pattern.

The positive Qnet anomalies observed in the Adriatic Sea are due to SWR anomalies. The latter account for more than 100% of the Qnet anomaly in the positive Qnet anomaly areas, with the exception of a small area close to strait if Otranto where also LH flux contributes of about 20%. Contributions from SH and LWR to the positive Qnet anomalies are less than 10% and 30%, respectively. These findings suggest that the additional heat gain we see in most of the Adriatic during EMSs, compared to normal conditions, is attributed to SWR (net) anomalies. In limited areas across the Adriatic where we see Qnet anomalies of nearly zero, even negative values, it is the LH that systematically forms these values.

In contrast, in the southern-central basin along the African coasts, the negative Qnet anomalies are due to the LH flux. The contribution of LH flux there exceeds 100%, while the positive SWR anomalies correspond to a contribution of about -40% that compensates for the extra LH loss. Contributions from the negative SH and LWR anomalies in this area are less than 15% and 30% respectively.

We have included such numerical values to support the comparison of the different variables in the already discussed areas in Sect. 3.3.1. We believe that it is not necessary to include a figure mapping these percentages for the Qnet components (as they are actually the existing anomaly maps divided by the Qnet anomaly).

As regards the structure of Results section, please see our previous answers in C10 and C13.

**C17.**
**Lines 495-496:** "*Negative MLD anomalies in the western part of the Gulf of Lions also imply wind-induced mixing reduction in the vertical (Fig. 7f,g)*". First of all, I am not sure whether the MLD anomalies shown in Fig(7g) can be directly compared and linked with the anomalies of the rest of the Fig.7 figures. According to the caption of Fig.7 all the anomalies were calculated relative to the 1950-2020 period, whereas the MLD ones were based on the 1987-2019 period. Does that mean that the MLD anomalies were also computed for the EMS summers with respect to the non-EMS summers of 1987-2019 or that the writers simply created a climatology of the MLD for 1987-2019 and then computed the anomalies? Clarify. In any case, the comparison of anomalies between 1987-2020 and 1950-2020 is not advisable since the period before the 1987 encompasses a lot of the non-climate change signal. The writers should compute the anomalies of all the variable for the same period in order to be able to substantiate any claims on the link between the spatial distribution of wind and the rest of the air-sea heat flux variables. At the moment, there is no point in connecting these fields between them as their anomalies depict different things here.

MLD anomalies were computed by subtracting the mean climatological summer MLD value from the mean EMS value (averaged over the 4 EMSs). The difference here is that the considered climatological period is shorter than the entire study period (1987-2019 vs 1950-2020). We agree that comparing anomaly maps computed based on different climatological periods (e.g., wind speed VS MLD) is not statistically sound. However, we have checked that this not significant in our case: we have produced anomaly maps of the examined variables based on the same clim period as MLD (1987-2019). We show below these anomaly maps for Qnet, latent heat and wind speed (as they are often discussed together in the text) computed based on the entire study period (left, as in the manuscript) and based on the same period as MLD (right). Anomaly maps for the two cases are very similar, therefore we believe that showing the MLD anomalies relative to a different climatological period does not affect our conclusions. We have added a short note in the revised manuscript reporting that we have examined the impact of using a different reference period for MLD.

[Figure]

**C18. Discussion:**
- Currently there is no Discussion section in the paper and Results section contains comments which could belong to the discussion section of the paper. The entire paper needs restructuring as well as re-writing with the help from a native speaker.

Thank you for this comment.
On the structure of the paper, please see our answer in C10 and C13.
As regards any language issues, please note that the paper has been revised accordingly.

**Minor Comments:**

**Hereafter we provide a specific answer under every comment, if a question is raised or if clarifications are needed. Otherwise, please note that your suggestions and corrections (e.g., re-wording) have been carefully taken into account in the revised manuscript.**

**Abstract:**
**C19. Line 6**: Suggestion to re-write:"*The Mediterranean Sea (MS) has been experiencing significant surface warming, particularly pronounced during* **the** *summers when? Which years?] and ,* **which was** *associated with devastating impacts*" to *(on what?where? Ecosystems? Society?)*"

Thank you for this comment. This sentence aims to report that warming trends in the basin are greater during summers compared to other seasons, not compared to other summers. This will be clarified in the revised abstract. Also, devastating impacts "on marine life" has been specified.

**C20. Line 9**: What does "*SST substructure*" mean here? For someone that reads this term the first time in the abstract it is not clear. Maybe write it differently?

Thank you for noting this. Substructure of extreme summers is a statistical characterization for the way daily SST values are organized over the duration of these summers. The revised abstract properly introduces the term, explaining that by investigating SST substructures of EMSs we aim to reveal which part of the SST distribution is primarily responsible for the EMS, i.e. if these summers result primarily due to the warmest or the coldest part of the distribution being warmer than normal.

**C21. Line 10**: "*...identified in most of the basin...*". Do the writers mean that the EMSs appear more often covering a large area of the Mediterranean basin at the same time or that EMSs usually occur in many of the sub-basins of the Med.Sea (but not simultaneously necessarily)?. Better to rephrase this sentence.

Thank you for noting this. It is indeed not clear. We mean that the reported finding applies to the greatest part of the basin. It will be rephrased as follows: *Results show that, in most of the basin, EMSs are formed primarily due to the higher rank days (i.e., warmer days) being warmer than normal.*

**C22. Line 11**: When the writer refers to "*ranked daily SST*", ranking can be anything and relative to different things. Here it is not explained what kind of ranking is used. It is a bit of a vague statement.

Thank you for this comment. In the revised abstract, we used the term SST distribution (instead of *ranked daily SST distribution)* in the abstract, as more details on the ranking method are presented in methods.

**C23. Line 13:** "*..and more frequent than usual..*". Does "usual" mean relative to the climatology of the summer MHWs or the climatology of the summer period used to define the EMS (which by the way it is not mentioned and it need to be given as an information)? Also "*....mainly in **the** northern MS regions.*"

Thank you for this comment. In the sentence "*Marine heatwaves within EMSs are more intense, longer lasting, and more frequent than usual*", usual means the normally expected MHW conditions for that period of the year, i.e., the mean summer MHWs conditions within the entire period 1950-2020.

**C24.Line 14:** "*However, the relative contribution of MHWs in EMSs is more pronounced in the central and eastern basin*". In the previous sentence the writers mention the characteristics of MHWs, which are more pronounced in the northern part of the basin during an EMS period. This sentence however, refers to a contribution (what kind of contribution is this?) of MHWs in the EMS, which is a bit confusing that is more pronounced in a different part of the basin. Also, by definition of "Extreme Marine Summers" one can understand that EMS is a different type of MHW and this sentence refers to a contribution of MHW to the EMS? It is a bit confusing. Consider rephrasing it.

Thank you for noting this, this sentence has been corrected, also including relevant results for MHW days (please, see our answer on MHW days analysis in C15 and the revised abstract). The revised text clarifies on the basis of which MHW property we report enhanced MHW conditions during EMSs.

**C25.Line 17:** "*Upper ocean preconditioning is **also important for** the EMS formation..*"
Line 20: "*...regardless of **the** long-term trends*".

**Introduction:**
**C26.Line 22**: "***The** global ocean..*". The end of this sentence needs a citation.
**C27.Line 23-24**: "*..Environmental and societal implications –current and projected in the future– underline the need for continuous ocean monitoring and deeper understanding of the ocean climate, in terms of natural variability and anthropogenic climate change*"
could be re-written as:
"The current e*nvironmental and societal implications highlight the need to improve our understanding of the anthropogenic forcing influence on the ocean climate through continuous ocean monitoring*".
**C28.Lines 25-30**: *These lines could be re-written and re-arranged as shown :*
"***The** Sea Surface Temperature (SST) is a fundamental climate variable and global climate change indicator. A rapidly growing literature has shown its key role in the intensification of atmospheric/oceanic events and processes, e.g., heavy precipitation events (Pastor et al., 2015), surface air temperature variations (Xu et al., 2019), MHWs under global warming (Frölicher et al., 2018), increase in global wave power (Kaur et al., 2021). **More** importantly, SST is the oceanic parameter that regulates air-sea energy exchanges, reflecting the role of the ocean's thermal inertia (Deser at al., 2010).*"
**C29.Lines 31-44**: I am not sure if such a long overview of the SST trends is needed in this introduction. I would better save the information for comparison with the paper's result in the discussion section. The writers could very briefly give a range of the documented SST trends from all the literature and continue with the paragraph of lines 45-49.
**C30.Lines 47-49**: *The sentence could be re-written as: "The high-emission scenarios (SSP5-8.5) of the CMIP6 multi-model projections suggest an SST increase of 0.8° C to 3.5° C in the near- (2021-2040) and long-term (2081-2100) 21st century, respectively, relative to 1995-2014 (Iturbide et al., 2021).*
**C31.Line 57-59**: *".. By that time, AMO has entered a declining phase, in agreement with the observed upper ocean cooling in the Atlantic that reversed the previous warming trends (Robson et al., 2016)".* Previous warming trends of the Atlantic or the Med.Sea?Not clear.

Thank you for noting this. We will specify in the revised introduction that we refer to the Atlantic.

**C32.Line 65**: Instead of "*...warm extreme oceanic events such as Marine Heatwaves (MHW), have attracted great research interest...*" better to write: "..extreme warm ocean temperature events, such as Marine heatwaves have gained great research interest.."
**C33.Line 66-71**: These sentences could be re-written as: "***Over the past decades**, **an** increased MHW intensity and frequency have been documented based on observational and modelled SST datasets, (Oliver et al., 2018; Holbrook et al., 2019; Darmaraki et al.,*

*2019a; Juza et al., 2022; Pastor and Khodayar, 2023; Dayan et al., 2023). Further increase of MHWs trends is expected at a global and Mediterranean scale over the 21st century (Oliver et al., 2019; Darmaraki et al., 2019b; Plecha and Soares, 2019; Hayashida et al., 2020), due to anthropogenic forcing and especially under high-emission scenarios (Oliver et al., 2019)."*

**C34.Line 73**: *"...warming trends and warm extremes is highly motivated by their detrimental impacts..."*

**C35.Line 79**: *"...Climate-related local extinctions.."* of what exactly?

**C36.Line 84:** *"Summer periods are of particular interest as they are associated with greater surface warming both in present and future climate studies".* The greater surface warming is a given on summers of any time period (past or future). The reason behind the interest in summers is that this elevated warming has profound effects on marine ecosystems/communities and not because it is associated with elevated warming in current and future climate studies.

Thank you for your comment. This sentence aims to highlight that surface warming trends for summers are greater compared to other seasons, which is a motivating factor for focusing on the summer season. We agree that the primary motivation for studying warm water conditions in a warming Med. Sea context is their impacts on marine life. This is why the Introduction includes a dedicated paragraph (lines 72-82). To avoid such misinterpretation, we will rephrase this sentence as follows: *"Summers are of particular interest as they are associated with greater surface warming **trends, compared to other seasons**, both in present and future climate studies".*

**C37.Line 85**: *"...Actually, .tThe Mediterranean warming rates calculated..."*

**C38.Line 86**: *""..Additionally, Indeed model projections suggest that that maximum SST increase is expected in summers".* Are you referring to the Med.Sea or to the ocean in general? Better to remain consistent and talk about the Med. Basin.

Thank you, we refer to the Med. basin. Information on the past/future trends of SST and MHWs (and impacts of ocean warming) in the Introduction refer only to the Med. Sea in the revised manuscript.

**C39.Line 87**: *"..Moreover, summer MHWs present the highest SST anomalies (Gupta et al., 2020) and are associated with a stronger ecological footprint (Oliver et al., 2019)".* Is this true on average for the global ocean or the Med. Sea? Better to build the argument behind the choice to investigate summer SST anomalies based on similar facts/studies about the Med.Sea.

This is true for the global ocean (the cited papers study the global ocean), but this is also true for the Med. Sea. Please, see also C39.

**C40.Lines 84-92**: Too many sentences with too many connecting words (e.g. Additionally, moreover, In addition etc). It is like similar arguments are being "thrown" one after the other but without forming a clear argument on the importance of summer SST. Re-write this paragraph, maybe starting with the reasons of summer SST importance step by step.

Thank you for your comment. The paragraph has been re-written in a clearer way.

**C41.Lines 94-95**: "*Considering the above, we propose the concept of extreme marine summer (EMS), from **an** ocean perspective, and we put our effort to explore **here** Mediterranean EMSs in a climatological framework.*"What do you mean by the climatological framework?

We investigate extreme summers in relation to cilmatology. We show and discuss the deviation of each examined quantity in EMSs (SST distribution, MHWs, drivers) from the climatologically expected summer conditions. This has been rephrased in the revised Introduction as follows: *[…]to explore here Mediterranean EMSs in relation to summer climatological conditions.*

**C42.Lines 96-105**: The entire paragraph here needs re-writing (preferably with help from a native speaker). Also it does not belong to the introduction, rather it belongs to Methods as it describes in detail the concept of EMs. Also when the writers talk about the *mean summer SST*, which years have examinated? It is not stated.

Thank you for this comment. Mean summer SST is computed for all years within 1950-2020, at each grid point. These are the timeseries we use to detect EMSs. This is now clear in the revised text. An updated presentation of the EMS concept and motivation for the srtudy is included in the revised Introduction, and a clear EMS definition along with a step-by-step explanation of the EMS detection process is provided in the revised Methods Section, based on our answers in C3 and C8.

**C43.Line 100:** "*..due to uniformly increased SST values throughout its duration..*". What do you mean by *uniformly increased SST* ?If we are in a EMS then it is a given that the SST values are going to be elevated throughout its duration,no?So what is the uniformly about?

Summers are detected as extreme based on their mean summer SST, without taking into account how SST values are distributed during the season. Different parts of the SST distribution may be anomalously warm, leading to a mean seasonal SST above the EMS detection threshold. A uniformly elevated SST throughout the duration of a summer may be responsible for making the summer extreme, but it is not a given.

**C44.Line 101**: "*...due to warmer SSTs of extreme events alone. This relates to the fact that marine species have..*". What exactly is related with the marine species here? Not clear.

Thank you for noting this. Investigating SST substructures revealed that for a great portion of the Med. basin, it is primarily the warmest part of the SST distribution during EMSs that is anomalously warm. Marine species do not experience the same thermal stress in any case of elevated summer temperatures. They are, for example, more susceptible to temperature increases if they live in regions close to their thermal limits (e.g., Smale et al., 2019). Therefore, informing on whether the coldest, or the warmest summer days are warmer than normal during extreme summers in different regions in the basin is related to impacts on marine organisms. We have revised this paragraph accordingly.

**C44.Line 104:** "*..For instance, animals able to migrate to avoid anomalously warm water conditions lasting for several days may not be able to cope with longer lasting heat stress (Alexander et al., 2018)*". How much longer is the "longer lasting heat stress" from "a warm event whose duration is many days"? I do not understand this sentence. Is the duration of the EMS that differentiates it from the MHW definition and makes it more pertinent to the marine ecosystems? Or is it the change of the uniform SST change or the change in the SST substructures?

Extreme seasons are by definition of fixed duration. The timescale is a critical difference between EMSs and MHWs, which typically last for days. We do not claim that EMSs are more pertinent to marine ecosystems than MHWs. An extreme season is just a different perspective. Please, see also our previous answer. This different perspective, our motivation and a clearer EMS definition are presented in the updated Introduction and Methods sections in the revised manuscript.

**C45.Line 104-105**: "*Therefore, even in **the** absence of extreme SST values or extreme warm events, a summer season may present extreme mean conditions thereby strongly affecting marine life.*". How can a summer present extreme mean condition if the SST is not extreme and there is no extreme warm event?Confusing.

The characterization of extreme seasons is based on their mean seasonal SST, without taking into account how SST values are distributed during the season. Extreme daily SST anomalies may be responsible for an EMS, as they increase the mean seasonal SST, but this is not necessarily the case. Elevated SSTs during the summer without the occurrence of extreme daily SST values may lead to a mean seasonal SST above the EMS detection threshold.

**C46.Line 106**: "*An EMS may arise due to elevated SST anomalies taking place over the summer duration.. period*". What do the writers consider here as summer duration? June-August? May- September? Not specified. Also elevated SST anomalies..don't already mean.."favoring of initial thermal conditions", since there is already a positive temperature anomaly?Otherwise the writers need to explain what they mean by "initial thermal conditions".Also the period examined is nowhere to be found.

Thank you for your comment. The period July-August-Sep (JAS) has been considered as marine summer season in the manuscript. This choice has been explained in Methods, and we have specified this also in the revised Introduction.

The occurrence of an EMS is potentially favored by preconditioning. In that case, the summer begins (at 01 July) with an already increased SST compared to climatology. This in turn will affect (to a certain extent, depending on the atmospheric and oceanic conditions from that day onwards) the mean seasonal SST based on which EMSs are identified. The paragraph has been revised also according to our explanations in C47.

**C47.Lines 107-110**: "*The former may result from the interplay of atmospheric and oceanic factors, being the air-sea heat exchanges (turbulent and radiative fluxes), horizontal (Ekman and geostrophic currents) and vertical (entrainment, Ekman pumping) advection (Deser et al., 2010 and standard oceanography references therein)*". There is always an

interplay between the atmosphere and the ocean, whether there is an EMS or not. Perhaps the writers meant the EMS emerged due to specific atmospheric and oceanographic conditions in the basin?Also if the former (SST anomalies) may be due to air-sea interactions, the latter (favoring initial thermal conditions) may be due to.. what?

On the air-sea interplay:
As you note, the atmospheric and oceanic processes we mention are potential drivers of any SST anomaly, not only of the elevated anomalies observed during EMSs. This part was included in the Introduction as a baseline knowledge on what generally drives SST variations, thus also the SST evolution during EMSs. Later on, we propose a new approach to quantify the role of air-sea heat fluxes in the EMS formation ($P_{EMS}$ metric presented in Methods). This approach is conceived aiming to take into consideration that the role of air-sea interplay in forming SST changes is a given. This is why this specific task focuses on the most important SST changes for the EMS formation, rather than any normally expected SST variations during a summer.

On the initial thermal conditions: Investigating what forms the initial thermal conditions (pre-conditioning), i.e., why the mixed layer before an EMS is warmer or colder than climatologically, falls out of our objectives. The term "initial thermal conditions" is now removed from the revised manuscript following your recommendation in C66 (we refer to upper ocean preconditioning instead).

The paragraph has been revised taking into account C46-48 and C66.

**C48.Lines 111-112**: "*Several studies focusing on the Mediterranean SST variability have shown tThe crucial role of **the** air-sea heat fluxes in **the** Mediterranean SST variability and observed warming trends in particular **has been shown in several studies** (e.g., Skliris et al., 2012; Shaltout and Omstedt, 2014)".* Is this paragraph devoted to the drivers of an EMS or to the motivation behind the investigation of the air-sea heat fluxes? This paragraph needs to be re-written and its content has to be more clear.
**C49.***Lines 113:"In this context, an extra focus is put within this study **aims to** on understanding what is the driving role of air-sea heat fluxes in the formation of EMS".*
**Methods:**
**C50.Line 129**: "T*his **is a** dataset corresponds to foundation SST (SST free of from diurnal variations) SST product and deriveds from a combination of **the** HadISST2 and OSTIA datasets. Atmosphere ic variables from the ERA5 product are also...*"
**C51.Line 129-130:**"*... 10-meter wind speed, net short**wave, longwave**- and long-wave radiation as well as at the sea surface, latent and sensible surface heat fluxes, total cloud cover, **and** specific humidity.*
**C52.Lines 133:** "*All fields have a grid spacing of 0.25°x0.25° in longitude and latitude and are provided in hourly time intervals.". Are you using hourly time interval in the study or not? If not, better to talk about the timestep you are using the datasets in the study.* Also better to re-write the sentence and merge it with a previous sentence like this:" *This is a free from diurnal variations SST product derived from a combination of the HadISST2 and OSTIA datasets, with a 0.25°x0.25° spatial resolution, processed in a daily/monthly/hourly timestep".*
**C53.Lines 134-136***: "In addition to ERA5 (reference SST dataset), we use the CMEMS L4*

*satellite SST product (EU Copernicus Marine Service Product, 2022c) for the period 1982-2019, at 0.05°x0.05° grid spacing, tTo cross check the quality of the reference dataset against a high-resolution observational SST dataset,* **we also use the CMEMS L4 satellite SST product (EU Copernicus Marine Service Product, 2022c) for the period 1982-2019, at 0.05°x0.05° spatial resolution".**

**C54.***Lines 135-138:* Is the CMEMS MLD different from the mixed layer thickness production extracted from the CMEMS reanalysis?Why are they mentioned twice?Confusing.

Thank you for noting this. There is no actual difference, mixed layer thickness is the name of the variable in the CMEMS product. Only the term MLD is now used in the revised manuscript, to avoid confusion.

**C55.***Line 145: "After identifying EMSs, we apply the methods of Röthlisberger et al. (2020) to assess the associated SST substructures at each..".* What is an SST substructure? Clarify.

Substructure of an extreme summer is a statistical characterization for the way daily SST values are organized over the duration of the summer. In Line 99-101 [*A marine summer…alone*] we further explain the term. Please, also see our answer in C20.

**C56.Line 154**: "*The example of Fig. 1a shows for a certain grid point that the summer of 2003 has been entirely warmer than..*". When the writers say "*entirely warmer than*" do they mean that the entire SST distribution is anomalously warm (and not only the extreme percentiles)? Then they should define somewhere in the methods that they are looking at the SST distribution moments and their anomalies instead of using words like "entirely".

Thank you for this comment. Figure 1 shows that the entire SST distribution is anomalously warm, as every single day presents higher SST than climatologically. This is observed in all sub-figs of Fig. 1. Figure 1a shows the daily SST values from 01 Jul to 30 Sep, with every single one being above the climatological daily value over 1950-2020. Fig. 1b shows the same summer days but ranked, from the coldest to the warmest. We see that every rank day is above the mean of that rank day over 1950-2020, therefore all rank day anomalies (RDA) for this summer are positive. This summer is entirely warmer than usual, but this does not imply that every part of the SST distribution contributes equally to the mean seasonal anomaly.

We explain this part in the revised manuscript as follows: "*RDAs in this example are larger for the higher compared to the lower rank days, suggesting that this summer has become extreme, in terms of mean summer SST, primarily due to the warmer summer days being warmer than usual*"

**C57.Line 155-156**: "*RDAs in this example (Fig. 1c) are much higher for the higher rank days, suggesting that this summer is identified as extreme primarily due to the warmer summer days being warmer than usual*". By looking at Fig.1c I see all the days of 2003 being anomalously higher than the average ranking between 1950-2020. All the days have ranking anomalies ranging from 1-3 C. Is there any criterion based on which you decide that the 3C ranking anomaly is the one primarily linked to the 2003 warming? I see all days having an anomalous contribution. So I am not convinced that the warming of 2003 was due to only the warming of the warmest days. The warmest days have the

highest anomalies but does that mean that the anomalies of the rest of the days did not play a role in the 2003 warming? I do not understand.

Please, see also our previous answer (C56). We do not claim that the warmest rank days are the only responsible for this EMS. But we consider that the most responsible part is the one that presents the greatest SST anomalies, thus accounting for the greatest part of the seasonal SST anomaly. This is the criterion we follow to attribute the "extremity" of a summer to different parts of the SST distribution. Please, note again here that EMSs are detected based on mean summer SSTs. Our approach to divide the SST distribution into two parts (coldest/warmest) serves to reveal which part is primarily responsible for the mean summer SST. This does not mean that there are not elevated SST values in the other part.

We strongly believe that the updated definition of EMS along with the additional information on the rank day anomalies methodology in the revised manuscript will ensure that the points you raise will not remain unclear.

**C58.Section 2.2.2:** This section should be merged with Lines 143-145 since it is part of the EMS definition. Without this part the readers get confused with the EMS definition which is currently scattered here and there.

Please, see also C7. As noted there, we agree that the EMS definition and the fact that we detect and investigate EMSs both in the original and the detrended dataset should be explained within the same section. We have merged these parts in the revised manuscript.

**C59.Figure 2 caption**: Better to give the lat/lon coordinates used to split the Med. Into the different sub-basins instead of saying "Strait of Sicily up to 22E". The science must be reproducible, therefore, the exact coordinates of the subregions should be mentioned in the captions or in the description of the Methods. Also, the legend for the detrended SST timeseries in panel (b) says "*Med SST anomaly*" which is confusing, as usually the SST anomaly refers to the SST difference from the climatology. Better to just refer to the timeseries as "detrended summer SST". The zero line does not need a legend. It is evident.

By stating *"Strait of Sicily up to 2*2°" in the legend of Fig. 2, we do not imply the separation of WMED and CMED through a straight line of constant longitude at the Sicily Strait. We mean the strait itself being a natural boundary separating the 2 sub-basins, as shown in the following figure. This is now clear in the revised document.

[Figure]

**C60.Line 171-172**: Better to decide to use either past or present tense in the study. Currently some sentences are in past and other in present which is confusing.
**C61.Lines 175-179**: This paragraph may well belong to the introduction where the writers should give a brief overview of the EMSs and their added value. However, if the writers removed the signal of the multi-decadal SST oscillations how did they also remove "*any long-term variability of the Mediterranean SST*" apart from the multi-decadal oscillation one? Did you do an additional de-trending of the SST timeseries or does this refer to the multi-decadal signal? Clarify. This is confusing for the reader.
**C62.Line 178**: "*...into the actually warmest EMSs and...*"
**C63.Line 191:** "*...Therefore, such discrepancies resulting from different climatological periods are not considered significant in the scope of the present study*". What kind of discrepancies do you mean here?Do you test different climatology periods for the detection of MHWs?Why is this information relevant for the study?

Thank you for this comment. CMEMS satellite SST was additionally used in many tasks performed to investigate EMSs, in the context of ensuring that the reanalysis produces similar results with the observations. As the common period starts from 1982, we decided to use a 30-year climatology (common approach for MHW detection) starting at 1982, allowing for inter-comparing results from ERA5 vs CMEMS. However, to examine if a different climatology would significantly alter the resulting MHW properties from the ERA5 dataset, we also tested other climatological periods. Results showed some differences in the computed metrics which were considered not significant considering the objectives of this section. Here we investigate potential changes in main MHW characteristics during extreme seasons. If we aimed to provide a thorough discussion of MHW properties within 1950-2020, testing the impact of using different climatologies would be highly relevant to that objective. We considered adequate to provide this short explanation to the reader as the sensitivity of MHW-related findings to different choices for climatological periods used in MHW detection is widely discussed within the MHW community.

**C64.Lines 192-195**: "*...In addition, the availability of satellite SST data during the selected climatological period....to detrended SST data*".All these sentences need to be re-phrased with help from a native English speaker. Also, what is the point of mentioning the

comparison between the MHWs detected using ERA5 and those using satellite if a) the comparison is not shown anywhere and b) the results from their comparison are not used anywhere else in the study?

Please, see also our previous answer in C63. CMEMS satellite SST was additionally used in many tasks (e.g., rank day anomalies methodology, MHW detection) and results were inter-compared with the ones from the ERA5 dataset for the common period. Although we considered such tests a necessary pre-processing task, we opted not to present these results as they fall out of the paper's objectives and also to avoid increasing the length of the manuscript. The point of mentioning this comparison is to report that, at least for the recent decades, the utilized reanalysis dataset is able to produce similar results with the observations. Differences fall within acceptable ranges, considering that we inter-compare modeled against observational SST. Lines 193-194 already report that results (not shown) from inter-comparison of ERA5 against CMEMS SST revealed a satisfactory level of similarity. We provide below an example of MHW detection at a location in WMED using the two datasets:

[Figure]

As regards other tasks repeated using satellite SST, C14 includes an example of applying the methodology to reveal SST substructures in EMSs based on both datasets.

**C65.Line 204**: "*To investigate the role of surface heat fluxes during EMSs, we first compute the mean anomaly of EMS Qnet and its components with respect to their mean summer value over the study period...*". What do the writers mean here by computing the "*mean anomaly of EMS Qnet with respect to the mean summer value?* When one computes an anomaly and takes its average in time, then this process goes back to the mean value from where the anomalies were created. Clarify.

Thank you for your comment. We compute EMS anomalies aiming to compare extreme summer conditions against mean summer conditions. We compute EMS anomalies for any variable in the manuscript by subtracting the mean summer climatological value of that variable from its mean value over the 4 locally detected EMSs. So, the mean EMS value is the average over the 4 EMSs

at each location. This mean EMS value represents the EMS conditions (in terms of the examined variable) at that location. It is from this value that we subtract the climatological mean (average over all summers) to obtain the EMS anomalies.

**C66.Line 217**: "*Apart from the contribution of positive SST anomalies during the season, warmer than usual initial conditions may also favor...*". It is better to avoid using "initial conditions" since the writers are talking about upper ocean preconditioning in this section. The phrase "initial conditions" usually refers to initial conditions e.g. of a climate model etc.The entire section here, in fact, talks about the contribution of the ocean heat content to the EMS development whereas the previous is simply describing the atmospheric contribution. This is a common method to investigate drivers of extreme warm temperatures in the ocean, therefore the terminology should just refer to atmosphere and ocean heat contributions (or upper ocean preconditioning).
**C67.Line 222**:"*... and the corresponding EMS anomalies (June before an EMS – mean June for all years) at each grid point..*". Better to write this as an equation instead of using words.
**C68.Line 228:** "*To overcome any unfair OHC inter-comparisons due to different integration depths..*" Rephrase.
**Results:**
**C69.Line 239:** What is the reason behind the choice of these particular 3 points whose SST substructures are investigated? Why not other points of the MS ?Is it random?Justify if not. Also, what is the period examined and represented by the grey lines? Currently we are only aware of the EMS years from the Figure legends. Caption needs to mention the entire period examined.

By including Fig. 4 we aim to provide the reader examples of SST distributions for different summers, as such information is "hidden" in the fields of Fig. 3 that include averages over the local EMSs.
The selected locations serve as examples for different patterns of SST distributions during EMSs. At the location in WMED the warmest half contributes the most to the seasonal SST anomaly in all EMSs, although the shape of the SST distribution varies significantly among the EMSs. The location in the Levantine Sea again shows a greater contribution of the warmest part to the seasonal anomaly, but SST distributions here are more similar among the local EMSs. The location in the southern-central basin shows a greater contribution of the coldest part of the SST distribution during every local EMSs.
All sub-figures refer to the entire study period 1950-2020 and gray lines stand for non-extreme summers within this period. This has been specified in the legend.

**C70.Line 241**: "*Regardless **of** the variability..*"
**Figures 3 & 4**: What is the (K) in the titles and y axis label of Figures 3a) and 4a) respectively?Also panel 4 somehow is described as a whole before panel 3 of figures. I would suggest moving Figures 4 as Figures 3 and then describe the maps currently named as Figure 3, since anyway their description is now interrupted from the description of Figures 4.
**C71.Figure 4.** The caption of the figure is missing the reference to plots d,e,f. They are only referred to as bottom row. They should be referenced together with the top row. e.g. a,d) southern of the Gulf of Lions (…) etc.

**C72.Lines 266-267**: *"To gain insight into the SST substructures over the basin, Figs 3b and 3c show the fractional contributions of the warmest and the coldest half of the RDA to the formation of EMSs"*. What do you mean by *"warmest and coldest half"*? Which ranks are the warmest and coldest ones out of the distributions shown here? Not clear.This is important in order to also understand the contribution maps of Figure 3. Also, what is the contribution of the middle ranks? The ones which are not too cold or warm?And how is the contribution calculated?Relative to what quantity? This is not clear.
- The description of Figures 3d-f comes later in Section 3.1.3 and after the description of Figures 4 & 5 which is confusing for the reader. It seems ad-hoc and the reader has to scroll up and down the manuscript. The description of the figures should be done sequentially. So section 3.1.2 should come after section 3.1.3

On the SST substructures: Please, see our relevant answer in C4. As clarified therein, the warmest and coldest half of the RDA distribution range from the 1$^{st}$ (i.e., the coldest) to the 46$^{th}$ rank day and from the 47$^{th}$ to the 92$^{nd}$ (i.e., the warmest) rank day, respectively. On the middle part, please see our answer in C14.
On the description of figures: While merging the 3.1.1, 3.1.2 and 3.1.3 sections (in a single 3.1) following your suggestion, we also rearranged the presentation of results. In the way results are now structured within the revised Sect. 3.1, the figures are described sequentially. We believe that this change has really improved the text the document's flow.

**C73.Line 275**: If the variance of the RDA is calculated following the standard mathematical equation, Var = (RDA-mean RDA)^2/(number of summers). Are the writers sure that this quantity applied to the anomalous filed does not bring the Variance back to the mean? And probably this is why the maps of Fig5 resemble a lot the mas of Figure 3?Why did the writers choose the variance and not the standard deviation?

Thank you for your comment. By definition, the mean of the RDAs equals to zero. Therefore, the variance of RDAs computed for the 71 years reduces to the following:

$$Var = \frac{1}{71}\sum_{s=1}^{71}\frac{1}{92}\sum_{d=1}^{92}(RDA_{d,s})^2$$

which is an adequate measure of variability for RDAs, representing their spread. The fractional contribution from the coldest/warmest part of all summer days to Var are computed as follows:

$$VContr_{coldest} = \left(\frac{1}{71}\sum_{s=1}^{71}\frac{1}{92}\sum_{d=1}^{46}(RDA_{d,s})^2\right)/Var$$

$$VContr_{warmest} = \left(\frac{1}{71}\sum_{s=1}^{71}\frac{1}{92}\sum_{d=47}^{92}(RDA_{d,s})^2\right)/Var$$

The same statistical approach has been employed by Röthlisberger et al. (2020) to interpret the substructures of extreme (atmospheric) summers in relation to climatological variability.
On the similarity of the fields Fig. 5 vs Fig. 3, please see our answer in C77.

The above equations are now included in Methods, and references to them are provided within the text, where needed.

**C74.Lines 279-280**: "*Results come in agreement with Shaltout and Omstedt (2014) who that examined the SST variability...*"
**C75.Lines 279-281**: "*Results come in agreement with Shaltout and Omstedt (2014) who examined the SST variability in the basin showing that the maximum and minimum seasonal stability are found close to the southern Levantine sub-basin and the Gulf of Lions, respectively.*". This type of comments belong to the discussion and not in the result section.
**C76.Lines 282-283**: *"We then calculate the fractional contribution of the coldest and the warmest half of the ranked summer days to the RDA variance for all 71 summers (Figs 5b,c)"*. Again which ones are the warmest and coldest half of ranked summer days, you need to name then. Otherwise it is vague. Also how do you calculate this contribution? Not clear.

Please, see our answer in C73.

**C77.Lines 286-287**: "*In other words, the locations where the warmest (coldest) part of the rank day distribution has higher spread, experience EMSs primarily due to the contribution of the warmest (coldest) part of the EMS rank day distribution*". In Figure 5a) I only see higher spread in northwest Med. Sea where there is also a high contribution from the warmest and coldest half of rank days (from Figure3). However there is also a high contribution from the warmest and coldest half or rank days in e.g. the north Aegean, where the spread (Fig5a) does not seem to be high. How do the writers explain this? Is their statement here true for all the areas of the Med. Sea?

Thank you for your comment. The sentence "..*the locations where the warmest (coldest) part of the rank day distribution has higher spread..*" does not refer to the seasonal variance presented in Fig.5a. It refers to the contribution from a specific part (warmest/coldest) of the distribution to the seasonal variance (total variance of summer SST within 1950-2020). The spread of each part expressed as a percentage of the seasonal spread is presented in Figs 5b,c. In both areas you mention (NW Med with the highest and the Aegean with a much lower spread), the warmest part shows higher spread compared to the coldest (Figs 5b,c), not compared to other locations. What we say is that the part presenting the greatest spread over the years at a particular location, is the same part that is the most responsible for the seasonal SST anomaly in EMSs at that location. This is how we interpret the similarity between the fields in Figs 3b,c that refer to EMSs, with Figs 5b,c that refer to all summers (same for Figs 3e,f vs Fig5e,f using detrended data).

**C78.Lines 290**: "*At the location at 41.5° N-5° E (Fig. 4a), SST substructures appear particularly different among the EMSs.*".Although this is true I can also discern other years (grey lines) with the "warmest" part of the SST distribution contributing more to the summer temperatures (following the criteria of high values just like the writers). So what is the difference in this contribution with the contribution of the EMS RDA?

Thank you for this comment. The contribution from the two parts to the seasonal anomaly can be computed for any summer. In this work however we detect and explore the warmest summers in the basin. This does not mean that there are no similarities with the rest of the summers as regards patterns of the SST distribution. The non-extreme years are shown here also to illustrate the similarity you observed and that was explained in our previous answer in C78: that the part of the distribution that contributes the most to the EMS anomaly is the part that presents the largest spread climatologically. Please, see also specific clarifications in your following comments.

**C79.Lines 292**: *"This is the part presenting the largest RDA spread in the rest of the years as well (6.7° C and 5.4° C for the warmest and the coldest part, respectively) (Fig. 4a)"*. Which years? The non EMS or the 3 other EMSs?Not clear. Also, what is the spread? Can you give the variance value?

Thank you for this comment. The rest of the years in the context of this sentence are the non-EMS years (gray lines). At this location (Gulf of Lions), the warmest part explains 58.8% of the seasonal variance (i.e., of the summer SST variance computed over 1950-2020). Instead of reporting ranges (maximum - minimum values) of RDAs in the 3 examples, we have reported in the revised manuscript numerical values for the contributions from the two parts to the total variance. This has been done to improve consistency, as variance is the measure of spread presented in the manuscript.

**C80.Line 293**: "*Similarly, at the location at 35°N-28°E (Fig. 4b), the highest rank days which contribute the most to the mean EMS RDA are also the most varying from summer to summer within the study period.*". There is no location 35° N-28° E indicated in Fig4b. The point 35° N-28° E is located in the central Levantine basin, which according to Fig5b) is characterised by an increased contribution from the high rank days. However, by looking at Fig5a) I do not see a particularly large spread in the area. In fact,the spread (variance) is much smaller than the one shown for the northwest basin.

In case your first comment concerns the reason why we cited Fig. 4b here, we agree that this should better move to the end of the sentence.
For the discussed location in the Levantine Sea, Fig. 5b shows an increased contribution from the higher (compared to the lower) rank days to the seasonal variance (55.8% and 44.2%, respectively). The variance is lower at this location (compared to others, like in NW basin), but this is not what we examine here. We do not compare the variance between different locations. Instead, we juxtapose contributions from the two parts a) to the EMS SST anomalies and b) to the seasonal variance (variance of summer SST within the entire study period). Please, see our clarifications in C77.

**C81.Line 295**: "*The RDA range of the warmest part here is 3.3° C (2.7° C for the coldest part)*". When "range" is used then two numbers should be given that represent a *range* of what? What do the writers mean something else here. It is confusing.

Please, see our answer in C79. We have shown the range of ranked daily values within each part of the distribution, in terms of maximum - minimum value. This has been replaced with numerical

values for the contributions from the two parts to the seasonal variance, to be consistent within the text.

**C82.Lines 296-297**: "In contrast, at the location 34.5° N-13° E (Fig. 4c), which exhibits greater RDA variability in the first rank days of the RDA distribution, RDA values of the coldest half display a range of 4.6° C (3.5° C for the warmest part)". Looking at Fig 4c) I see the RDA ranging from 1K – 3K (is this Kelvin or something else?) for the coldest half of the rank days. So where does the "range of 4.6 ° C" come from?Unless the writers mean something else.

Please, see our clarifications in C79.

**C83.Line 299**: "...*the part of the RDA distribution that contributes the most to the EMS RDA is the one presenting the largest spread climatologically*". Where is this climatological spread shown exactly? Not clear.

Please, see our clarifications in C77 and C80. Figures 5b,c are the ones that show the contribution from each part to the total variance (computed for all summers within 1950-2020). This is the spread of each part expressed as a percentage of the seasonal spread (i.e., of the seasonal variance shown in Fig 5.a). When we say "*the one presenting the largest spread*" we mean the spread of one part compared to the other part at a particular location, not compared to other locations.

**C84.Line 300**: "*When the multi-decadal trend is removed, smaller SST anomalies in EMSs are found in the entire MS. (Figs 3a,d)*". The anomalies are smaller relative to the climatology but also in comparison with the non-detrended dataset.

Thank you for this comment. The computed SST anomalies are, by construction, relative to a climatology (either in the original or in the detrended dataset). Figs 3a,d show that the deviation of extreme summer SST from mean summer SST is smaller in the detrended compared to the original dataset. We have added "*with respect to the original dataset*" in the revised manuscript to avoid any confusion.

**C85.Line 306**: "*In both fields, RDA peaks are found in the Gulf of Lions and the Ligurian Sea, followed by the Adriatic Sea.*" The Results section is where specific numbers should be given. Consider mentioning the the RDA peak numbers you are referring to here. In general a range of values that represent big or small differences in the RDA between the left and right column of Fig3 should be given in the Result section.
**C86.Lines 313**:"*In particular, in the detrended dataset, locations where the coldest rank days contribute to the RDA variability more than 50% are not restricted in the southern-central area north of the African coasts as in the original dataset. It is also the central Adriatic, part of the southern Levantine and the north Aegean Seas that exhibit a slightly enhanced contribution of the cool summer days to the RDA variability (Figs 5e,f).*".
Rephrase the sentence and give more representative numbers. The result section is where values should be given, whereas the discussion section is used for commenting on the differences between the numbers of the results section.
**C87.Lines 320**: "To better illustrate this, the SST substructures in the three example cases

discussed above are very similar for the original and the detrended dataset, despite the actual EMSs being warmer (Fig. 4a-c and Fig. 4d-f, respectively)". The actual EMSs being warmer...than what? Not clear.

Thank you for this comment. Here we compare the actual with the detrended dataset. In the quoted sentence we mean the actual EMSs being warmer than the ones identified using the detrended dataset. We have specified this in the sentence.

**C88.Lines 320-324**: Re-phrase with help from a native speaker.
**C89.Lines 326-329**: "In addition, SST substructures in EMSs seem to be independent of the selected study period. Sensitivity tests performed for different sub-periods show a consistent statistical behavior of the detrended SST dataset compared to the original one (not shown).". What kind of consistent statistical behavior do the writers mean here? Not clear.

Thank you for this comment. In the previous paragraph (Lines 319-324) we discuss the similarity of the revealed SST substructures between the original and the detrended dataset. These next lines (quoted lines 326-329) aim to report that repeating this analysis based on different study periods results in similar spatial patterns of SST substructures with the analysis based on the entire study period, thus also similar between the two datasets.

**C90.Line 331**: "...EMSs are formed based on a "background: SST substructure field, largely depending on the climatological ranked daily SST variability in the MS". I thought the SST substructures where represented through the RDA and not the climatological ranking of the daily SST. This sentence is confusing.

Thank you for this comment. SST substructures are indeed represented through RDAs. RDA is the SST anomaly of a rank day relative to the mean climatological value of that rank day. For example, the coldest day of summer 2003 is the $1^{st}$ rank day of this summer. Its RDA is computed by removing from the SST of that day, the mean SST of the $1^{st}$ rank day of all summers. The sentence in Line 331 combines results from the original and detrended dataset. In this context, the word climatological is used to highlight the role of a baseline (climatological) pattern on top of which the multi-decadal signal makes the extreme seasons even warmer.

**C91.Line 334**: "*Analyzing the EMS SST substructures in the previous section revealed that EMSs in the basin are commonly formed due to the warmer summer days being warmer than normal*". This is not true always, since the example case of the Central Med. Sea (Fig.4c) shows that the EMS develop mostly due to the..coldest rank of days becoming Warmer.

Thank you for noting this. You are right, the word "commonly" does not necessarily imply that it is not true basinwide. We will remind the main exception of this pattern in this sentence.

**C92.Line 336**: "*To complement these findings, this section investigates the role of MHW events during EMSs.*". Why does the role of MHWs acts as complementary to the EMS? Are the writers sure that there are EMSs without a MHW occurring?

Please, see our answer in C8.

**C93.Line 336-337**: Re-phrase the sentences.
**C94.Line 337-338**: "*Then, we examine the relative role of MHWs in EMSs by means of changes in MHW properties during EMSs (Sect. 3.2.2) with respect to mean MHW conditions*". Do the writers mean the non-EMS conditions when referring to "mean MHW conditions"?. Not clear.

Thank you for this question. No, we mean the mean summer MHW conditions. Please, see our explanation in C15 and C100.

**C95.Lines 359 – 371**: These paragraphs belongs to a discussion section and not to the Results.
**C96.Lines 269-371**: "*This difference is mostly related to our choice to focus on summer MHWs that begin and decay within the JAS summer period. Taking into account all events throughout the year, similar MHW duration values are reproduced (not shown)*".Do the writers mean that duration of the MHWs would change if the examined period would be between June – August or any other season of the year? What are the similarities of the MHW duration when taking into account all the events of the year? Similarities with the MHW duration of the current study or with previous studies?Not clear.

Thank you for this comment. In lines 367-68 we write "*However, all aforementioned studies report larger MHW durations in the south-eastern MS compared to our results*". These studies consider events over the entire year. We checked that our metrics change when taking into account all events of the year and the resulting mean duration fields in the basin are then similar with literature. We have clarified in the revised text that here we refer to previous studies.

**C97.Line 374-376**: "*Consequently, they represent the mean seasonal conditions of i) all summer MHWs in the study period and ii) the MHWs occurring in EMS, respectively, both not expected to be indicative of the most extreme MHW conditions.*".What do the writers mean here by "*both not expected to be indicative of..*"? The mean seasonal conditions are not expected to be indicative of MHW conditions. However the EMS are summers with extreme warm conditions. So why are they not expected to be indicative of MHW conditions?

Thank you for your comment. We write that they are not expected to be "*indicative of the most extreme MHW conditions*", not indicative of MHW conditions. This is because we average MHW metrics, we do not detect MHWs and pick the most extreme ones. Summer MHW conditions are the mean MHW conditions averaged over the 71 summers of the study period. Analogously, EMS MHW conditions are the mean MHW conditions averaged over the 4 EMSs. Moreover, MHWs are detected in this work based on the $90^{th}$ percentile threshold which is relatively easy to exceed and thus not expected to capture only the most extreme events.

**C98.Lines 377-378**: "*MHWs may promote the formation of an EMS either by being more frequent, of greater duration or of greater intensity than usual, thus, contributing in*

*different ways –not necessarily bearing extreme characteristics– with positive SST anomalies to the seasonal SST being the focus of this study*". I am not sure the difference between the EMS and the MHWs is properly explained so far in order to understand why the MHWs *may promote EMS formation* instead of being part of/always present when an EMS is happening or how the EMS conditions are differentiated from MHW conditions, or why an EMS cannot promote a MHW instead.

Please, see our answer in C8.

**Section 3.2.2**
**C99.Line 380**: "*To understand the role of MHWs in EMSs, this section presents the mean EMS anomalies of MHW properties with respect to their mean summer values.*". The way this sentence is written is confusing for the reader. Does that mean that this section presents the EMS anomalies of MHW properties relative to the mean EMS summer values or relative to the mean summer MHW values?

Thank you for your comment. We compare each MHW property between extreme summers VS climatological summers. Therefore, the EMS anomaly of a MHW property is relative to the mean summer value of that MHW property (mean value of the property over the 71 summers).

**C100.Figure 6**: Does the computation of the mean MHW properties during 1950-2020 include the EMS years in the average or not? Because if the EMS years are included in this average what is the point in creating the anomalies of MHW properties only during EMS years (Figure 6d,e,f)? Clarify.

Thank you for this comment. EMS anomalies for any variable in the manuscript are computed by subtracting the mean climatological value of the variable from its mean value over the 4 locally detected EMSs (EMSs – All). The point is to compare extreme conditions against mean conditions in the study period. By excluding the EMS-years when computing the climatological mean value, we would certainly obtain higher SST anomalies for EMSs, as they would have been computed relative to colder average summer conditions. We believe both approaches are acceptable, but we chose to include all summers as in this way the reference period corresponds to the actually average conditions.

**C101.Line 385**: "*In terms of mean basin values, mean summer MHW intensity equals 1.56° C while when isolating events during EMSs, it drops to 1.45° C (Table 1). This suggests either the suppression of MHWs in EMSs or the existence of alternative mechanisms potentially enhancing the MHW conditions in EMSs.*". Table 1 shows a basin-mean Imean of 1.41 ° C instead of 1.45° C. I am not sure I understand how the reduction of Imean during EMS-MHW conditions can mean either a suppresion or enhancement of MHW conditions. What do the writers mean here?What kind of mechanism are they referring to and how this differentiates when there is MHW suppresion from MHW enhancement?

Thank you for this comment.

The second row of the Table corresponds to results from the detrended dataset, noted here as EMD (standing for Empirical Mode Decomposition that was applied for detrending). We understand now that this has obviously been confusing, as it highly resembles EMS.

In this part we compare basin-mean values of MHW intensity for MHWs occurring within EMSs (3rd column) against MHW intensity for all summer MHWs (2nd column). As we find a smaller basin-mean intensity in the case of MHWs occurring within EMSs, compared to mean summer MHW conditions, we assume that this is either due to generally weaker MHW conditions during EMSs or that it is only the intensity of events that presents the observed reduction during EMSs. In fact, we see that frequency and duration are increased during EMSs compared to normal conditions, which supports the latter assumption. The anomaly maps of Fig. 6 show that it is over the southern Med. Sea areas where we mainly see the negative intensity anomalies in EMSs (resulting in the reduction of Imean presented in Table 1). In the same areas though, the number of events per summer and their duration are greater in EMSs than usual.

Regardless of the above explanation, we have reformulated this small part of the MHW section in order to remove any discussion of basin-mean values. This is suggested following a relevant comment of Reviewer 1 on the interpretation of basin-mean MHW metrics. We acknowledge that such values do not present new information on top of the maps in Fig. 6. For this reason, we have removed Table 1 and the few references to this Table from the MHW section.

**C102.Lines 400-401**: "*MHW analysis of this part further suggests that enhanced MHW conditions commonly contribute to the observed SST anomalies in EMSs*".. I am not sure that the comparisons of Fig.6 shows this result (see comment of this Section in Major comments). But even if this is true, I do not understand why the MHWs contribute to the EMS conditions instead of being the reason why an EMS is formed in the first place? Also in Lines 407-408 the writers mention:"This suggests that in this location, the higher rank days which are the most responsible for the EMS formation (Fig. 3b), face more difficulty in producing distinct MHWs". Does the EMS create MHWs or the other way around in the end? These two sentences are a bit contradictory.

Thank you for this comment. First, please see also our clarifications on the EMS definition and differences with MHWs in C3 and C8, respectively. MHWs contribute to EMSs (and not the other way around) in the sense that the mean seasonal SST value is formed according to the SST variations throughout its duration. Analogously, a part of the rank day distribution, e.g., the highest (warmest) rank days, may be more responsible than the other part of the distribution for the elevated SST anomalies during a season. In this sense, attributing an EMS to specific parts of SST distributions, or to MHW occurrences comes from the smaller time scales of the latter compared to an entire season (rather than from a direct cause-and-effect perspective).

**C103.Lines 399-415**: These two paragraphs are discussing the same thing and should be merged in one. Either more numbers of the indicative duration and intensities should be given here or move this sentences as part of the discussion.
**C104.Section 3.2.3**:
Although this is a Results section the description of MHW properties gives no numbers. Rather it is limited in comparisons of "greater" or "smaller" magnitude. Either move

these paragraphs to the discussion section offering possible mechanisms behind the results described or simply refer to the differences seen in MHW properties with the detrended datasets using numbers.

**C105.Lines 424**: Apart from the multi-decadal signal being removed is the trend value of the summer also removed here like in lines 170 – 173? Not clear.This information was never mentioned again in the manuscript after the beginning of the paper.

Thank you for your comment. Please, see our relevant answer in C5.

**C106.Lines 426 – 42**7:"*This suggests the presence of similar event-driving mechanisms and reveals the positive contribution of the multi-decadal signal to the observed sea surface warming, this time specifically via the SST anomalies caused by MHWs.*". How exactly has the positive contribution of the multi-decadal signal to the observed SST warming and (especially through the MHWs) been revealed here if the mechanisms have not changed at all in this and the original dataset? I do not understand. Clarify

When removing the multi-decadal variability the magnitude of MHW intensity reduces but the intensity spatial distribution is remarkably similar between the original and the detrended dataset. These findings may be seen in the figs below (left: using the Original dataset, as in the manuscript; right: using the detrended dataset). These results suggest that the warmer events in the original (compared to the detrended dataset) result from the warming signal included in multi-decadal variability. For this reason we say that the multi-decadal trend contributes to more intense events. The striking similarity in the intensity fields most probably suggests that events are regulated by the same mechanisms, apart from the role of the discussed trend that is related to longer timescales. However, as this has not been investigated and supported in this paper, we have removed the part *"suggests the presence of similar event-driving mechanisms"* from this sentence.

[Figure]

[Figure]

**C107.Lines 432-437**: These are arguments that belong to a discussion section and not to the results as they are commenting on literature and possible explanations.

Please, see our previous answers in C10.

*Line 432-433:* "During EMSs, MHW intensification is greater when using the original dataset. This is more pronounced in the eastern basin and particularly in the Aegean Sea". Apart from Table 1, Fig. 6 should be cited here as well.
**C108.Table 1:** The categorization in "Orig" and "EMS" is not a good choice of names. Better to find some other names to distinguish between the two datasets used here.

Thank you for your comment. Please, see our answer in C101.

C109.Line 439: "*Moreover, we observe a similar behavior of MHW properties in the detrended compared to the original dataset*" I thought the writers were already talking about the comparison between the detrended and original dataset. So why repeat the argument Here?

Thank you for noting this. Yes, we were already talking about the comparison with the detrended dataset. This sentence refers to a specific behavior, described right next: *longer lasting events in EMSs (compared to the mean state) counterbalance the lower event intensity values in EMS*. However, we totally agree that the way this sentence begins may be confusing for the reader. This will be corrected in the revised manuscript.

C110.Lines 440: "'...*longer lasting events in EMSs (compared to the mean state) counterbalance the lower event intensity values in EMS (Table 1).*"____________This sentence is really
unclear. I am not sure where to look in Table 1. Are you comparing the MHW days between the mean summer MHWs and the EMSs MHWs on the Orig dataset or the days between mean summer MHWs and the EMSs MHWs or the duration of events between the Orig and EMD datasets? Clarify. Better to rename the datasets of Table 1 and everytime you compare between them, refer to what datasets you compare. And also, how do you know that longer lasting events are "counterbalancing" the lower intensity? What do you mean by this? Not clear.
Thank you for this comment. Please, see our answer in C111.
C111.Lines 442-446: This paragraph refers to which Figure exactly? Not really sure where to look for these results.Is it if Figure 6? Very confusing.

Thank you for this comment. As MHW results based on detrended data are to a great extent similar to the original, we opted not to include them in the manuscript. We note in the manuscript that these specific results are not shown. The relevant figure, being the equivalent to Fig. 6 of the manuscript but based on detrended data, is provided below:

[Figure]

We have replaced this short section (3.2.3) with a very short paragraph reporting that MHW analysis based on detrended data yielded similar findings and briefly noting differences with the

original dataset, without providing any discussion. As the MHW Sects 3.2.1 and 3.2.2 have been merged following your suggestion in C12, this short paragraph has been incorporated into that analysis. As previously explained, the Table 1 has been removed.

**C112.Line 442**:" *...in absence of long term trends….*". Why do you need to refer to the absent trends and what do you mean by that? That there are some trends that you have not shown or that there are no trends in the particular areas you are referring to? Not clear.

Thank you for this comment. We refer to the use of the detrended dataset, which is free of long term trends. The specific areas we mention here are those exhibiting positive EMS anomalies for all three examined MHW metrics (see draft figure above). This suggests an enhanced role of MHWs during EMSs in these areas, regardless of long term trends.

**C113.Lines 444 – 445**: "*Among these regions, the Aegean Sea stands out as the one presenting a pronounced MHW contribution to EMSs both in the original and the detrended data...*". Again where am I supposed to look for these results? Is it figure 6 or not? Figure 6 does not include comparison between the detrended and the original dataset, no? These statemens here look like they come from nowhere since there is no reference to any figure inside the paper or any other result.

Thank you for this comment. Please, see our previous answer in C111.

**C114.Lines 446-447**: "*...Given that, the pronounced role of MHWs in EMSs in the eastern basin can be only partly attributed to the sea surface warming trend in the MS during the past few decades.*". How do the writers know that the pronounced role of MHWs in the MES in the eastern basin is only "partly" attributed to SST warming? Where do they show that it is "only partly" attributed? Have they compared any other variable such as trends of the MLD, winds?(e.g. trends of lower winds in the area or trends of shallower than normal MLD). I find these arguments a bit unfounded since there are no results to look at and compare at this point.

Thank you for this comment. Please, see our previous answer in C111.
To clarify our initial intent: By comparing MHW anomalies for EMSs between the original and the detrended dataset (Fig. 6 vs figure provided above), we see that the Aegean Sea presents positive EMS anomalies for all examined MHW properties in both cases. In other words, the Aegean Sea appears to present enhanced MHW conditions during extreme summers (compared to climatological summers) independently of long term trends. However, the positive EMS anomalies in this area are slightly larger in the case of the original data. This potentially suggests that the long term trends, including the warming trend in the area during the past few decades, may be to a certain extent responsible for this finding.
However, as noted in C11, we have removed such discussion, as it is not properly supported without providing additional figures.

**C115.Section 3.3**
The entire section here describes results on air-sea heat fluxes without giving any indicative numbers on the increased or decreased contributions presented in Fig.7. The

description here is something between a Results section and a Discussion. The entire section should be re-written where some indicative numbers of the spatial distribution of the air-sea heat fluxes should be given, separately as Results and then in the Discussion section there should be a comparison with current knowledge and literature.

Please, see our previous answers in C10 and C14.

**C116.Lines 449-458**: Is this analysis only for the original dataset and the detrended one is only shortly discussed in Section 3.3.5?Clarify in this paragraph.

Yes, this analysis is only for the original dataset. This will be clarified.

**C117.**Lines 465 – 467:" *Moreover, positive Qnet anomalies appear mainly in areas where enhanced MHW conditions in EMSs were detected (northern Mediterranean regions and particularly the Aegean and the Adriatic Seas; Fig. 7A vs Figs 4d-f)*". I am not sure what is the point behind the connection between Fig.7a maps with the results from Fig.4d-f that refer to points. After all, the points that were (randomly?) selected from 3 regions (Fig.4) cannot represent the EMS profile of the entire area (from where they came for). In addition, we do not know if the EMSs of other points, e,g in the south-central Med. Sea exhibit the same behaviour as those in Fig.4,but are in areas characterised by negative Qnet. So I am not sure what is the argument behind this sentence. Is a cause-effect result implied here?

Thank you very much for noting this, it is a typo in the reference. Instead of "Fig. 7a vs Figs 4d-f" we should have written Fig. 7a vs Figs **6**d-f. Our apologies as this obviously made you struggle to make sense. The similarity in spatial distribution of positive anomalies in the fields of the correct figures is noticeable.

**C118.Lines 468-469**: "This similarity is at least partly attributable to the widely explored driving role of air-sea heat fluxes in the formation of MHWs". How can a similarity between a map and 4 points (?) be attributed to "a widely explored role or air-sea heat fluxes"? What exactly is the similarity here and how is it related to the fact that we know a lot about air-sea heat fluxes? I do not understand the argument here. Clarify. Also, the studies cited here are global studies examining drivers of MHWs at a local scale. Better to cite studies that refer to the role of air-sea heat fluxes in Mediterranean MHWs and not on global events.

Please, see our previous answer in C118.

**C119.Line 475**: Either 'above-mentioned' or 'aforementioned'.Replace.
**C120.Lines 485 – 487**: This is a far-fetched claim to be made. Although there seems, indeed,to be a slight reduction of MLD across the Turkish coasts, I am not convinced that the reduced upwelling can be inferred solely from an MLD indicator. Especially because the MLD refers to extreme EMS summer which, at the moment, I am not quite sure which period exactly they refer to and if this period coincides with the exact period of the upwelling in the area. Also, the anomalies of the MLD apparently are computed with respect to a different baseline period, which cannot perhaps be directly compared with

the anomalies of the rest of the variables of Fig.7, let alone to be linked with the impacts of EMS. In addition, is this reduction of MLD significant enough to suggest there is going to be reduced upwelling? Either compute trends of the upwelling index in the area or the authors should entirely remove this claim from the paper as it is not substantiated.

Thank you for this comment. MLD anomalies are computed for EMSs as for all variables (mean-EMS value minus mean summer value). For information on the period that EMSs refer to and how they are defined, please refer to C3. As regards the reference period for the computation of MLD anomalies, please refer to C17 where we justify our choice. We agree that the reduced MLD in the discussed area is insufficient evidence for reduced upwelling. Even if these lines express a hypothesis, they do not offer substantial information and we have entirely remove them from the manuscript.

**C121.Lines 487 – 488**; "SH flux (Fig. 7d) and net SWR anomalies (Fig. 7c) in the eastern Aegean and the central Levantine Seas are negative but of a much smaller magnitude.". How much is the magnitude here? Quantify this information wherever possible.

We have revised this section accordingly. Please, see our answer in C16.

**C122.Figure 7.** In the text shortwave radiation is referred to as SWR, whereas in Figure.7 as SW. Be consistent on the naming of variables everywhere.
**C123.Lines 502 – 50**6: How can the Qnet anomalies in the Adriatic Sea be "*almost fully determined*" by shortwave radiation (line 503) when there is also a reduction of LH losses according to Fig.7b? I think the LH also plays a significant role here and it is not just the SWR.

Thank you for your comment. Negative LH flux anomalies indeed indicate a reduction of the LH flux in EMSs. Given however that we do not use absolute numerical values for fluxes, the negative LH anomalies suggest smaller negative real values of LH flux, i.e., greater heat loss. A relevant note exists in the legend of Fig. 7 to avoid confusions. In general, negative heat flux anomalies correspond to either a reduced heat gain or increased heat loss from the sea surface (always relative to normal heat exchange).

Please, see also our previous answer in C16. We show therein why the additional heat gain (in EMSs compared to normal conditions) that we see in most of the Adriatic is attributed to SWR anomalies. We also explain this in the following comment.

**C124.Line 504-505**: "*The increased LH loss in EMSs here is exceeded by the gain in net SWR, leading to the observed net heat gain*". This sentence does not make sense at all. I do not understand what the writers want to say here. Also, I do not see anywhere an increase or LH losses. I rather observe a reduction of LH losses.

Please, see our previous answer and the answer in C16.

This sentence refers to the Adriatic Sea where we generally observe: positive SWR anomalies, negative LH anomalies and positive Qnet anomalies in EMSs, relative to their mean climatological

values. Please note that downwards fluxes have been considered as positive. Positive anomalies (red color) therefore indicate either a greater heat gain than normal or smaller heat losses than normal. The negative LH flux anomalies here indicate greater LH losses than usual.

**C125.Lines 514 – 515**: "*MLD and wind speed anomalies tend to present the same sign, especially where large anomalies appear*.". Although this is true for some regions of the basin, I can see also other areas (e.g. southeast Levantine) where the windspeed is lower than normal and MLD is higher than normal. Again, at the moment there cannot be a direct comparison between MLD and any other variable of Fig.7 given the fact that the anomalous fields are computed based on different reference periods.

Please, see our relevant answer in C17.

**C126.Lines 509 – 520**. This paragraph again is a mix between Results (without any indicative numbers) and a Discussion which does not reveal anything novel. The writers try to claim a relationship between wind and MLD (which is already well-known), albeit on anomalous fields that are computed based on different periods. Again, this paragraph should be rewritten separately for the Results section, where indicative numbers are given and the comments should be addressed in the Discussion section.

Thank you for this comment. It is important to clarify that we present and discuss findings for extreme summers in relation to climatology, we do not relate wind and MLD in general. Despite the existing literature on this relationship, we considered adequate to examine the seasonal anomalies of MLD and wind speed during the studied EMSs, and shortly discuss them in relation to other variables. As regards the different reference periods, please see our justification in C17.

**C127.Lines 531-532**: "*These fields suggest that LWR and SWR (standing for net values from now on) in EMSs work complementarily in the basin.*" What do you mean by complementarily? I do not understand.

Thank you for noting this. SWR and LWR anomalies work inversely in many areas presenting opposite signs and this is described in the following paragraph in the manuscript. Therefore "complementarily" may be confusing, potentially suggesting the opposite meaning. This part has been revised accordingly.

**C128.Lines 536**: The Figures of the supplementary material better be named as e.g. Fig.**S**3a instead of Fig.**B**3a. What does "**B**" stand for? I have never seen this in the literature so far. Also from Fig.B3a, I can see positive anomalies of cloud coverage in the central Aegean but also negative cloud coverage to the southern-southeast Levantine, where the LWR anomalies are close to zero. So the statement "Indeed, positive EMS anomalies of total cloud cover derived from ERA5 are found in this area (Fig. B3a) leading to increased downwards LWR and the observed reduced net LWR loss from the sea surface (Fig.7e)". Is not entirely true for the Levantine basin.

According to the manuscript composition guidelines *"Appendices should be labelled with capital letters: Appendix A, Appendix B etc. Equations, figures and tables should be numbered as (A1), Fig. B5 or Table C6, respectively. "*

On your comment on Fig. B3a: We agree this is true for a large portion of the Aegean and Levantine Seas but not entirely true for the Levantine basin. We see that moving towards the southern-southeast Levantine the cloud coverage anomalies become negative, SWR anomalies become positive and LWR anomalies approach zero values, suggesting clear sky conditions during EMSs in these areas (from top to bottom, you can see gathered below Fig. B3a, Fig. 7c and Fig. 7e). This has been specified in the revised manuscript.

[Figure]

**C129.Line 538-539**: "Particularly in parts of the Adriatic and the northern Ionian Seas, the SWR surplus is responsible for the net heat gain observed in EMSs (Fig. 7a vs Figs 7b-e)". Since there is Section 3.3.3, where the contribution of each air-sea heat flux is quantified, better to combine these two section together. There is no point in describing qualitatively the air-sea heat contributions here, without giving any quantifiable information and then create a new section later (3.3.3) where the contributions are actually computed. Also, by looking the maps of Fig.7 I cannot deduce that it is only SWR surplus that is responsible for the net heat gain of the EMSs in the Adriatic and the Ionian Sea. I would argue that reduced LH losses contribute as well.

Please, see also our previous answer in C16. We show therein why the additional heat gain (in EMSs compared to normal conditions) that we see in most of the Adriatic is attributed to SWR anomalies.

**C130.Line 541**: "*Results suggest that Qnet anomalies (either positive or negative) in EMSs are primarily formed by LH fluxes in most areas, followed by SWR in the western and central basin and particularly in the Adriatic Sea.*". This contradicts the sentence just above, where SWR is responsible for the net heat gain!

Thank you for noting this. We agree that the word "followed" here may be misleading. Our intention was to report that SWR is the second most important component in forming Qnet basinwide (in terms of spatial extent), not the second most important in the Adriatic Sea (where it is the most important). This has been corrected in the revised text.

**Section 3.3.2:**
As before, this section appears more as a Discussion section rather than a results section as no quantification is provided. This description (if needed) needs to be transferred to a discussion section.
**C131.Line 550-552**:"*Differences in the behavior of surface heat fluxes are expected to exist among the locally detected EMSs and the mean EMS state presented in the previous Section. To shed light on this, Fig. 8 illustrates how common is a positive anomaly for each of the examined heat flux components during EMSs*". I do not see how the frequency of positive anomalies occurrence at each grid point, constitutes "*a mean EMS state*" while the analysis performed in the previous sections is on "*locally detected EMSs*". Since both analysis yield maps that means that both fields refer to locally detected properties. I suggest to change the title of the entire section here.

Thank you for this comment. We do not claim that the occurrence frequency of positive anomalies at a particular location constitutes a mean EMS state. The provided frequency number, ranging from 0/4 up to 4/4, serves to inform on how the mean EMS values (i.e., values averaged over the 4 EMSs) are formed. For example, if SWR anomalies in the Adriatic would present positive values in only one EMS, being however high enough to lead to a positive SWR anomaly averaged over the 4 EMSs, such a mean EMS SWR anomaly would be misleading.

The section is entitled "Differences among local extreme marine summers" as it actually shows differences between the 4 extreme summers of each location. The expression "*locally detected EMSs*" wherever used in the manuscript only aims to remind that EMSs are detected independently at each location (which is not a given). To avoid any confusion related to spatial differences, we have slightly renamed the section "Differences among extreme marine summers".

**C132.Figure 8**. I suggest changing the color pallete in this figure since red usually means higher and purple/blues lower. The red colour draws the attention a lot to the lower values of the frequency of occurrence.
**C133.Lines 556-557**: "*The areas where positive Q net anomalies in EMSs are more frequently met are the same as where the largest positive mean EMS Q net anomalies were found (Fig. 7a vs Fig. 8a).*". I am not sure this is true. For example in Fig.7a, I can see higher Qnet anomalies in the Adriatic and the northeast Aegean, which do not exactly coincide with the highest frequency of Qnet anomalies of Fig.8a. For a better comparison between these two maps, it is better to use a pattern correlation coefficient instead of just a visual

comparison. What is the point of showing both figures 7 & 8? I do not see any differences in the main message behind these two sections and the one coming after that (Section 3.3.3).

Thank you for this comment. On the point to show this analysis, please see our previous answer in C131. On the example you mention, we totally agree on what you observe, but we wouldn't really expect to see an identical behavior of a variable in every single EMS at a particular location. We considered however important to examine how the mean-EMS values are formed, and provide this information to the reader in a short sub-section. The idea of using these maps was considered a concise means for providing this information.

**C134.Lines 566-567**: "*In the same locations, SWR radiation appears increased in most EMSs, reinforcing the mean EMS findings.*". What kind of EMS findings are reinforced here exactly?

The mean-EMS findings are presented in Fig. 7. As regards SWR, we see in the discussed locations that there are positive SWR anomalies (Fig. 7c). In these lines we report that SWR is greater than usual in most EMSs (3/4 or 4/4) at these locations. If this was not the case, the mean EMS findings would not be supported in the sense that the use of a mean EMS value would be less meaningful.

**C135.*Lines 569 – 571***: *"Negative Qnet anomalies appear in every EMS in the Ligurian, the south-central as well as in a few specific spots in the WMED Sea, despite the systematically increased SWR. In these areas wind appears always reduced in EMSs (although not followed by decreased LH loss), while suppression of vertical mixing is observed only in half (or less) of the EMSs (Fig. 8c)*". I am not sure where is the figure that shows negative Qnet anomalies as Fig.8 demonstrates maps on positive Qnet anomalies only. So I am not sure why Fig.8c is cited here.Confusing, Clarify.

Thank you for your comment. Figure 8 does not show only positive anomalies. Fig. 8a shows how many EMSs present positive Qnet anomalies. The discussed areas are colored in red color which stands for 0. Therefore, in these areas the anomaly of Qnet relative to climatology (i.e., relative to the mean value of Qnet over all summers in 1950-2020) is negative in every EMS. As regards Fig. 8c, it is cited because it shows how many EMSs present negative MLD anomalies. For the discussed areas, although there are weakened winds in every local EMS (Fig. 8b), MLD anomalies are negative in half (or less) of the EMSs (Fig. 8b), suggesting that the weakened winds in EMSs are not necessarily accompanied by reduced MLD in these areas.

**C136.Lines 578 – 580**: "*...we expect that the responsible cool summer days (being warmer than usual) are mostly the early summer ones*.". The fact that the early summer days are...the cooler summer days with respect to the days during the heart of the summer .. is obvious. I do not understand the point behind this sentence.

Thank you for your comment. The cooler summer days are not necessarily the early summer ones, considering the JAS marine season. In the discussed area (in the south-central basin off the African coasts), the coldest half of the SST distribution is found to contribute the most to the seasonal SST anomalies in EMSs. But this finding alone does not inform on whether this part is formed by the

early or the late summer days. The increased mixed layer heat content before the beginning of almost every EMS in this area motivated us to further investigate the SST distribution during EMSs in the area, which confirmed that SST in July of these EMSs present greater anomalies than September. Please, see also our answers in C137 and C140.

**C137.Lines 580 – 583**: "*In all EMSs in this area, the early summer (July) SST anomalies are the largest within the season. The increased mixed layer heat content in June is reflected in the positive SST anomalies of June while, also in May, SST in this area was found marginally larger than climatologically.*" Which figures shows all this? Confusing.

Thank you for this comment. In Line 580 we note that results from this test are not included in the manuscript. We show these results below. For the colored area in the map, we may see the area-averaged seasonal cycle of SST with colored lines for the EMS-years, and gray lines for the rest of the years. The dashed line shows the mean monthly climatological SST for the reference period 1950-2020. We examined the seasonal cycle using both the original and the detrended dataset (top and bottom figure, respectively). This was done to check if the SST anomalies in June (relative to climatological values for June) before the beginning of an EMS in this area are positive also when the surface warming signal is removed. Indeed, late spring (June) SST values before EMSs are warmer than usual in this area also in the detrended dataset, i.e., independently of the warming trend, as noted in lines 589-90. Please, see also our answer in C140 below.

[Figure]

**C138.Lines 584**: "*The temporal coverage of the modelled temperature and MLD used in the OHC calculation inevitably limits this task in using...*". Which task do you refer to? Not Clear.

Thank you for noting this. We refer to the OHC calculation. The computation of the anomaly of any variable in EMSs requires that the EMS-years fall within the available period for that variable. EMS anomalies for MLD and OHC are presented only using the original dataset, as in the detrended dataset EMSs may fall out of the available period (1987-2019).

**C139.Lines 586 – 587**: "*For this reason, a complementary task included the use of the detrended dataset only for locations that experience EMSs within the available period..*"? Which one exactly is the available period, not clear.

The period 1987-2019, this has been specified in the revised manuscript.

**C140.Lines 589 – 590**:" *In the region discussed above (south-central basin), increased upper ocean heat content and spring SST values are observed before each EMS also in the detrended dataset, i.e., independently of the warming trend*". Isn't this something that one should expect given that as the summer approaches, the heat content of the ocean is ..bound to increase more and more? I am not sure why the writers expected that the detrended dataset should show "no preconditioning" or no increase of the heat content of the ocean as we move to the summer. Also how much warming above average is the preconditioning considered as "preconditioning" and not just the normal increase of the period before the EMS? Not clear this.

Thank you for this comment. The mixed layer heat content anomaly (i.e., the pre-conditioning index in this work) is computed relative to its mean climatological value (June-before-EMS – mean-June). Therefore, a positive value of this anomaly indicates a greater heat content than usual, not the normal increase of heat content as the summer approaches.
As regards the original vs the detrended dataset: In the original dataset EMSs fall within more recent years. This means that the ocean heat content anomalies before the beginning of these summers are expected to be affected by the upper ocean warming signal, which is not the case in the detrended dataset. Using either of the datasets, the proposed pre-conditioning index does not capture, by construction, the normal heating as the summer approaches, but deviations from climatology.

**C141.Lines 594 – 596**. "*Usefulness of this index is highlighted when examining single summers (e.g., the exemplary case of EMS 2015 in Sect. 3.3.4). Even if it is positive in most cases, its contribution is often enhanced when and where there is no causal link between the surface net heat balance and the observed surface warming thus revealing it constitutes an actually contributing EMS formation factor*". Does the "surface net heat balance" refer to Qnet? Keep consistent names throughout the paper. Generally it is not a very good practice to refer to a section and describe its results before the section itself is presented in the paper. It is disrupting for the reader. In addition, I have not seen anywhere so far in the paper **any causal link** between the surface Qnet (surface net heat balance?) and the SST warming. The only thing the authors have presented are visual comparisons between maps of physical variables, which are not supported by any good "statistical index" denoting the degree of similarity in their spatial distribution. So I would be careful not to use the words "causal" link, unless specific analysis has been done for specific events that prove the causal link.

Thank you for these comments. We will remove the reference to the section not yet presented to avoid disrupting the reader.
As regards Qnet: Yes, it is the surface net heat balance. We agree on the use of a single term, this has been corrected in the revised paper. The areas with positive Qnet anomalies during EMSs

experience greater heat gain during these summers than climatologically. This finding potentially suggests a causal link between the extra heat uptake and the anomalously warm sea surface but it is indeed not further supported in this section where we only present and discuss seasonal anomalies. We have revised such wording in this section, as the degree to which Qnet drives EMSs is quantified later on, using the proposed metric.

**C142.Line 597**: Apart from the fact that the entire paragraph does not belong to the Results section, as it mostly comments on the results, the writers suddenly go from Fig. 8d at the start of the paragraph, to referring Section 3.3.4 (that has not yet been presented) and then suddenly mention Fig.9d (also not presented yet) without having present figures Fig9.a-c before that. In general the Figure references should be more structured in the text and should come with an order.
**C143.Lines 598-599**:*"Although horizontal advection is not investigated in this study, this finding potentially suggests that EMS preconditioning hardly takes place in areas of intensified circulation."*. Which areas do you refer to exactly?Clarify.

These lines consider the results presented in the previous sentence for the Alboran Sea and the area surrounding the Strait of Sicily.

**C144.Section 3.3.3**

I am not sure how the results from this section differ from those in Sections 3.3.1 – 3.3.2. I understand the writers used an extra way to quantify this time the contributions from the different air-sea heat fluxes (apart from the quantification of the contribution that was necessary to begin with). But what is the added value of another yet way to see the contributions of the different air-sea heat fluxes to the EMS? I do not understand, qualitatively, what is the new knowledge we gained in Section 3.3.3 compared to Section 3.3.2. and 3.3.1. For example, in line 608, where the contributions are discussed, the Fig.7g is also cited to support the hypothesis that was also evident from the quantified contributions themselves. Probably this quantification metric is more meaningful than the descriptions from Section 3.3.1 and 3.3.2. However, at the moment the metric is not very clear in its explanation in the Appendix. It is quite difficult to understand how the writers constructed it, e.g. what is the difference between SSTA cum >0 (which is stated as SST anomaly in the Appendix), with the ΔSST′obs?Is the second data anomalies fromSST′obs?Is the second data anomalies from
observations only? It is very confusing.
In addition, before the writers start describing the PEMS of Fig.9, they should give a qualitative definition at the start of the section to the readers as to what each positive or negative PEMS means and then continue with the description of their spatial destribution. At the moment the PEMS meaning is scattered here and there when the spatial distributions are analysed.

Thank you for your comment.
On the distinct content of 3.3.1, 3.3.2 and 3.3.3:
Section 3.3.1 presents seasonal anomalies of the examined variables (Qnet, its components, MLD and OHC for pre-conditioning) for EMSs relative to climatology. Section 3.3.2 informs on

differences in the behavior of the examined variables between the 4 EMSs at each location. Section 3.3.3 quantifies the role of air-sea heat fluxes in the formation of EMSs (It does not inform on the different air-sea heat fluxes). The seasonal anomalies for the examined variables in 3.3.1 are important findings as we aim to investigate potential drivers and concurrent conditions, but they do not answer the question: "what drives the formation of EMSs". The added value of 3.3.3 is that it actually quantifies the driving role of air-sea heat fluxes in forming EMSs.

In Line 606-608 we refer to areas where MLD and WSPD seasonal anomalies are negative, which suggests wind-driven shoaling of the mixed layer. In these areas, $0 < P_{EMS} < 100$, which suggests that air-sea heat fluxes are not the only EMS driver. Therefore, we considered adequate citing such information on the sub-surface during the same EMSs in these areas.

On the metric construction:
$SSTA_{cum}$ during a summer sub-period (from day $t_0$ to day $t_1$) is the cumulative SST anomaly used to separate sub-periods according to their contribution to the mean seasonal SST. We only work with sub-periods of positive $SSTA_{cum}$. $SSTA_{cum}$ values are also used later on to weight the contribution of each sub-period when producing a single metric value for the season.
As presented in Appendix, it is computed as follows:

$$SSTA_{cum} = \sum_{t_0}^{t_1}(SST_t - SST_{tclim})$$

where $SST_t$ and $SST_{t_{clim}}$ the daily SST of day $t$ and the corresponding climatological day $t_{clim}$, respectively. Indices $t_0$ and $t_1$ stand for the start and end day of each phase, respectively.

$\Delta SST'_{obs}$ during a summer sub-period is the change in daily SST anomaly:

$$\Delta SST'_{obs} = SST'_{t_1} - SST'_{t_0} = (SST_{t_1} - SST_{t_1clim}) - (SST_{t_0} - SST_{t_0clim})$$

where $SST'_{t_0}$ and $SST'_{t_1}$ are the daily SST anomalies at the start and the end day, respectively.

The subscript *obs* stands for observed (based on the reanalysis SST).

We have added the word cumulative when referring to $SSTA_{cum}$, as calling it SST anomaly is indeed confusing. We also agree on your suggestion to include at the start of Sect. 3.3 a qualitative definition on what positive or negative PEMS means. This has been included in the revised manuscript.

**C145.Line 606**: "*In positive P EMS areas with 0 < P EMS <100%, other mechanisms are expected to work complementarily (i.e., towards greater SST anomalies)*". What does complementarily mean? What does it mean towards greater SST anomalies?

Thank you for this comment. Please, see our answer in C144 for Line 606.
In areas where $0 < P_{EMS} < 100$ we expect additional mechanisms to contribute to the observed SST anomalies in EMSs. The metric focuses on summer sub-periods where SST changes towards greater anomalies. This means that during these sub-periods the SST either increases at a greater

rate than usual or decreases at a slower rate than usual. In other words, when this criterion is satisfied (mathematically summarized in $\Delta SST'_{obs} > 0$) the SST anomaly of the last day of the examined period is always higher than the SST anomaly of the first day of that period. Therefore, during the selected sub-periods SST changes towards greater anomalies. In areas where $0 < P_{EMS} < 100$ we expect additional mechanisms (on top of the air-sea heat fluxes) to contribute to these SST changes, i.e., to drive these SST changes together (complementarily).

**C146.Line 612**: "*..hence oceanic processes definitely drive EMSs in these areas*". I would avoid statements with "definitely" since the contributions from oceanic processes have not been actually proven here. They have just been inferred.

**C147.Figure 9**: I am not sure what panels b), c) e) f) are showing exactly. What is the "warming phase" and "cooling phase" that are being subtracted here? How are they represented mathematically? Also I do not understand what is the use of an index where the warming and cooling phase are being subtracted. Clarify. In addition, in the citation of the figure, the writers should clarify the qualitative meaning of the PEMS metric and not just refer to it like : " P*EMS metric values (%) for the contribution….*". The citation is supposed to make the reader understand what are they looking at. At the moment I only understand that I am seeing some percentages of contribution.

Thank you for this comment. The process to isolate warming/cooling phases and compute the metric is presented step-by-step in the Appendix A. The mathematical explanation of the metric's computation separately for the warming and cooling phases is provided in Lines 865-872. There is not any "subtraction" of warming or cooling phases.

As explained in the manuscript, the selected warming (cooling) phases are the summer sub-periods where SST increases (decreases) at a greater (lower) rate than usual. The purpose of this additional task was to examine the role of heat flux in EMSs, separately during the selected warming and the selected cooling phases. We totally agree that info on the qualitative meaning of the metric should be included in the Figure's caption. The caption has been updated accordingly.

**C148.Lines 627, 639**: 1) "*These values indicate that oceanic processes are primarily responsible for the observed EMS SSTs in these areas*", "*Hence high SST anomalies leading to EMSs in these areas are formed despite the thermal energy deficit at the sea surface during the same periods.*"These sentences are confusing. You are saying that there are EMSs happening but..there is thermal energy deficit. How is this possible? Clarify.

Thank you for this comment. Negative $P_{EMS}$ indicates that the surface heat fluxes do not work in favor of the formation of EMSs. In such cases, the air-sea heat exchange corresponds to a thermal energy deficit (always relative to climatological values for the examined period of the year). We should have specified in the quoted sentence that we refer to the energy deficit at the sea surface due to non-favoring (for the formation of EMSs) heat exchange with the atmosphere, not an energy deficit in general. This has been corrected in the revised manuscript.

**C149.Lines 631 – 633**: "T*he present section complements this finding by revealing that a thermal energy deficit at the sea surface is consistently observed in these areas even*

*when focusing solely on highly effective (for the EMS formation) warming phases*".
Rephrase.
**C150.Line 650:** "*Results also reveal a strong link between MHW properties and surface heat fluxes (Fig. 9a vs Fig. 6d)*". I think this is a proven statement, multiple times already??? Why do you need to prove it again?

Thank you for this question. Results from the proposed metric that we discuss here are based on the summer sub-periods that are the most responsible for the EMS SST anomalies. The link we report in this sentence is indeed not revealed here for the first time, it is however further supported in relation to the formation of EMSs.

**C151.Lines 654- 655**: "*The spatial similarity of MHW intensity and PEMS over the basin suggests that the crucial role of surface heat fluxes found for the northern MS is associated with their ability to drive high SST anomalies (thus intense MHWs)*". I still cannot understand the difference between EMS and MHWs in terms of high SST anomalies. They are both referring to high SST anomalies in general.

Thank you for this comment. Please, see our answers in C8.

**C152.Section 3.3.4**
The example of the Marine summer 2015 kind of suggests that there is not much of a difference between EMSs and MHWs. More specifically, Lines 569 – 661, are justifying the exploration of the EMS 2015, based on a citation about a MHW in the summer of 2015!. The writers need to either clarify what is the difference of EMS from the MHWs and thus, their usefulness compared to the MHWs or re-think the idea of EMS in total. In principle, these two definitions demonstrate two different ways to capture extreme warm temperatures in the ocean. Either, use one of them without comparison with the other or re-think the idea of the EMS.

Please, see our answers in C8.

**C153.Lines 679-681**: "*On the other hand, the preconditioning index is clearly more pronounced here than in the positive part (Fig. 10d). Indeed, early summer days (July 2015) present...*". I thought the early summer days of the preconditioning phase was the month of June according to Section 2.2.5

Thank you for this comment. The months Jul-Aug-Sep have been considered as marine summer period, so early summer days fall within July. The considered preconditioning phase is the month June as explained in Methods.

**C154.Lines 685-686**: "*Following 2015, other widely experienced EMSs such as 2003, 2012, 2018 (in descending order by means of spatial extent) are examined (not shown).*". These years also constitute MHW years in the Med. Sea. So I cannot see the usefulness/added value of the EMS definition in identifying extreme ocean periods when compared with the MHW definition.

Thank you for this comment. Please, see our answers in C8.

**C155.Section 3.3.5**
If the writers re-think the idea of the EMS and justify its usefulness as an index then the analysis of the contribution of air-sea heat fluxes in the detrended datasets deserves to be shown as a comparison to the equivalent results with the original dataset and its figures should be juxtaposed and commented in the main text.

Thank you for this suggestion. As noted in C13, we have moved the analysis of heat fluxes based on the detrended dataset right after the equivalent analysis of the actual data (3.3.1). We have kept the corresponding figure in the Appendix, to avoid interrupting the manuscript flow as well as because this figure in the form that is currently presented in the Appendix already juxtaposes results from the 2 datasets.

**Conclusions and Summary:**
**C156.Lines 794- 796**: "*Nevertheless, the multi-decadal signal contributes to the observed surface warming through more intense and slightly longer lasting MHWs, without affecting the MHW intensity spatial distribution, thus most probably also the event driving mechanisms*". I do not understand how the multi-decadal signal contributes to more intense events without affecting the intensity spatial distribution and the event driving mechanisms. How do you infer that the event-driving mechanism is not affected? Clarify.

Thank you for this comment. Please, see our relevant answer in C106. This sentence has been removed as it is not sufficiently supported.

---

## Author Response (AR2)

**Reply to Reviewer and changes in the revised manuscript**

**Reviewer's comments:**

Having read the authors' response to my previous review I still have doubts about the authors' definition of EMS. Although in general they have satisfactorily answered or clarified most of the issues I raised in the review I cannot yet recommend the publication of this paper.

Thanks to the authors' detailed explanations I have clearly understood their definition of EMS but I must say that, even understanding their intention, I think their concept of EMS is incomplete to my understanding and I do not see the usefulness of its application in research on SST and MHWs in the Mediterranean.

First of all, reading the title, the reader would expect a definition that includes both temporal and spatial aspects, since it speaks of "Mediterranean Sea". The fact that the definition is based on a grid point approach without any spatial consideration, even if it is posterior, I believe detracts from the interest or usefulness of the EMS concept. The definition, as it stands, may lead to a theoretical situation, although unlikely but not impossible by the way it is defined, where the 4 years qualified as EMS may be different at each neighbour point in the grid, which makes the EMS concept less consistent. Knowing about the spatio-temporal coherence of the SST, it is possible that in a large part of the points of the grid the years qualified as EMS coincide, reducing the theoretically possible variability, but this is something that the authors should check/show in their results. In fact, in the figures representing three points in the Mediterranean there are few coincidences in the EMS years. I do not feel comfortable looking at a map of, for example, mean SST anomaly without knowing if the averages at nearby, or distant, points have been made over the same years (or a short series of years). The authors should report whether the years considered EMS are a short distribution or whether there is a large set of EMS years with large variability. A suggestion would be to evaluate the number of grid points having an EMS each year to look for years concentrating the highest number of grid points.

Moreover, the definition would only be consistent for the period under study. In the years 2022 and 2023, very intense and extensive SST anomalies and MHWs have been recorded in the Mediterranean that would most likely fall into the group of 4 detectable EMSs in the analysis. Only a slight extension of the series could change the nature of 50% of the EMS over large areas of the basin. This is a major weakness of the definition even though the authors have performed analyses that indicate that the spatial distribution or substructure of SSTs does not change, which is interesting. But the EMS years could change. From such a definition one would expect that subsequent years could be analysed and determined to be extreme or non-extreme without affecting the qualification of previous years, at least not 50% of them.

At this point, I cannot recommend the publication of this article unless the authors consider introducing a requirement for spatial coherence. In this regard I recommend the publication of one of the co-authors "Flaounas, E., et al (2021). Extreme wet seasons – their definition and

relationship with synoptic-scale weather systems. Weather and Climate Dynamics", specifically section 2.2 "Spatial and temporal coherence of extreme wet seasons" for a similar analysis.

I encourage you to run the analysis suggested as an improved definition of EMS could be an interesting/important metric for SST analysis both for scientists and stakeholders.

**Authors' response:**

We would like to thank the Reviewer for their careful reading and the fruitful comments. In the following we reply to each one of the queries.

This study focuses on the Mediterranean Sea and therefore the title accurately provides the geographical boundaries of our research. Regarding the grid-point definition of EMSs, our approach is primarily -along with its starting point- statistical. An EMS is therefore defined as a period with a very low probability of occurrence. This allows us to investigate separately each grid point with regards to the substructures of its own distribution of SSTs. Such a statistical analysis would be valid either we considered each grid point, different sub-regions of the basin or the Mediterranean Sea as a whole.

An alternative method as the "patches-approach" in Flaounas et al. (2021) suggested by the Reviewer would be also feasible. Nevertheless, this would be a data-driven approach and thus fundamentally different since it would prohibit a statistical analysis of substructures as done in this study. Indeed, the formation of patches with coherent physical (or SST) characteristics would be a data-driven one, connecting seasonally biased SST distributions and making it hard to consider EMSs as events of equivalently low-probabilities among the patches. We agree with the Reviewer that the spatio-temporal coherence of the SST would hardly make neighbor grid points to be considered as independent but here we investigate whether the low probability of forming an EMS has a common statistical origin (e.g. due to the warmer SSTs being warmer than usual). With this, we quantify the statistical origin of EMSs across the Mediterranean basin, while the second part of our study aims to physically interpret our statistical results of the first part. To be more clear on the statistical motivation of our approach we now enrich the explanations on the EMS definition in Methods.

Given our reply on the statistical motivation of our methodological approach, a question might be raised about whether EMSs are indeed extraordinary events with large spatial coverage and significant impact on the region. To address this question, we follow the Reviewer's suggestion and we first present the spatial extent of all identified EMSs in terms of their percentage coverage of the basin, illustrated in the bar graph in Fig. 1. As anticipated, the EMS-years are predominantly concentrated in the post 2000 period. Some exceptions during the 50-60's are largely attributed to the multi-decadal oscillation of the Mediterranean SST, as minimum SST values within the study period 1950-2020 occur during the 70's (Fig. 2a in the manuscript). None of the years within 1963-1990 have been characterized as extreme in any location within the basin.

The EMS-years presenting the greatest spatial coverage concentrating more than 50% of the grid points, are 2015, 2003, 2018 and 2012, covering 72% of the basin, 58%, 54% and 53%, respectively. These important (form a basin-wide perspective) EMSs exhibit different spatial patterns (Fig. 2). Such spatial variations among EMSs and across the basin are naturally expected, as diverse factors acting at different spatial and temporal scales within the basin may lead to above-threshold mean summer local SST values. For example, in Sect. 3.3.5 (Illustrative example of EMS 2015), we demonstrate different mechanisms operating over the western and the central/eastern basins during summer 2015 (this summer has not been qualified as EMS in only 28% of the basin). To address the Reviewer's query on the spatial coherence of SST during the aforementioned summers, Fig. 2 shows the locations experiencing the same EMS.

The spatial coherence of EMSs in Fig. 2 is rather evident, however, enclosed sub-basins and areas more susceptible to typical wind regimes (e.g., Aegean Sea and the Gulf of Lions) or areas of distinct circulation patterns (e.g., Alboran Sea), are more prone to the effect of local processes; therefore, they are further expected to differentiate from neighboring areas. For those interested in understanding the physical mechanisms underlying the spatial spread of extreme warm conditions during a specific EMS, the role of local processes in spreading or dissipating warm anomalies during that summer should be investigated across the areas of interest, in line with their objectives (beyond the scope of the current work). The per grid point analysis we follow here captures the extreme conditions experienced at local scale. Transition from EMS to non-EMS neighboring points is expected, as with any peak-over-threshold approach. This does not contradict the spatial coherence of SST, as observed in Fig. 2; it merely reflects that warm conditions are not equally extreme across different –even neighboring– areas.

We now present in the manuscript the spatial coverage of the most extended EMS conditions, (maps of Fig. 2 depicting areas experiencing specific EMSs). In this way, the reader will have a clearer idea on the EMS detection output based on this additional information on the spatial extent of the most significant EMS cases.

As previously noted in this reply document, we regard an EMS as an event of low probability. The question that the Reviewer raises with their query is whether EMSs are also to be statistically consistent with similar substructures and similar interpretations. Our analysis as correctly pointed out by the Reviewer shows that indeed EMSs are statistically consistent. The spatial distribution of SST substructures remains unchanged when we apply our analysis on the entire study period, parts of the study period, or based on the detrended dataset. In all these cases we observe the same statistical behavior: local SST substructures remain the same despite the different sets of 4 EMSs detected at each grid point. In the manuscript, this is evident through the comparison of the SST substructure fields derived from the original vs the detrended dataset (Fig. 3a-c vs Fig. 3d-f of the manuscript). The example distributions of Fig. 4 of the manuscript illustrate this in more detail. They show that the part of the SST distribution contributing the most to the seasonal SST anomaly at a specific location is the same in the original and the detrended dataset although three out of the four EMS-years differ between the two datasets (e.g., the anomalously warm colder summer days in the southern-central basin; Fig. 4c vs Fig. 4f of the manuscript). This finding is important, implying inherent characteristics of the anomalously high SST during summers in the

Mediterranean basin. More than that our results suggest that inherent characteristics of EMSs are unrelated to the specificities of individual summers or the climate-change warming signal. Therefore, including data from 2022 is not expected to significantly alter our results even if the EMSs are not the same.

[Figure]

*Figure 1: Percentage of Mediterranean Sea grid points experiencing EMS conditions*

[Figure]

*Figure 2: EMSs concentrating more than 50% of the Mediterranean Sea grid points: 2003, 2012, 2015 and 2018 (from top to bottom, respectively). Yellow stands for areas experiencing EMS conditions and blue for non-EMS conditions*

**Changes in the revised manuscript:**

To be more clear on the statistical motivation of our approach and provide additional information on the spatial extent of the most significant EMS cases, the explanations on the EMS definition and detection in Methods have been further enriched as follows.

Lines 147-154 (in the revised manuscript):

*(...) This approach captures periods with a low probability of occurrence experienced locally, allowing for a statistical analysis of their properties in a consistent way across the basin, based on the methodology of Röthlisberger et al. (2020). In this way we investigate separately each grid point with regards to the substructures of its own distribution of SSTs. To offer the reader an understanding on the EMS detection output regarding the spatial extent of the most significant EMS-years, Fig. B1 illustrates the spatial extent of EMSs concentrating at least 50% of the MS grid points (2015, 2003, 2018 and 2012, covering 72% of the basin, 58%, 54% and 53%, respectively).*

Figure 2 (depicting areas experiencing the most extended EMSs) has been included in the supplementary figures of Appendix B (Fig. B1)